# Infinite Limits of Multi-head Transformer Dynamics

**Blake Bordelon, Hamza Chaudhry, Cengiz Pehlevan**
John A. Paulson School of Engineering and Applied Sciences
Center for Brain Science
Kempner Institute for the Study of Natural and Artificial Intelligence
Harvard University
Cambridge, MA 02138
`blake_bordelon@g.harvard.edu`
`hchaudhry@g.harvard.edu`
`cpehlevan@seas.harvard.edu`

## Abstract

In this work, we analyze various scaling limits of the training dynamics of transformer models in the feature learning regime. We identify the set of parameterizations that admit well-defined infinite width and depth limits, allowing the attention layers to update throughout training–a relevant notion of feature learning in these models. We then use tools from dynamical mean field theory (DMFT) to analyze various infinite limits (infinite key/query dimension, infinite heads, and infinite depth) which have different statistical descriptions depending on which infinite limit is taken and how attention layers are scaled. We provide numerical evidence of convergence to the limits and discuss how the parameterization qualitatively influences learned features.

## 1   Introduction

Increasing the scale of transformer models has continued to improve performance of deep learning systems across many settings including computer vision [1, 2, 3, 4] and language modeling [5, 6, 7, 8, 9]. However, understanding the optimization stability and limiting behavior of these models under increases in model scale remains a core challenge.

One approach to scaling up systems in a stable and predictable way is to identify parameterizations of neural networks that give approximately scale-independent feature updates during training [10, 11, 12]. The mean field parameterization, commonly referred to as $\mu$P, is a well-known example that satisfies this property [13, 14, 15]. When such parameterizations are adopted, the learned internal representations in hidden layers of the network are very similar across model scales [16, 17], but performance tends to improve with model scale [10, 11, 12]. Further, theoretical results about their limits can often be obtained using Tensor Programs [14] or dynamical mean field theory (DMFT) techniques [15, 17].

In this work, we develop a theoretical treatment of randomly initialized transformers. We study various scaling limits of the training dynamics of these models including the infinite key/query dimension limit, the infinite head limit, and the infinite depth limit. Concretely, our contributions are the following:

1. We derive a DMFT for feature learning in randomly initialized transformers with key/query dimension $N$, attention head count $\mathcal{H}$ and depth $L$. From the derived DMFT action, we identify large $N$, large $\mathcal{H}$ and large $L$ limits of the training dynamics.

38th Conference on Neural Information Processing Systems (NeurIPS 2024).

2. We analytically show that the large key-query $N \to \infty$ limit requires the $\mu$P scaling of key/query inner product with $1/N$, even if key/queries are reparameterized to decrease the size of their updates from gradient descent.

3. From the limiting equations, we show that this $N \to \infty$ limit causes multi-head self attention trained with stochastic gradient descent (SGD) to effectively collapse to single-head self attention since all heads follow identical dynamics.

4. To overcome this limitation, we analyze the infinite head $\mathcal{H} \to \infty$ limit while fixing $N$. We show there is a limiting *distribution* of attention variables across heads at each layer throughout training. Despite $N$ being finite, the infinite-head $\mathcal{H} \to \infty$ limit leads to concentration of the network's output logits and learned residual stream feature kernels, giving deterministic training dynamics.

5. Finally, we examine large depth limits of transformers with residual branch scaling. We illustrate and discuss the tension between parameterizing a model so that it has a non-trivial kernel at initialization while maintaining feature learning within the multi-head self attention (MHSA) and multi-layer perceptron (MLP) blocks.

## 1.1 Related Works

Hron et al. [18] studied the Neural Network Gaussian Process limit of multi-head self attention in the infinite-head $\mathcal{H} \to \infty$ limit. They showed that, at initialization, there is a limiting distribution over attention matrices and that the outputs of the multi-head attention block follow a Gaussian process, establishing a connection to kernel methods. Dinan et al. [19] develop a similar theory of transformers at initialization and compute the Neural Tangent Kernel associated with this architecture as the dimensions per head $N \to \infty$ using a $\frac{1}{\sqrt{N}}$ scaling of the key-query inner product within each attention layer. One of our key theoretical results is showing that this picture of a *distribution over learned attention heads* persists throughout training in the feature-learning regime as $\mathcal{H} \to \infty$ (though the distribution of residual stream variables generally becomes non-Gaussian).

Several works have analyzed the signal propagation properties of transformers at initialization at large key/query dimension $N$ and large depth $L$ [20, 21, 22, 23] including providing modifications to the standard transformer architecture [22, 24]. In this work, we pursue large depth limits of transformers by scaling the residual branch as $L^{-\alpha_L}$ with $\alpha_L \in [\frac{1}{2}, 1]$, which has been shown to converge to a limit not only at initialization [25, 26, 27], but also throughout training in the feature learning regime [11, 12, 27]. However, we argue that in transformers that $\alpha_L = 1$ is preferable as it enables the attention layers to update non-negligibly as $L \to \infty$.

Yang et al. [10] introduced the $\mu$P scaling for attention layers which multiplies the key/query inner product with $\frac{1}{N}$ rather than the more commonly used $\frac{1}{\sqrt{N}}$ [5]. They show empirically that this change improves stability of training and transfer of optimal hyperparameters across different values of $N$. Vyas et al. [16] empirically found that such $\mu$P transformers learn attention matrices that become approximately consistent across different heads and model sizes, suggesting that models parameterized in $\mu$P learn similar representations across scales.

In addition to work on infinite width and depth limits of deep networks, there is also a non-asymptotic approach to optimizer design and scaling based on controlling the norm of weight updates [28]. This approach coincides with $\mu$P width-scaling when the spectral norm of the weights is used as the measure of distance [29], and can achieve hyperparameter transfer for a wide array of optimizers and initialization schemes [30**?** ].

## 2 Parameterizations with Feature Learning Limits

We consider a transformer architecture with $L$ layers, $\mathcal{H}$ heads per layer, and $N$ dimensional keys/ queries per head. Transformers are often defined in terms of $d_{\text{model}} = \mathcal{H} d_{\text{head}} = \mathcal{H} N$ which can be increased by scaling the number of heads or the dimension of each head, where $N$ is often written $d_{\text{head}}$. Our goal is to determine the set of parameterizations that allow the attention layers to undergo non-trivial feature learning in the various $N, \mathcal{H}, L \to \infty$ limits. The analysis of these limits is performed with batch size and number of training steps $t$ fixed while the other architectural parameters are taken to infinity.

## 2.1 Model Scalings

The network's output is computed by a depth $L$ recursion through hidden layers $\ell \in [L]$ starting with the first layer $h^1_{\mathfrak{s}}(x) = \frac{1}{\sqrt{D}}W^0 x_{\mathfrak{s}} \in \mathbb{R}^{N\mathcal{H}}$ where $x_{\mathfrak{s}} \in \mathbb{R}^D$ is the input at spatial/token position $\mathfrak{s}$. Preactivations in subsequent layers $h^\ell$ are determined by a forward pass through the residual stream which contains an attention layer and a MLP layer

$$h^{\ell+1}_{\mathfrak{s}} = \tilde{h}^\ell_{\mathfrak{s}} + \frac{\beta_0}{L^{\alpha_L}}\text{MLP}\left(\tilde{h}^\ell_{\mathfrak{s}}\right) \ , \ \tilde{h}^\ell_{\mathfrak{s}} = h^\ell_{\mathfrak{s}} + \frac{\beta_0}{L^{\alpha_L}}\text{MHSA}\left(h^\ell\right)_{\mathfrak{s}} . \tag{1}$$

The constants $\gamma_0$ and $\beta_0$ control the rate of feature learning and the *effective depth* respectively [1]. We will consider $\alpha_L \in [\frac{1}{2}, 1]$.[2] The multi-head self attention layer (MHSA) with pre-layer-norm[3] is

$$\text{MHSA}\left(h^\ell\right)_{\mathfrak{s}} = \frac{1}{\sqrt{N\mathcal{H}}}\sum_{\mathfrak{h}\in[\mathcal{H}]}W^\ell_{O\mathfrak{h}}v^{\ell\sigma}_{\mathfrak{h}\mathfrak{s}} \ , \quad v^{\ell\sigma}_{\mathfrak{h}\mathfrak{s}} = \sum_{\mathfrak{s}'\in[\mathcal{S}]}\sigma^\ell_{\mathfrak{h}'\mathfrak{s}\mathfrak{s}'}v^\ell_{\mathfrak{h}'\mathfrak{s}'}$$

$$v^\ell_{\mathfrak{h}\mathfrak{s}} = \frac{1}{\sqrt{N\mathcal{H}}}W^\ell_{V\mathfrak{h}}\bar{h}^\ell_{\mathfrak{s}} \ , \quad \bar{h}^\ell_{\mathfrak{s}} = \text{LN}(h^\ell_{\mathfrak{s}}), \tag{2}$$

where $\sigma^\ell_{\mathfrak{h}} \in \mathbb{R}^{\mathcal{S}\times\mathcal{S}}$ are the attention matrices passed through a matrix-valued nonlinearity $\sigma\left(\mathcal{A}^\ell_{\mathfrak{h}}\right)$.[4] For a sequence of length $\mathcal{S}$, the pre-attention matrix $\mathcal{A}^\ell_{\mathfrak{h}} \in \mathbb{R}^{\mathcal{S}\times\mathcal{S}}$ is defined as

$$\mathcal{A}^\ell_{\mathfrak{h}\mathfrak{s}\mathfrak{s}'} = \frac{1}{N^{\alpha_\mathcal{A}}}k^\ell_{\mathfrak{h}\mathfrak{s}} \cdot q^\ell_{\mathfrak{h}\mathfrak{s}'} \ , \quad k^\ell_{\mathfrak{h}\mathfrak{s}} = \frac{1}{N^{\frac{3}{2}-\alpha_\mathcal{A}}\sqrt{\mathcal{H}}}W^\ell_{K\mathfrak{h}}\bar{h}^\ell_{\mathfrak{s}} \ , \quad q^\ell_{\mathfrak{h}\mathfrak{s}} = \frac{1}{N^{\frac{3}{2}-\alpha_\mathcal{A}}\sqrt{\mathcal{H}}}W^\ell_{Q\mathfrak{h}}\bar{h}^\ell_{\mathfrak{s}}. \tag{3}$$

The exponent $\alpha_\mathcal{A}$ will alter the scale of the pre-attention variables $\mathcal{A}^\ell_{\mathfrak{h}}$ at initialization. The input matrices have shape $W^\ell_{V\mathfrak{h}}, W^\ell_{K\mathfrak{h}}, W^\ell_{Q\mathfrak{h}} \in \mathbb{R}^{N\times N\mathcal{H}}$, while the output matrices have shape $W^\ell_{O\mathfrak{h}} \in \mathbb{R}^{N\mathcal{H}\times N}$. All of the weights $W^\ell_{O\mathfrak{h}'}, W^\ell_{Q\mathfrak{h}}, W^\ell_{K\mathfrak{h}}$ are initialized with $\Theta(1)$ entries while $W^\ell_{K\mathfrak{h}}, W^\ell_{Q\mathfrak{h}}$ have entries of size $\Theta(N^{1-\alpha_\mathcal{A}})$ which ensures that all key and query $k, q$ vectors are $\Theta(1)$ at initialization. The pre-attention variables $\mathcal{A}^\ell_{\mathfrak{h}} \in \mathbb{R}^{\mathcal{S}\times\mathcal{S}}$ at each head $\mathfrak{h}$ are determined by key $k^\ell_{\mathfrak{h}\mathfrak{s}}$ and query $q^\ell_{\mathfrak{h}\mathfrak{s}'}$ inner products. The MLP layer consists of two linear layers with an element-wise nonlinearity $\phi$ applied in between, where $W^{\ell,2}, W^{\ell,1} \in \mathbb{R}^{N\mathcal{H}\times N\mathcal{H}}$ are initialized with $\Theta(1)$ entries:

$$\text{MLP}(\tilde{h}^\ell_{\mathfrak{s}}) = \frac{1}{\sqrt{N\mathcal{H}}}W^{\ell,2}\phi\left(\tilde{h}^{\ell,1}_{\mathfrak{s}}\right) \ , \ \tilde{h}^{\ell,1}_{\mathfrak{s}} = \frac{1}{\sqrt{N\mathcal{H}}}W^{\ell,1}\bar{\tilde{h}}^\ell_{\mathfrak{s}} \ , \ \bar{\tilde{h}}^\ell_{\mathfrak{s}} = \text{LN}\left(\tilde{h}^\ell_{\mathfrak{s}}\right). \tag{4}$$

$\mu$P scaling [13, 31, 14, 15] downscales the readout of the last layer compared to standard and NTK parameterization [32]. Thus, we define the output of the model as

$$f = \frac{1}{\gamma_0 N\mathcal{H}}w^L \cdot \left(\frac{1}{\mathcal{S}}\sum_{\mathfrak{s}}h^L_{\mathfrak{s}}\right) \tag{5}$$

[5] where $h^L_{\mathfrak{s}} \in \mathbb{R}^{N\mathcal{H}}$ are the final layer preactivations at spatial/token position $\mathfrak{s} \in [\mathcal{S}]$. The parameter $\gamma_0$ is an additional scalar that controls the rate of change of the internal features of the network relative to the network output [33].

## 2.2 Learning Rate Scalings

In order to approximately preserve the size of internal feature updates, we must scale the learning rate $\eta$ appropriately with $(N, \mathcal{H}, L)$. However, this scaling depends on the optimizer. In Table 2, we provide the appropriate scaling of learning rates for SGD and Adam for any $\alpha_L \in [\frac{1}{2}, 1]$ and $\alpha_\mathcal{A} \in [\frac{1}{2}, 1]$. In what follows, we focus on the SGD scaling and theoretically analyze the $N \to \infty$, $\mathcal{H} \to \infty$, and $L \to \infty$ limits of the training dynamics. We also provide in Table 2 details about what prefactor the first layer should be multiplied by and the initial weights divided by to ensure convergence to the $L \to \infty$ limit. Example FLAX implementations of these parameterizations for vision and language modeling transformers are provided in Appendix B.

---

[1]The scale of the update to the residual stream due to each layer will be $\mathcal{O}(\gamma_0\beta_0^2/L)$.

[2]If $\alpha_L < \frac{1}{2}$ or $\alpha_\mathcal{A} < \frac{1}{2}$ some of the forward pass variables would diverge at initialization as $L \to \infty$ or $N \to \infty$ respectively. If $\alpha_L > 1$ then updates to internal residual blocks will diverge as $L \to \infty$.

[3]The LN denotes layer-norm which we let be defined in terms of each vector's instantaneous mean and variance $\text{LN}(h) = \frac{1}{\sqrt{\sigma^2+\epsilon}}(h - \mu\mathbf{1})$ where $\mu = \frac{1}{N\mathcal{H}}\mathbf{1} \cdot h$ and $\sigma^2 = \frac{1}{N\mathcal{H}}|h - \mu\mathbf{1}|^2$.

[4]The nonlinearity is generally softmax. For autoregressive tasks, it is taken to be causal.

[5]Instead of pooling over space, outputs $f$ could also carry spatial index $\mathfrak{s}$ (such as next word prediction).

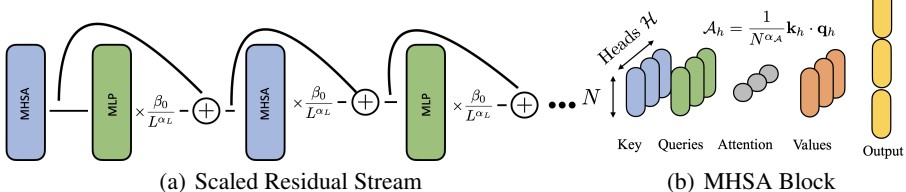

| | (a) Scaled Residual Stream | (b) MHSA Block |

Figure 1: Schematic representations of the transformer architecture we model. (a) The forward pass through the residual stream is an alternation of MHSA and MLP blocks scaled by $\beta_0 L^{-\alpha_L}$. (b) The MHSA block computes keys, queries, values, and attention variables to produce a concatenated output of dimension $d_{\text{model}} = N\mathcal{H}$.

| Optimizer | Global LR | First/Last Layer Rescale Multiplier | First/Last Layer Std. Dev. |
|-----------|-----------|-------------------------------------|----------------------------|
| SGD | $\eta_0 N\mathcal{H} L^{2\alpha_L-1}$ | $L^{\frac{1}{2}-\alpha_L}$ | $\Theta(1)$ |
| Adam | $\frac{\eta_0}{\sqrt{N\mathcal{H}}} L^{-1+\alpha_L}$ | $L^{1-\alpha_L}\sqrt{N\mathcal{H}}$ | $\frac{1}{\sqrt{N\mathcal{H}}} L^{-1+\alpha_L}$ |

Table 1: The learning rates which should be applied to obtain the correct scale of updates for SGD or Adam optimizers. In addition, the weight variance and multiplier for the first layer may need to be rescaled (relative to eq (5)) with width/depth depending on the parameterization and optimizer.

Our analysis assumes that at each step $t$ of SGD or Adam a mini-batch $\mathfrak{B}_t$ of size $\Theta(1)$ is used to estimate the loss gradient. We assume that the minibatches are fixed. Further, the number of total training steps is assumed to not be scaled jointly with the model size. Therefore the analysis provided here can cover both online SGD for a fixed number of steps or full batch GD (repeating data) with a $\Theta(1)$ sized dataset.

## 3 Infinite Limits of Learning Dynamics

In this section, we first analyze the infinite dimension-per-head $N \to \infty$ limit of training. We find that for this limit, the $\mu$P rule of $\alpha_{\mathcal{A}} = 1$ is necessary and show that all heads collapse to the same dynamics. To counteract this effect, we next analyze the infinite head $\mathcal{H} \to \infty$ limit of the training dynamics at fixed $N, L$, where we find a limiting *distribution* over attention heads. We will conclude by analyzing the statistical descriptions of various infinite depth $L \to \infty$ limits. [6].

### 3.1 Mean Field Theory Treatment of the Learning Dynamics

To obtain the exact infinite limits of interest when scaling dimension-per-head $N$, the number of heads $\mathcal{H}$, or the depth $L$ to infinty, we work with a tool from statistical physics known as dynamical mean field theory (DMFT). Classically, this method has been used to analyze high dimensional disordered systems such as spin glasses, random recurrent neural networks, or learning algorithms with high dimensional random data [34, 35, 36, 37, 38, 39]. Following [15, 11], we use this method to reason about the limiting dynamics of randomly initialized neural networks by tracking a set of deterministic correlation functions (feature and gradient kernels) as well as additional linear-response functions (see Appendix D). The core conceptual idea of this method is that in the infinite limit and throughout training, all neurons remain statistically independent and only interact through collective variables (feature kernels, neural network outputs, etc). Further the collective variables can be computed as *averages* over distribution of neurons in each hidden layer or along the residual stream. This DMFT description can be computed using a path integral method that tracks the moment generating function of the preactivations or with a dynamical cavity method (see Appendix D).

### 3.2 Scaling Dimension-Per-Head $N$

One way of obtaining a well-defined infinite parameter limit of transformers is to take the $N \to \infty$ limit, where $N$ is the dimension of each head. A priori, it is unclear if there are multiple ways of scaling the key/query inner product. Concretely, it is unknown what values for the exponent $\alpha_{\mathcal{A}}$ are admissible for the pre-attention $\mathcal{A} = \frac{1}{N^{\alpha_{\mathcal{A}}}} \boldsymbol{k} \cdot \boldsymbol{q}$. The keys and queries are uncorrelated at initialization which motivated the original choice of $\alpha_{\mathcal{A}} = \frac{1}{2}$ [5, 18]. Yang et al. [10] assume the entries of the key and query vectors move by $\Theta(1)$, implying $\alpha_{\mathcal{A}} = 1$ is necessary since the

---

[6]Training time, sample size, sequence length/spatial dimension are all treated as fixed $\Theta(1)$ quantities.

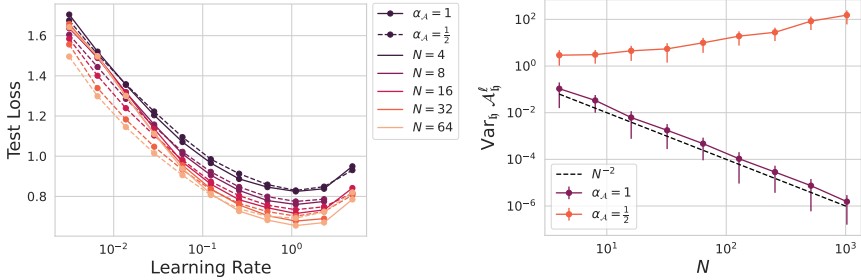

(a) Hyperparameter Transfer for Various $\alpha_{\mathcal{A}}$    (b) Attention variance across heads

Figure 2: Increasing dimension-per-head $N$ with heads fixed for $\alpha_{\mathcal{A}} = \{1, \frac{1}{2}\}$. (a) Both $\alpha_{\mathcal{A}} = 1$ and $\alpha_{\mathcal{A}} = \frac{1}{2}$ exhibit similar hyperparameter transfer for vision transformers trained on CIFAR-5M over finite $N$ at $\mathcal{H} = 16$. (b) The variance of attention variables across the different heads of a vision transformer after training for 2500 steps on CIFAR-5M. For $\alpha_{\mathcal{A}} = 1$ the variance of attention variables decays at rate $\mathcal{O}(N^{-2})$ and for $\alpha_{\mathcal{A}} = \frac{1}{2}$ the variance does not decay with $N$.

update to $\boldsymbol{k}$ is correlated to $\boldsymbol{q}$ and vice versa. However, it is possible to obtain $\Theta(1)$ updates to the attention variable for alternative values of $\alpha_{\mathcal{A}}$ if we choose the change to key (also query) entries after gradient descent to be $\delta k_i \sim \Theta(N^{-1+\alpha_{\mathcal{A}}})$. We show that this scaling can approximately preserve optimal hyperparameters across $N$ in Figure 2 (a) and give similar dynamics under SGD Appendix C. However, as we show in Appendix E.1.2, any well defined $N \to \infty$ limit of SGD requires $\alpha_{\mathcal{A}} = 1$. The reason is not that keys and queries become correlated, but rather that the scale of the backward pass must be controlled to ensure the dynamics remain stable (non-divergent) under SGD training. After performing two or more gradient descent steps, we demonstrate that the backpropagation signals will diverge as $N \to \infty$ unless initial key and query weight matrices are downscaled to have variance of order $\Theta_N(1)$. In Appendix E, we provide a DMFT analysis of the $N \to \infty$ limit of the transformer training dynamics. We summarize the result of that analysis informally below.

**Result 1 (Infinite Dimension-Per-Head $N$)** *(Informal) A stable feature learning $N \to \infty$ limit of transformer SGD training requires taking $\alpha_{\mathcal{A}} = 1$ ($\mu P$ scaling), even if key/query updates are allowed to be rescaled to account for their correlation. The limiting dynamics of training are governed by the residual stream kernel $H^{\ell}_{\mathfrak{s}\mathfrak{s}'}(\boldsymbol{x}, \boldsymbol{x}', t, t') = \frac{1}{N\mathcal{H}} \boldsymbol{h}^{\ell}_{\mathfrak{s}}(\boldsymbol{x}, t) \cdot \boldsymbol{h}^{\ell}_{\mathfrak{s}'}(\boldsymbol{x}', t')$ and a collection of inner product kernels in each head $\mathfrak{h}$ that concentrate as $N \to \infty$*

$$V^{\ell}_{\mathfrak{h}\mathfrak{s}\mathfrak{s}'}(\boldsymbol{x}, \boldsymbol{x}', t, t') = \frac{1}{N} \boldsymbol{v}^{\ell}_{\mathfrak{h}\mathfrak{s}}(\boldsymbol{x}, t) \cdot \boldsymbol{v}^{\ell}_{\mathfrak{h}\mathfrak{s}'}(\boldsymbol{x}', t') \ , \ Q^{\ell}_{\mathfrak{h}}(\boldsymbol{x}, \boldsymbol{x}', t, t') = \frac{1}{N} \boldsymbol{q}^{\ell}_{\mathfrak{h}\mathfrak{s}}(\boldsymbol{x}, t) \cdot \boldsymbol{q}^{\ell}_{\mathfrak{h}\mathfrak{s}'}(\boldsymbol{x}', t') \quad (6)$$

$$K^{\ell}_{\mathfrak{h}\mathfrak{s}\mathfrak{s}'}(\boldsymbol{x}, \boldsymbol{x}', t, t') = \frac{1}{N} \boldsymbol{k}^{\ell}_{\mathfrak{h}\mathfrak{s}}(\boldsymbol{x}, t) \cdot \boldsymbol{k}^{\ell}_{\mathfrak{h}\mathfrak{s}'}(\boldsymbol{x}', t') \ , \ \mathcal{A}^{\ell}_{\mathfrak{h}\mathfrak{s}\mathfrak{s}'}(\boldsymbol{x}, t) = \frac{1}{N} \boldsymbol{k}^{\ell}_{\mathfrak{h}\mathfrak{s}}(\boldsymbol{x}, t) \cdot \boldsymbol{q}^{\ell}_{\mathfrak{h}\mathfrak{s}'}(\boldsymbol{x}, t), \quad (7)$$

*alongside residual-stream gradient kernels and response functions in the sense of [15, 11]. The NN output logits $f(\boldsymbol{x}, t)$ evolve deterministically according to the above kernels as well as kernels for the gradient vectors $\boldsymbol{g}^{\ell} \equiv \gamma_0 N \mathcal{H} \frac{\partial f}{\partial \boldsymbol{h}^{\ell}}$ which appear in the backward pass. These variables become identical across heads such that for any $\mathfrak{h}, \mathfrak{h}' \in [\mathcal{H}]$, $\mathcal{A}^{\ell}_{\mathfrak{h}\mathfrak{s}\mathfrak{s}'}(\boldsymbol{x}, t) = \mathcal{A}^{\ell}_{\mathfrak{h}'\mathfrak{s}\mathfrak{s}'}(\boldsymbol{x}, t)$. All preactivations on the residual stream and key/query/value variables within a MHSA block are statistically independent across neurons and can be described by a single scalar stochastic process*

$$h^{\ell+1}_{\mathfrak{s}}(\boldsymbol{x}, t) = h^{\ell}_{\mathfrak{s}}(\boldsymbol{x}, t) + \beta_0 L^{-\alpha_L} \tilde{u}^{\ell}_{\mathfrak{s}}(\boldsymbol{x}, t) + \beta_0 L^{-\alpha_L} u^{\ell+1}_{\mathfrak{s}}(\boldsymbol{x}, t)$$

$$+ \eta_0 \gamma_0 \beta_0^2 L^{-1} \sum_{t' < t} \sum_{\mathfrak{s}' \in [\mathcal{S}]} \int d\boldsymbol{x}' \left[ \tilde{C}^{\ell}_{\mathfrak{s}\mathfrak{s}'}(\boldsymbol{x}, \boldsymbol{x}', t, t') \tilde{g}^{\ell}_{\mathfrak{s}'}(\boldsymbol{x}', t') + C^{\ell}_{\mathfrak{s}\mathfrak{s}'}(\boldsymbol{x}, \boldsymbol{x}', t, t') g^{\ell}_{\mathfrak{s}'}(\boldsymbol{x}', t') \right]$$

$$k^{\ell}_{\mathfrak{h}\mathfrak{s}}(\boldsymbol{x}, t) = u^{\ell}_{K\mathfrak{h}\mathfrak{s}}(\boldsymbol{x}, t) + \sum_{t' \mathfrak{s}'} \int d\boldsymbol{x}' C^{k\ell}_{\mathfrak{s}\mathfrak{s}'}(\boldsymbol{x}, \boldsymbol{x}', t, t') q^{\ell}_{\mathfrak{h}\mathfrak{s}'}(\boldsymbol{x}', t') \quad (8)$$

*where $\tilde{u}^{\ell}, u^{\ell}, u^{\ell}_{K\mathfrak{h}}$ are Gaussian processes with covariances $\Phi^{\ell,1}, V^{\ell\sigma}, H^{\ell}$ respectively. Analogous equations hold for the queries and values. In the limit, the kernels $H^{\ell}_{\mathfrak{s}\mathfrak{s}'}(\boldsymbol{x}, \boldsymbol{x}', t, t') = \langle h^{\ell}_{\mathfrak{s}}(\boldsymbol{x}, t) h^{\ell}_{\mathfrak{s}'}(\boldsymbol{x}', t') \rangle$, $\mathcal{A}^{\ell}_{\mathfrak{h}\mathfrak{s}\mathfrak{s}'}(\boldsymbol{x}, t) = \langle k^{\ell}_{\mathfrak{h}\mathfrak{s}}(\boldsymbol{x}, t) q^{\ell}_{\mathfrak{h}\mathfrak{s}'}(\boldsymbol{x}, t) \rangle$, etc. are computed as averages $\langle \cdot \rangle$ over*

*these random variables. The deterministic kernels $C^\ell, \tilde{C}^\ell$ can also be expressed in terms of single site averages of residual variables and head averages of MHSA variables. The kernels $C^\ell, \tilde{C}^\ell, C^{\prime k^\ell}$ depend on the precise mini-batches of data $\mathfrak{B}_t$ presented at each step $t$ which we assume are known.*

We derive this result using a Martin-Siggia-Rose path integral formalism [40] for DMFT in Appendix E. Full DMFT equations can be found in Appendix E.2. Following prior works on DMFT for infinite width feature learning, the large-$N$ limit can be straightforwardly obtained from a saddle point of the DMFT action [15, 11, 41, 17].

**Collapse of Attention Heads**    As $N \to \infty$, multi-head self-attention will effectively compute the same outputs as a single-head self-attention block. We theoretically derive this effect in Appendix E.2.1 and demonstrate it empirically in Figure 2 (b). This experiment shows that in $\alpha_\mathcal{A} = 1$ ($\mu$P) transformers trained for 2500 steps on CIFAR-5M [42], the variance of attention matrices across heads decreases with $N$. However, we note that if the scaling exponent is chosen instead as $\alpha_\mathcal{A} = \frac{1}{2}$ there is non-decreasing diversity of attention variables across heads. This result is consistent with recent empirical findings that attention variables in $\mu$P transformers converge to the same limiting quantities at large $N$ with $\mathcal{H}$ fixed for different initializations and also across model sizes [16]. This aspect of transformers in the large-$N$ limit is potentially undesirable as some tasks could require learning multiple attention mechanisms. Furthermore, this suggests scaling the model in this limit could increase computational cost without improving performance. To circumvent this, we explore if there exist well defined limits at finite $N$ with a diversity of attention variables across heads.

### 3.3   Scaling Number of Heads

In this section, we take $\mathcal{H} \to \infty$ with the inner dimension $N$ fixed. Rather than having all kernels concentrate, the kernel of each head of the MHSA block follows a statistically independent stochastic process. This picture was shown to hold at initialization by Hron et al. [18]. Here, using a DMFT analysis, we show that it continues to hold throughout training in the feature learning regime.

**Result 2 (Infinite Head Limit)** *(Informal) The $\mathcal{H} \to \infty$ limit of SGD training dynamics in a randomly initialized transformer at any key/query dimension $N$, scaling exponents $\alpha_\mathcal{A}, \alpha_L$, and any depth $L$ is governed by head-averaged kernels for pairs of input data $\boldsymbol{x}, \boldsymbol{x}'$ at training times $t, t'$ and spatial/token positions $\mathfrak{s}, \mathfrak{s}'$ such as*

$$V^{\ell,\sigma}_{\mathfrak{s}\mathfrak{s}'}(\boldsymbol{x}, \boldsymbol{x}', t, t') = \frac{1}{N\mathcal{H}} \sum_{\mathfrak{h}=1}^{\mathcal{H}} \boldsymbol{v}^{\ell\sigma}_{\mathfrak{h}\mathfrak{s}}(\boldsymbol{x}, t) \cdot \boldsymbol{v}^{\ell\sigma}_{\mathfrak{h}\mathfrak{s}'}(\boldsymbol{x}', t') \qquad (9)$$

*which converge to deterministic values as $\mathcal{H} \to \infty$. The attention variables $\{\boldsymbol{k}^\ell_\mathfrak{h}(\boldsymbol{x}, t), \boldsymbol{q}^\ell_\mathfrak{h}(\boldsymbol{x}, t), \boldsymbol{v}^\ell_\mathfrak{h}(\boldsymbol{x}, t), \mathcal{A}^\ell_\mathfrak{h}(\boldsymbol{x}, t)\}$ within each head become statistically independent across heads and decouple in their dynamics (but not across dimensions within a head). Each residual stream neuron becomes independent and obeys a single site stochastic process analogous to Result 1, but with different kernels.*

We derive this and the full DMFT in Appendix E.3, showing that the joint distribution of head-averaged dynamical quantities satisfies a large deviation principle and the limit can be derived as a saddle point of a DMFT action.

To gain intuition for this result, we first examine variables $H^\ell$ and $\mathcal{A}^\ell_\mathfrak{h}$ at initialization. In Figure 3, we plot the convergence of a $N = 4, L = 8$ vision transformer's residual stream kernel $H^\ell$ to its $\mathcal{H} \to \infty$ limit at rate $\mathcal{O}(\mathcal{H}^{-1})$ in square error, consistent with perturbative analysis near the limit [17]. Next, we plot the distribution (over heads) of $\mathcal{A}_\mathfrak{h}$ at a fixed pair of spatial/token positions for a fixed sample. This is a non-Gaussian random variable for finite $N$, but as $N \to \infty$ the distribution of $\mathcal{A}$ will approach a Gaussian with variance $\Theta(N^{1-2\alpha_\mathcal{A}})$.

We then investigate training dynamics as we approach the $\mathcal{H} \to \infty$ limit. In Figure 4 (a) we show the test loss on CIFAR-5M as a function of the number of training iterations. The performance tends improve as $\mathcal{H}$ increases and the model approaches its limit. In Figure 4 (b) we show that all of the models are converging in function space by measuring the squared error between finite $\mathcal{H}$ head models and a proxy for the infinite $\mathcal{H}$ model. Since the $\mathcal{H} \to \infty$ limit is essentially uncomputable, we approximate it as the ensemble averaged predictor of the widest possible models, a technique used in prior works [16, 11]. We again see that at early time, the logits of $\mathcal{H}$ head models converge to the limit proxy at a rate $\mathcal{O}(\mathcal{H}^{-1})$, but after continued training the convergence rate weakens. This effect

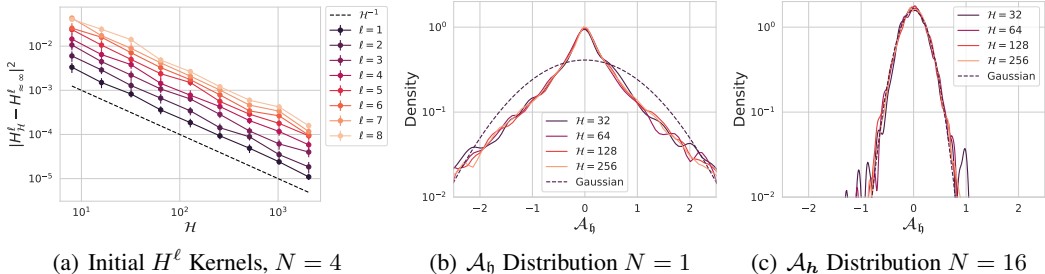

(a) Initial $H^\ell$ Kernels, $N = 4$      (b) $\mathcal{A}_{\mathfrak{h}}$ Distribution $N = 1$      (c) $\mathcal{A}_{\boldsymbol{h}}$ Distribution $N = 16$

Figure 3: The initial kernels converge as $\mathcal{H} \to \infty$ and are determined by (possibly non-Gaussian) distributions of $\mathcal{A}_{\mathfrak{h}}^\ell$ over heads in each layer. (a) Convergence of $H_{\mathfrak{s}\mathfrak{s}'}^\ell(\boldsymbol{x}, \boldsymbol{x}') = \frac{1}{\mathcal{H}N}\boldsymbol{h}_{\mathfrak{s}}^\ell(\boldsymbol{x}) \cdot \boldsymbol{h}_{\mathfrak{s}'}^\ell(\boldsymbol{x}')$ in a $L = 8, N = 4$ vision transformer at initialization at rate $\mathcal{O}(\mathcal{H}^{-1})$. (b) The density of $\mathcal{A}_{\mathfrak{h}}^\ell$ entries over heads at fixed spatial location converges as $\mathcal{H} \to \infty$ but is non-Gaussian for small $N$. (c) As $N \to \infty$ the initial density of $\mathcal{A}$ approaches a Gaussian with variance of order $\mathcal{O}(N^{1-2\alpha_{\mathcal{A}}})$.

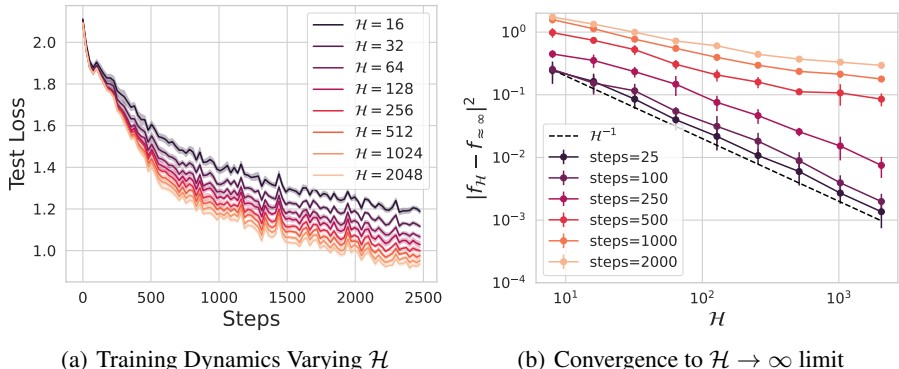

(a) Training Dynamics Varying $\mathcal{H}$      (b) Convergence to $\mathcal{H} \to \infty$ limit

Figure 4: Approaching the large head limit $\mathcal{H} \to \infty$ in early portion of SGD dynamics for a vision transformer trained on CIFAR-5M with $(L, N) = (2, 4)$ and $(\gamma_0, \beta, \alpha_{\mathcal{A}}) = (0.05, 4, \frac{1}{2})$ and losses averaged over 10 random inits (colored error bars are standard deviations). (a) As $\mathcal{H}$ increases the loss and the variability over random initial seed decreases. (b) The mean square difference between output logits for $\mathcal{H}$ head models and a proxy for the infinite head model on a held out batch of test examples. Following prior works, our proxy for the limit is the ensemble averaged outputs of the widest models [16, 11].

has been observed in $\mu$P networks in many settings [16] and a theoretical model of this was provided in recent work which argues it arises from low-rank effects in the finite $\mathcal{H}$ kernels [39].

### 3.4 Infinite Depth Limits

We next describe the infinite depth limits which depend on the choice of $\alpha_L$. Below we informally describe the main finding which again uses a DMFT formalism and is based on analyses in recent works on infinite depth networks from Bordelon et al. [11] and Yang et al. [12].

**Result 3 (Infinite Depth Limit)** *(Informal) The training dynamics for $\mathcal{H}, L \to \infty$ with $L^{-\alpha_L}$ branch scaling with $\alpha_L \in [\frac{1}{2}, 1]$ is described by a differential equation for residual variables $h_{\mathfrak{s}}(\tau, t)$ in layer time $\tau = \lim_{L \to \infty} \frac{\ell}{L}$ for the residual stream*

$$h_{\mathfrak{s}}(\tau, \boldsymbol{x}, t) = \beta_0\, \delta_{\alpha_L, \frac{1}{2}} \int_0^\tau du_{\mathfrak{s}}(\tau', \boldsymbol{x}, t)$$

$$+ \eta_0 \gamma_0 \beta_0^2 \sum_{t' < t} \int d\boldsymbol{x}' \int_0^\tau d\tau' C_{\mathfrak{s}\mathfrak{s}'}(\tau', \boldsymbol{x}, \boldsymbol{x}', t, t') g_{\mathfrak{s}'}(\tau', \boldsymbol{x}', t') \quad (10)$$

*where the Brownian motion term $du_{\mathfrak{s}}(\tau, \boldsymbol{x}, t)$ survives in the limit only if $\alpha_L = \frac{1}{2}$ and has covariance*

$$\langle du_{\mathfrak{s}}(\tau, \boldsymbol{x}, t) du_{\mathfrak{s}'}(\tau', \boldsymbol{x}', t') \rangle = \delta(\tau - \tau') d\tau d\tau' \left[ \Phi_{\mathfrak{s}\mathfrak{s}'}(\tau, \boldsymbol{x}, \boldsymbol{x}', t, t') + V_{\mathfrak{s}\mathfrak{s}'}^\sigma(\tau, \boldsymbol{x}, \boldsymbol{x}', t, t') \right] \quad (11)$$

*and the deterministic kernel $C_{\mathfrak{s}\mathfrak{s}'}(\tau, \boldsymbol{x}, \boldsymbol{x}', t, t')$ can be expressed in terms of head-averaged kernels and response functions. The weights inside each hidden MHSA layer or each MLP layer are frozen in the $L \to \infty$ limit unless $\alpha_L = 1$. All response functions are suppressed at $L \to \infty$ unless $\alpha_L = \frac{1}{2}$.*

Below we provide a couple of short comments about this result. The proof and full DMFT is provided in Appendix E.4.

1. At initialization $t = 0$, the only term which contributes to the residual stream layer dynamics is the integrated Brownian motion $\int_0^\tau du(\tau')$ which survives at infinite depth for $\alpha_L = \frac{1}{2}$. For $\alpha_L = 1$ this term disappears in the limit. The structure of $C(\tau)$ is also modified by additional response functions at $\alpha_L = \frac{1}{2}$ [11] which we show disappear for $\alpha_L = 1$.

2. The weights within residual blocks (including the MHSA block) can be treated as completely frozen for $\alpha_L < 1$ in the $L \to \infty$ limit, which leads to the simplified statistical description of the preactivations in those layers. However, the residual stream variables $h(\tau)$ do still obtain $\Theta_L(1)$ updates. At $\alpha_L = 1$ the weights in the MHSA blocks evolve by $\Theta_L(1)$, causing additional feature evolution in the model.

3. A consequence of our large $N$ and large $L$ result is that the $N, L \to \infty$ limit with $\alpha_L = \frac{1}{2}$ (the parameterization studied by [11, 12]) would lead to $\mathcal{A}^\ell_{\mathfrak{h}\mathfrak{s}\mathfrak{s}'}(\boldsymbol{x}, t) = 0$ for all time $t$. Thus the MHSA blocks would only involve average pooling operations over the spatial indices, despite the residual stream kernels $H^\ell$ updating from feature learning.

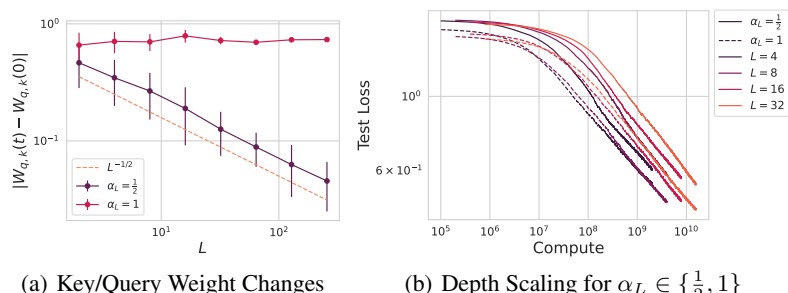

(a) Key/Query Weight Changes      (b) Depth Scaling for $\alpha_L \in \{\frac{1}{2}, 1\}$

Figure 5: Depth scaling in a vision transformer on CIFAR-5M with $\alpha_L \in \{\frac{1}{2}, 1\}$. (a) The key and query weights move by $1/\sqrt{L}$. (b) The compute scaling laws with models at fixed width $N, \mathcal{H}$ and varying depth $L$. At large $L$, the $\alpha_L = 1$ (dashed) models perform better at fixed compute.

First, we note in Figure 5 that the weights within each attention block freeze as $L \to \infty$ with $\alpha_L = \frac{1}{2}$ case but move at a constant scale for $\alpha_L = 1$. As a consequence, the loss at large $L$ can be lower in the $\alpha = 1$ parameterization.

We can see some numerical evidence for the first of these effects in Figure 6 (a)-(b) where initially training at large $L$ is slower than the base model and the initial kernel appears quite different for $L = 4$ and $L = 64$. The initial kernel will decrease in scale as $L \to \infty$ for $\alpha_L = 1$ since the preactivation vectors lose variance as we discuss in Appendix E.4, resulting in slower initial training. However, we note that the final learned feature kernels are quite similar after enough training.

In summary, our results indicate that the $\alpha_L = 1$ parameterization is the one that allows attention layers to actually be learned in the limit $L \to \infty$, but that this parameterization leads to a less structured kernel at initialization.

## 4 Experiments in Realistic Settings

In practice, large scale neural networks do not generally operate close to their limit. Given the costs of training large networks, one would ideally operate in a regime where there is a guarantee of consistent improvements with respect to model scale. In pursuit of this goal, we apply our theoretical findings of this paper to training language models on a larger natural language dataset, a Transformer with causal attention blocks trained on the C4 dataset [43] with Adam optimizer. As mentioned in 2.2, while our exact theoretical description of these infinite limits focus on SGD, we can implement an appropriate scaling for Adam which preserves the scale of internal feature updates. This allows us

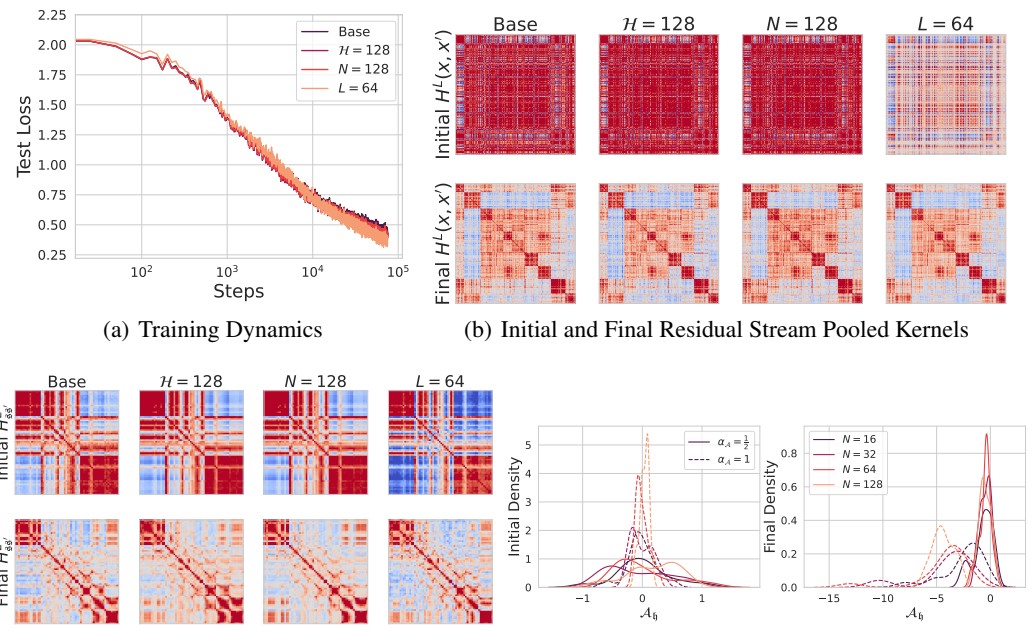

(a) Training Dynamics

(b) Initial and Final Residual Stream Pooled Kernels

(c) Spatial Kernels for Single Sample

(d) Attention Distributions Before and After Training

Figure 6: Initial and final representations are converging as model scale increases after one pass of training on the full CIFAR-5M with SGD+momentum. The base model is a $(N, \mathcal{H}, L) = (16, 16, 4)$ and $(\alpha_{\mathcal{A}}, \alpha_L, \beta_0, \gamma_0) = (1, 1, 4, 0.1)$. (a) The test loss dynamics for one pass through CIFAR-5M. The dynamics are very similar across different head-counts $\mathcal{H}$ but the early dynamics are changed for large depth $L$, consistent with our theory. (b) The initial and final feature kernels after spatial pooling at the last layer of the residual stream. The initial kernel at large $L$ is quite different for $\alpha_{\mathcal{A}} = 1$ due to suppression of Brownian motion on the forward pass, which we explain in Section 3.4. (c) The residual stream kernel across pairs of spatial positions for a single randomly chosen input sample. (d) The distribution of attention entries across heads at a fixed pair of spatial locations and data point. The initial variance of $\mathcal{A}$ decreases for $\alpha_{\mathcal{A}} = 1$ but the update is roughly consistent across $N$. For $\alpha_{\mathcal{A}} = \frac{1}{2}$ both initial and final distributions for $\mathcal{A}_{\mathfrak{h}}$ are consistent across $N$.

to investigate realistic training dynamics of our LLM as we take the $N, L, \mathcal{H} \to \infty$ limits. Training details are provided in Appendix F

In Figure 7 (a), we sweep over each of the model dimensions independently for each parameterization of $\alpha_{\mathcal{A}} \in \{1, \frac{1}{2}\}$ on the left and right respectively. For fixed $N$ and $L$, scaling $\mathcal{H}$ provides a similar increase in performance in both parameterization and appear to start converging to a final loss around 5, with slight benefit to $\alpha_{\mathcal{A}} = \frac{1}{2}$. For fixed $\mathcal{H}$ and $L$, scaling $N$ provides a similar increase in performance to scaling heads in when $\alpha_{\mathcal{A}} = 1$, but a substantial increase when $\alpha_{\mathcal{A}} = \frac{1}{2}$. This is in line with our predictions in Section 3.2 about the benefits of diversity across attention heads. Next, for fixed $N$ and $\mathcal{H}$, scaling $L$ provides little to no benefit in either parameterization as predicted in Section 3.4. Finally, we inspect the sample and spatial residual stream kernels of these models before and after training and find that the kernels are identical for both $\alpha_{\mathcal{A}}$, except for a slight difference for large $N$. Furthermore, they are extremely similar for large $N$ and large $\mathcal{H}$.

Taken together, these results suggest that scaling different model dimensions do indeed have different effects on training dynamics and final performance. This provides groundwork for future large-scale experiments systematically investigating their trade-offs, thereby identifying compute-optimal scaling of realistic architectures in parameterizations with well-defined limits.

## 5 Discussion

This paper provided analysis of the infinite head, depth and key/query dimension limits of transformer training in the feature learning regime. We showed that feature learning in $\mu$P multi-head transformers in the limit of $N \to \infty$ collapses to single-head self-attention. At finite $N$ and infinite heads $\mathcal{H} \to \infty$

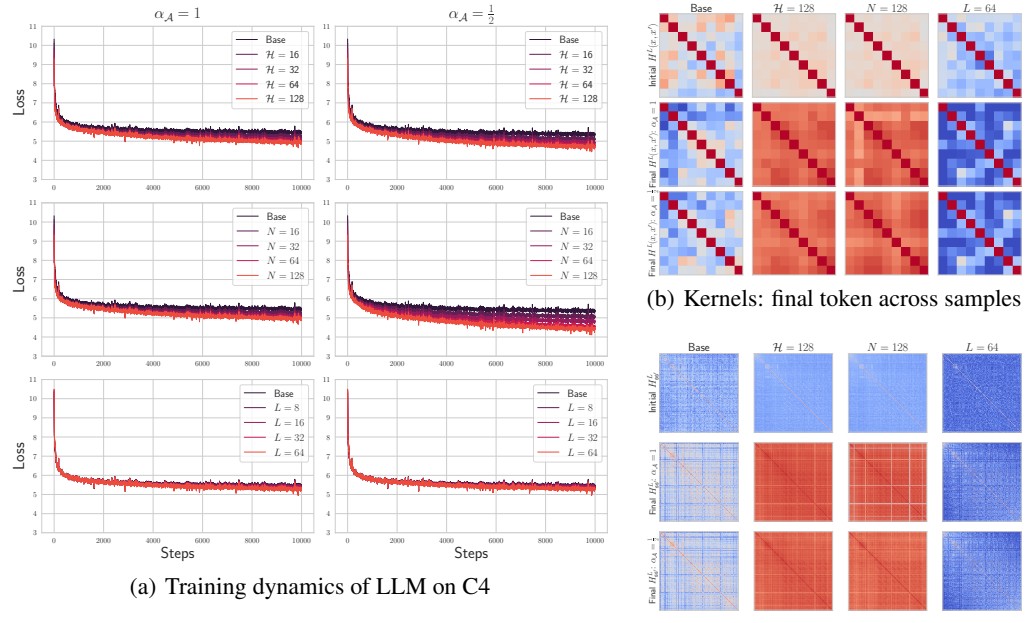

(a) Training dynamics of LLM on C4

(b) Kernels: final token across samples

(c) Kernels: tokens within single sample

Figure 7: Training dynamics and initial/final representations of decoder only language models trained on C4 converge with increasing model scale. The base model has $(N, \mathcal{H}, L) = (8, 8, 4)$ and $(\alpha_L, \beta_0, \gamma_0) = (1, 4, 0.25)$ and $\alpha_{\mathcal{A}} \in \{1, \frac{1}{2}\}$. (a) Train loss dynamics after 10000 steps on C4 using Adam optimizer. The dynamics improve consistently when scaling $\mathcal{H}$ for both values of $\alpha_{\mathcal{A}}$, with slight benefit to $\alpha_{\mathcal{A}} = \frac{1}{2}$. Scaling $N$ reveals a significant advantage to setting $\alpha_{\mathcal{A}} = \frac{1}{2}$. Scaling $L$ provides little improvement for either parameterization of $\alpha_{\mathcal{A}}$. (b) Initial and final residual stream kernels for the final token across samples for Base, $\mathcal{H} = 128$, $N = 128$, and $L = 64$ models. The first row is at initialization. The second and third rows are after training with $\alpha_{\mathcal{A}} \in \{1, \frac{1}{2}\}$ respectively. (c) Initial and final feature kernels across pairs of tokens for a single randomly chosen input sample. Note both types of kernels are identical across $\alpha_{\mathcal{A}}$ except for a slight difference at large $N$.

we showed that there is an alternative limit which maintains a *distribution over attention heads*. We discussed two different large depth limits of transformer training that reduce to differential equations in the residual layer time $\tau$. The depth scaling that maintains feature learning within all MHSA blocks ($\alpha_L = 1$) causes the initial kernel to lose structure from the initialization as $L \to \infty$, but allows learning of the self-attention variables, whereas the depth scaling that preserves structure from initialization ($\alpha_L = \frac{1}{2}$) leads to static layers.

**Limitations and Future Directions**   Currently exact theoretical analysis of the limit is focused on SGD (and can be easily extended to SGD+momentum [15]) while Adam is currently only reasoned with rough scaling arguments rather than an exact theoretical description of the limit. Since Adam is most commonly used to train transformers, a theory of the limiting dynamics of Adam in Transformers would be an important future extension. In addition, while we provide an exact asymptotic description of network training, the limiting equations are compute intensive for realistic settings which is why we focus our empirical investigations on training large width networks in the appropriate parameterizations. Lastly our techniques assume that the number of training steps is fixed as the scaling parameters of interest $(N, \mathcal{H}, L)$ are taken to infinity. However, it would be important to understand learning dynamics in the regime where model size and training times are chosen to balance a compute optimal tradeoff (or perhaps even training longer than compute optimal) [8, 39, 44]. In this regime, harmful finite model-size effects become significant and comparable to the finite training horizon [10, 16, 17, 39]. Thus stress testing the ideas in this work at larger scales and longer training runs would be an important future direction of research into scaling transformer models.

## Acknowledgements and Disclosure of Funding

BB would like to thank Alex Atanasov, Jacob Zavatone-Veth, Lorenzo Noci, Mufan Bill Li, Boris Hanin, Alex Damian, Eshaan Nichani for inspiring conversations. We would also like to thank Alex Atanasov and Jacob Zavatone-Veth for useful comments on an earlier version of this manuscript. BB is supported by a Google PhD fellowship. HC was supported by the GFSD Fellowship, Harvard GSAS Prize Fellowship, and Harvard James Mills Peirce Fellowship. CP was supported by NSF Award DMS2134157 and NSF CAREER Award IIS2239780. CP is further supported by a Sloan Research Fellowship. This work has been made possible in part by a gift from the Chan Zuckerberg Initiative Foundation to establish the Kempner Institute for the Study of Natural and Artificial Intelligence. The computations in this paper were run on the FASRC Cannon cluster supported by the FAS Division of Science Research Computing Group at Harvard University.

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

# Appendix

## A  Additional Figures

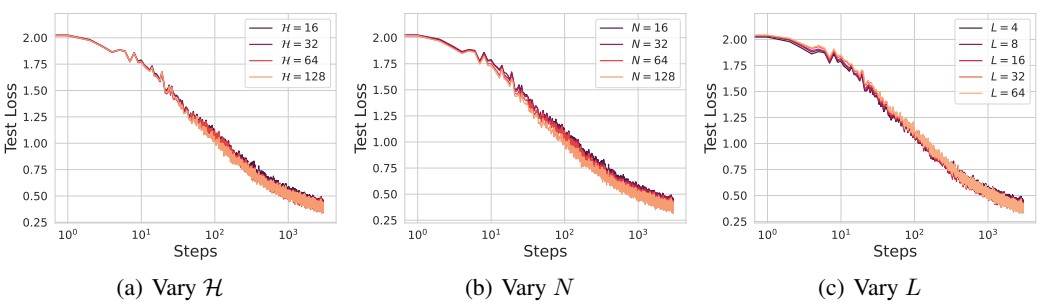

(a) Vary $\mathcal{H}$

(b) Vary $N$

(c) Vary $L$

Figure 8: One pass training on CIFAR-5M with vision transformers with the setting of Figure 6.

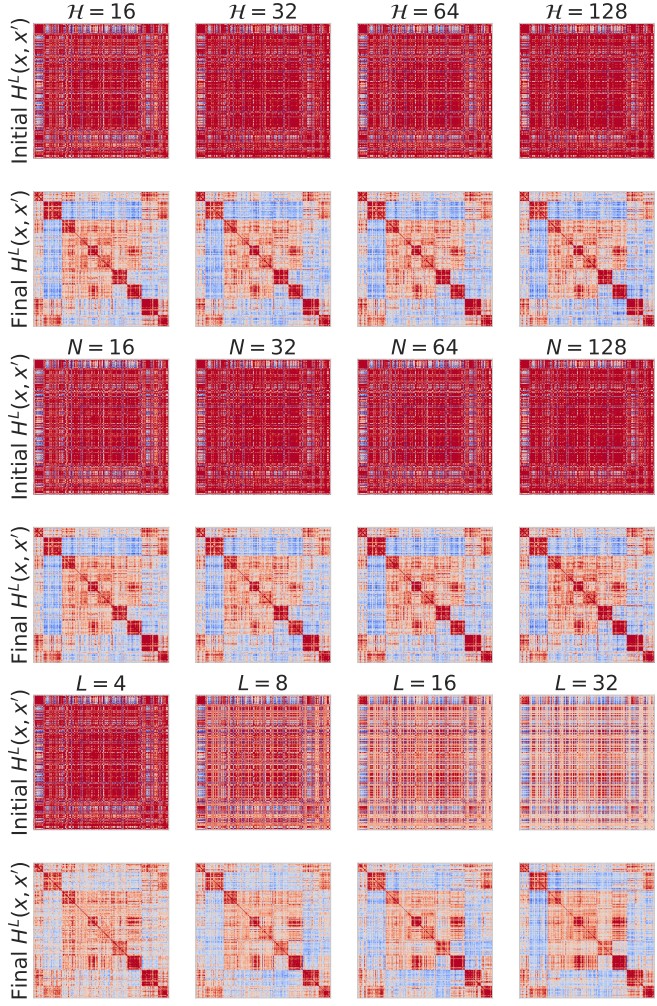

Figure 9: Examples of initial and learned kernels in final residual stream layer with various extrapolations of a base vision transformer model with $(\mathcal{H}, N, L) = (16, 16, 4)$ trained on CIFAR-5M.

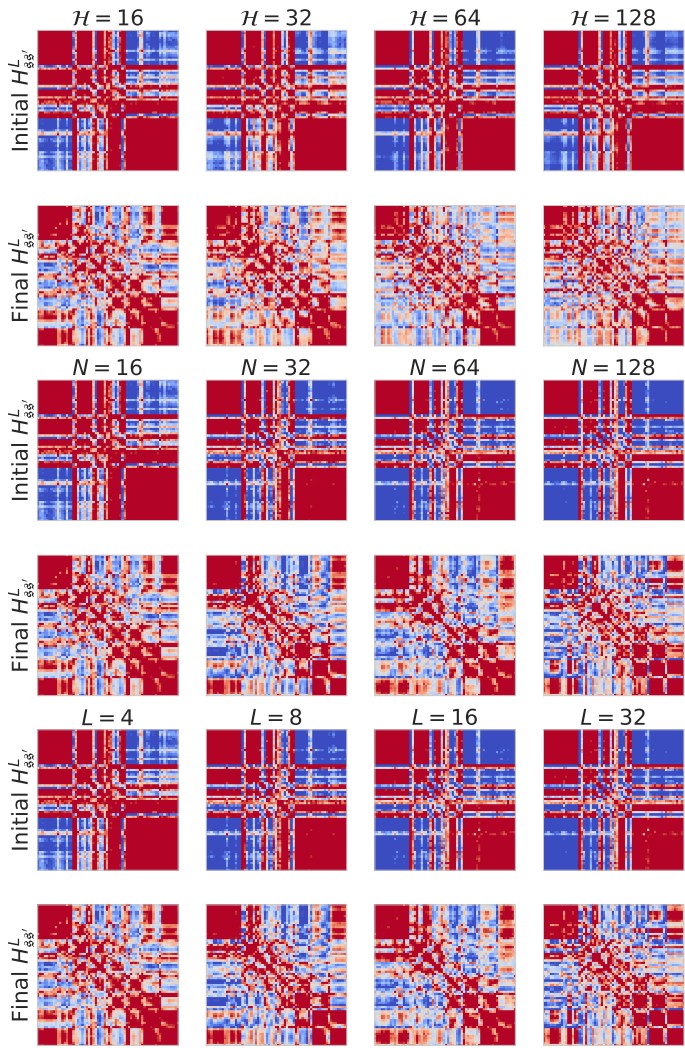

Figure 10: Spatial kernels for a single test point before and after training across $\mathcal{H}, N, L$ values.

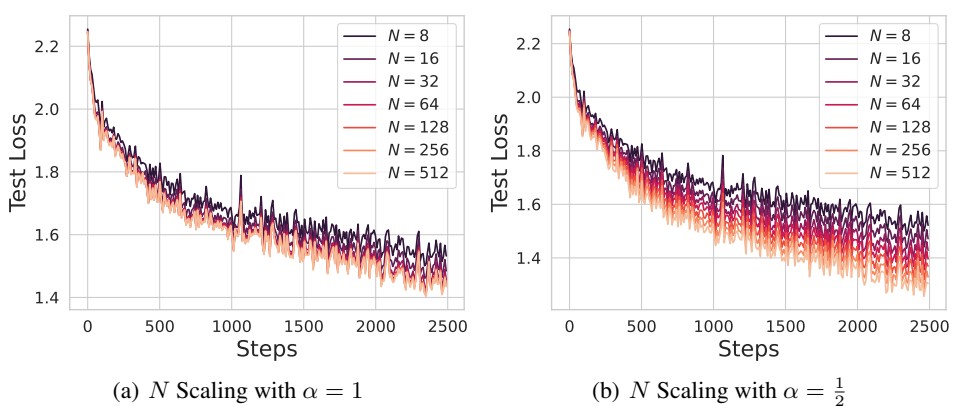

(a) $N$ Scaling with $\alpha = 1$

(b) $N$ Scaling with $\alpha = \frac{1}{2}$

Figure 11: Early training dynamics on CIFAR-5M in vision transformer with different dimension-per-head $N$ with heads fixed at $\mathcal{H} = 4$ for $\alpha_{\mathcal{A}} = \{1, \frac{1}{2}\}$.

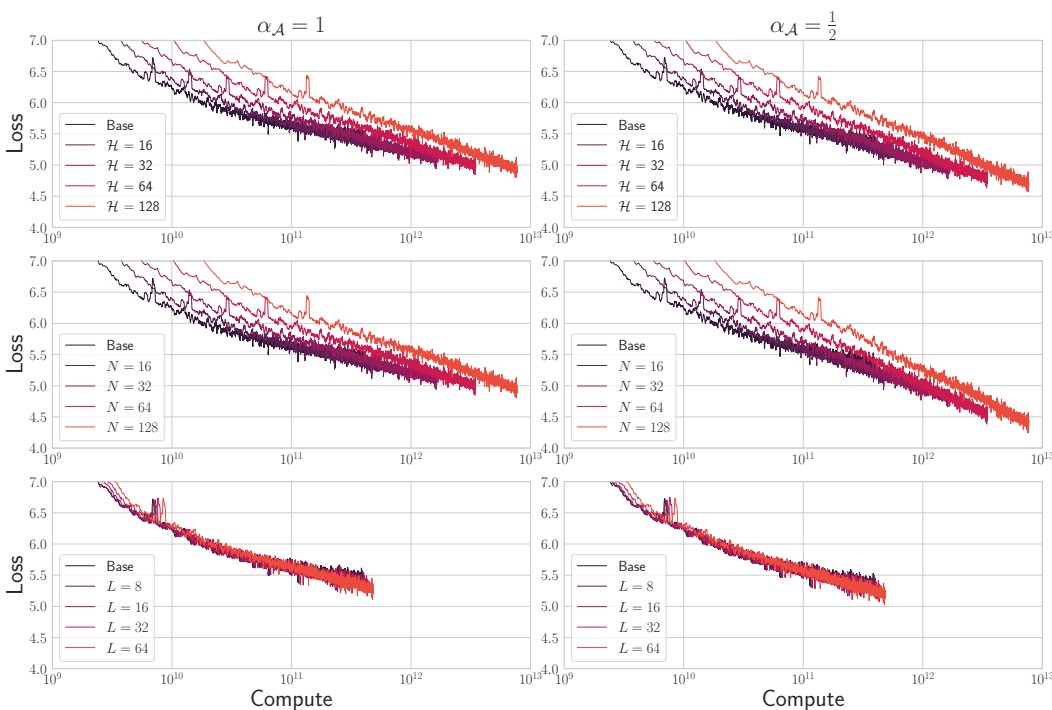

Figure 12: Performance of language models trained on C4 in main text Figure 7(a) as a function of compute, estimated as $\text{FLOPs} = 6 \times \text{Params}$. The base model has size $(N, \mathcal{H}, L) = (8, 8, 4)$ and we examine scaling up $N, \mathcal{H}, L$ with either $\alpha_{\mathcal{A}} = 1/2$ or $\alpha_{\mathcal{A}} = 1$. The $\alpha_{\mathcal{A}} = 1$ models perform better at fixed compute for either $N$ or $\mathcal{H}$ scaling. Increasing $L$ does not significantly increase compute in this regime since the embedding and decoding layers contribute most of the parameters.

# B   Implementations for Vision and Causal Language Modeling Transformers

We provide an example FLAX implementation of the vision transformer and causal language model.

We start by defining a fixed layernorm operation

```
from flax import linen as nn
import jax.numpy as jnp

class LN_Fixed(nn.Module):

    eps: jnp.float32 = 1.0e-6
    @nn.compact

    def __call__(self, x):

        features = x.shape[-1] # number of features
        mean = jnp.mean( x , axis = -1 ) # mean of x
        var = jnp.var( x , axis = -1 ) # var of x
        out = (x - mean[:,:,jnp.newaxis] ) / jnp.sqrt( var[:,:,jnp.
    newaxis] + self.eps )
        return out
```

The MHSA layer is implemented as the following where scale_exp represents $\alpha_{\mathcal{A}}$.

```
# MHSA attention layer
from einops import rearrange

class Attention(nn.Module):
    """Multi-head Self-Attention Layer"""
    scale_exp: jnp.float32
    dim: int
    heads: int

    def setup(self):

        self.c = 1.5 - self.scale_exp # exponent for the scale factor
        kif_qk = nn.initializers.normal(stddev = self.dim**(self.c -
    0.5) ) # possible scaling with N
        kif_v =  nn.initializers.normal(stddev = 1.0 ) # O_N(1)
    entries
        # computes key, query, value
        self.qk_layer = nn.Dense(features = 2 * self.heads * self.dim,
     kernel_init = kif_qk, use_bias = False)
        self.v_layer = nn.Dense(features = self.heads * self.dim,
    kernel_init = kif_v, use_bias = False)
        self.out_layer = nn.Dense(features = self.heads * self.dim,
    kernel_init = kif_v, use_bias = False)
        return

    def __call__(self,inputs):

        qk = self.qk_layer(inputs) / self.heads**(0.5) / self.dim**(
    self.c)
        qk = rearrange( qk, 'b l (h d) -> b h l d' , h = self.heads) #
     (batch, heads, loc, d )
        q,k = jnp.split(qk, 2, axis = -1) # gives q, k each of shape (
    batch, heads, loc, d )

        v = self.v_layer(inputs) / jnp.sqrt( inputs.shape[-1] )
        v = rearrange(v, 'b l (h d) -> b h l d', h = self.heads)
        A = self.dim**(-self.scale_exp) * jnp.einsum('ijkl,ijml->ijkm'
    , q, k) # batch x heads x loc x loc
        sigma_A = softmax( A, axis=-1 )
```

```
31          out = jnp.einsum('ijkl,ijlm->ijkm', sigma_A, v) # (batch, head
      , loc, d)
32          out = rearrange(out, 'b h l d -> b l (h d)')
33          out = self.out_layer(out) / jnp.sqrt( out.shape[-1] )
34          return out
```

The two layer MLP block is implemented as the following with $\phi = $ gelu nonlinearity.

```
1  class MLP_Block(nn.Module):
2      """Two Layer MLP Block"""
3      features: int
4
5      @nn.compact
6      def __call__(self,x):
7          N = self.features
8          kif = nn.initializers.normal(stddev = 1.0) # O_N(1) entries
9          h = nn.Dense(features = N, kernel_init = kif, use_bias = False
      )(x) / jnp.sqrt(N)
10          h = nn.gelu(h)
11          h = nn.Dense(features = N, kernel_init = kif, use_bias = False
      )(h) / jnp.sqrt(N)
12          return h
13
```

We also allow for a trainable positional encoding matrix.

```
1
2  class PositionalEncoding(nn.Module):
3      """Trainable Positional Encoding"""
4      d_model : int        # Hidden dimensionality of the input.
5      max_len : int  # Maximum length of a sequence to expect.
6      scale: jnp.float32 # scale parameter for initialization
7
8      def setup(self):
9          # Create matrix of [SeqLen, HiddenDim] representing the
      positional encoding for max_len inputs
10          self.pos_embedding = self.param('pos_embedding',
11                                          nn.initializers.normal(stddev
      = self.scale),
12                                          (1, 1+self.max_len, self.
      d_model))
13
14      def __call__(self, x, train=True):
15          B,T,_ = x.shape
16          x = x + self.pos_embedding[:,:T] / self.scale
17          return x
```

Each residual block is implemented as the following. Below we show the $\alpha_L = 1$ implementation.

```
1  # Residual Block
2  class ResidBlock(nn.Module):
3
4      dim: int
5      heads: int
6      features: int
7      L: int
8      scale_exp: jnp.float32 = 1.0
9      beta: jnp.float32 = 4.0
10
11      @nn.compact
12      def __call__(self,x):
13          h = LN_Fixed()(x)
14          h = Attention(dim = self.dim, scale_exp = self.scale_exp,
      heads = self.heads)( h )
15          x = x + self.beta / self.L * h
16          h = LN_Fixed()(x)
```

```
17            h = MLP_Block(features = self.features)(h)
18            x = x + self.beta / self.L * h
19            return x
```

Our vision transformer model consists of an embedding layer which is applied to each patch, a positional encoding layer, $L$ residual layers each containing a MHSA and MLP block, a spatial pooling operation, and a readout.

```
1
2  class VIT(nn.Module):
3
4      "simple VIT model with "
5      dim: int
6      heads: int
7      depth: int
8      patch_size: int
9      scale_exp: jnp.float32 = 1.0
10     adam_scale: int = 0.0
11     beta: jnp.float32 = 4.0
12
13     @nn.compact
14     def __call__(self, x):
15         d_model = self.heads * self.dim
16         L = self.depth
17         D = 3
18
19         # patchify images
20         x = rearrange(x, 'b (w p1) (h p2) c -> b (w h) (p1 p2 c)', p1
    = self.patch_size, p2 = self.patch_size) # (batch, loc,
    patch_ch_dim )
21
22         kif_first= nn.initializers.normal(stddev = d_model**(-0.5*self
    .adam_scale) * (L/self.beta)**(0.5 * (1.0-self.adam_scale)) ) #
    O_N(1) entries
23         kif = nn.initializers.normal( stddev = 1.0 ) # O_N(1) entries
24         kif_last = nn.initializers.normal(stddev = (L/self.beta)**(0.5
     * (1-self.adam_scale) ) )
25
26         # read-in weights
27         x = (L/self.beta)**(-0.5 * (1.0-self.adam_scale))*d_model
    **(0.5 * self.adam_scale) * nn.Dense(features = N, kernel_init =
    kif_first, use_bias = False)(x) / jnp.sqrt( D * self.patch_size**2
     )
28
29         # positional encoding
30         x = PositionalEncoding(d_model = d_model, max_len = (32//self.
    patch_size)**2, scale = d_model**(-0.5*self.adam_scale)*(L/self.
    beta)**(0.5 * (1.0-self.adam_scale)))(x)
31
32         # residual stream with pre-LN
33         for l in range(self.depth):
34             x = ResidBlock(dim = self.dim, heads = self.heads,
    scale_exp=self.scale_exp, features = d_model, beta=self.beta, L =
    L)(x)
35
36         # last norm layer
37         x = LN_Fixed()(x)
38         # pool over spatial dimension
39         x = x.mean(axis = 1) # (batch, d_model)
40         x = (L/self.beta)**(-0.5*(1-self.adam_scale)) * nn.Dense(
    features = 10, use_bias = False, kernel_init = kif_last)(x) /
    d_model**(1.0-0.5*self.adam_scale)   # for mean field scaling
41         return x
```

For the causal decoder only model, we need to modify the Attention layer and also prevent pooling over spatial indices before the readout.

```python
class Causal_Attention(nn.Module):

    scale_exp: jnp.float32
    dim: int
    heads: int
    qk_ln: bool = True

    def setup(self):

        self.c = 1.5 - self.scale_exp # exponent for the scale factor
        kif_qk = nn.initializers.normal(stddev = self.dim**(self.c -
    0.5) ) # possibly needs to be scaled with N
        kif_v =  nn.initializers.normal(stddev = 1.0 ) # O_N(1)
    entries
        # computes key, query, value
        self.qk_layer = nn.Dense(features = 2 * self.heads * self.dim,
     kernel_init = kif_qk, use_bias = False)
        self.v_layer = nn.Dense(features = self.heads * self.dim,
    kernel_init = kif_v, use_bias = False)
        self.out_layer = nn.Dense(features = self.heads * self.dim,
    kernel_init = kif_v, use_bias = False)
        return

    def __call__(self,inputs):

        qk = self.qk_layer(inputs) / self.heads**(0.5) / self.dim**(
    self.c)   # (batch, loc, 3*h*d)
        qk = rearrange( qk, 'b l (h d) -> b h l d' , h = self.heads) #
     (batch, heads, loc, d )
        q,k = jnp.split(qk, 2, axis = -1) # gives q, k each of shape (
    batch, heads, loc, d )

        v = self.v_layer(inputs) / jnp.sqrt( inputs.shape[-1] )
        v = rearrange(v, 'b l (h d) -> b h l d', h = self.heads)

        A = 1.0/ self.dim**(self.scale_exp) * jnp.einsum('ijkl,ijml->
    ijkm', q, k) # batch x heads x loc x loc
        exp_A =  jnp.einsum('ijkl,kl->ijkl', jnp.exp(A), jnp.tril(jnp.
    ones((v.shape[2], v.shape[2])))))
        phi_A = exp_A / exp_A.sum(axis = -1)[:,:,:,jnp.newaxis]

        out = jnp.einsum('ijkl,ijlm->ijkm', phi_A, v) # (batch, head,
    loc, d)
        out = rearrange(out, 'b h l d -> b l (h d)')
        out = self.out_layer(out) / jnp.sqrt( out.shape[-1] )
        return out

class LM_Transformer(nn.Module):
    """A simple Decoder only transformer"""

    dim: int
    heads: int
    depth: int
    scale_exp: jnp.float32
    adam_scale: int
    beta: jnp.float32
    VOCAB_SIZE: int

    @nn.compact
    def __call__(self, x, train = True):
```

```
52        d_model = self.heads * self.dim
53        L = self.depth
54        kif_first = nn.initializers.normal(stddev = d_model**(-0.5*
      self.adam_scale) * (L/self.beta)**(0.5 * (1-self.adam_scale) ) ) #
       O(1) entries
55        kif0 = nn.initializers.normal(stddev = 0.0 )
56        kif = nn.initializers.normal(stddev = 1.0) # O(1) entries
57        kif_last = nn.initializers.normal(stddev = (L/self.beta)**(0.5
       * (1-self.adam_scale)) * d_model**(-0.5*self.adam_scale) )
58
59        # embed the batch x sequence integers to
60        x = (L/self.beta)**( -0.5 * (1-self.adam_scale) )* d_model
      **(0.5 * self.adam_scale) * nn.Embed(self.VOCAB_SIZE, d_model,
      embedding_init = kif_first)(x) # batch x seq len x N
61
62        x = PositionalEncoding(d_model = d_model, scale = d_model
      **(-0.5*self.adam_scale) * (L/self.beta)**(0.5 *(1-self.adam_scale
      )) )(x)
63
64        for l in range(self.depth):
65            h = LN_Fixed()(x)
66            x = x + self.beta/L * Causal_Attention(dim = self.dim,
      scale_exp = self.scale_exp, heads = self.heads)(h)
67            h = LN_Fixed()(x)
68            x = x + self.beta/L * MLP_Block(features = d_model)(h)
69
70        x = LN_Fixed()(x)
71        x = (L/self.beta)**(-0.5 * (1 - self.adam_scale ) ) * nn.Dense
      (features = self.VOCAB_SIZE, use_bias = True, kernel_init = kif0)(
      x) / d_model**(1.0-0.5*self.adam_scale)   # for mean field scaling
72        return x
```

## C   Simple Heuristic Scaling Analysis

In this section, we heuristically work out the simple scaling analysis to justify the set of parameterizations and learning rates we consider. More detailed theoretical analysis for the limit of SGD training is provided in Appendix E where we exactly characterize the $N \to \infty, \mathcal{H} \to \infty$ and $L \to \infty$ limits. We consider taking heads $\mathcal{H}$, inner dimension $N$ and depth $L$ to infinity separately and attempt to control the scale of gradients and updates.

### C.1   Learning Rate Scalings

We show that the correct learning rate scaling for SGD is $\eta = \eta_0 N \mathcal{H} L^{2\alpha_L - 1}$. For Adam, the learning rate should be scaled as $\eta = \eta_0 N^{-1/2} \mathcal{H}^{-1/2} L^{-1+\alpha_L}$.

| Optimizer | Bulk Parameters LR | First Layer Rescale Factor |
|-----------|--------------------|-----------------------------|
| SGD | $\eta_0 N \mathcal{H} L^{2\alpha_L - 1}$ | $L^{-\frac{1}{2}-\alpha}$ |
| Adam | $\eta_0 N^{-1/2} \mathcal{H}^{-1/2} L^{-1+\alpha_L}$ | $L^{1-\alpha_L}$ |

Table 2: The learning rates which should be applied to obtain the correct scale of updates for SGD or Adam optimizers. In addition, the weight variance and multiplier for the first layer may need to be rescaled with depth depending on the parameterization and optimizer.

### C.2   Heuristic Analysis of Feature Changes Under SGD

In this section we consider performing a single update on a single example to all weight matrices.

$$\delta \boldsymbol{W}_{O\mathfrak{h}}^{\ell} \sim \frac{1}{L^{1-\alpha_L}\sqrt{N\mathcal{H}}} \boldsymbol{g}^{\ell+1} \boldsymbol{v}_{\mathfrak{h}}^{\ell\top} \tag{12}$$

where $g^{\ell+1} \in \mathbb{R}^{N\mathcal{H}}$ and $v_{\mathfrak{h}}^{\ell} \in \mathbb{R}^{N}$ have $\Theta(1)$ entries. Thus, computing a perturbation to the forward pass we find

$$\delta h^{\ell+1} = \delta h^{\ell} + \frac{1}{L^{\alpha_L}\sqrt{N\mathcal{H}}} \sum_{\mathfrak{h}=1}^{\mathcal{H}} \left( \frac{1}{L^{1-\alpha_L}\sqrt{N\mathcal{H}}} g^{\ell+1} v_{\mathfrak{h}}^{\ell\top} \right) v_{\mathfrak{h}}^{\ell}$$

$$= \delta h^{\ell} + \frac{1}{L} \left[ \frac{1}{N\mathcal{H}} \sum_{\mathfrak{h}} v_{\mathfrak{h}}^{\ell} \cdot v_{\mathfrak{h}}^{\ell} \right] g^{\ell+1} \tag{13}$$

The term in the brackets is $\Theta(1)$ and we see that the perturbation from each layer contributes $\Theta(L^{-1})$. As there are $L$ layers, this will give a total change to the final layer $h^L$ that is $\Theta(1)$.

For the Attention variables, we note that the

$$\delta W_{K\mathfrak{h}}^{\ell} \sim \frac{1}{L^{1-\alpha_L}\sqrt{N\mathcal{H}}} q_{\mathfrak{h}}^{\ell} h^{\ell\top} \;,\; \delta W_{Q\mathfrak{h}}^{\ell} \sim \frac{1}{L^{1-\alpha_L}\sqrt{N\mathcal{H}}} k_{\mathfrak{h}}^{\ell} h^{\ell\top} \tag{14}$$

where $q_{\mathfrak{h}}, k_{\mathfrak{h}}^{\ell} \in \mathbb{R}^{N}$ are the query and key for head $\mathfrak{h}$ and $h \in \mathbb{R}^{N\mathcal{H}}$ is the residual stream preactivation. We can thus compute the changes to the keys and queries due to changes in their associated weights

$$\delta k_{\mathfrak{h}}^{\ell} = \frac{1}{N^{\frac{3}{2}-\alpha_{\mathcal{A}}}\sqrt{\mathcal{H}}} \left( \frac{1}{L^{1-\alpha_L}\sqrt{N\mathcal{H}}} q_{\mathfrak{h}}^{\ell} h^{\ell\top} \right) h^{\ell} = \frac{1}{L^{1-\alpha_L} N^{1-\alpha_{\mathcal{A}}}} q_{\mathfrak{h}}^{\ell} H^{\ell}$$

$$\delta q_{\mathfrak{h}}^{\ell} = \frac{1}{N^{\frac{3}{2}-\alpha_{\mathcal{A}}}\sqrt{\mathcal{H}}} \left( \frac{1}{L^{1-\alpha_L}\sqrt{N\mathcal{H}}} k_{\mathfrak{h}}^{\ell} h^{\ell\top} \right) h^{\ell} = \frac{1}{L^{1-\alpha_L} N^{1-\alpha_{\mathcal{A}}}} q_{\mathfrak{h}}^{\ell} H^{\ell}. \tag{15}$$

where $H^{\ell} = \frac{1}{N\mathcal{H}} h^{\ell} \cdot h^{\ell} \sim \Theta(1)$. Combining these changes , we find the following update to the pre-Attention variables $\mathcal{A}_{\mathfrak{h}}^{\ell} = \frac{1}{N^{\alpha_{\mathcal{A}}}} k_{\mathfrak{h}}^{\ell} \cdot q_{\mathfrak{h}}^{\ell}$

$$\delta \mathcal{A}_{\mathfrak{h}}^{\ell} = \frac{1}{L^{1-\alpha_L} N} q_{\mathfrak{h}}^{\ell} \cdot q_{\mathfrak{h}}^{\ell} H^{\ell} + \frac{1}{L^{1-\alpha_L} N} k_{\mathfrak{h}}^{\ell} \cdot k_{\mathfrak{h}}^{\ell} H^{\ell} + \frac{1}{L^{2-2\alpha_L} N^{2-2\alpha_{\mathcal{A}}}} \mathcal{A}_{\mathfrak{h}}^{\ell} (H^{\ell})^{2}$$

$$= \Theta(L^{-1+\alpha_L}), \tag{16}$$

since $\frac{1}{N} k \cdot k, \frac{1}{N} q \cdot q \sim \Theta(1)$. This update to the attention variable due to changes in $W_K^{\ell}, W_Q^{\ell}$ will clearly die out as $L \to \infty$ unless $\alpha_L = 1$.

## C.3 Heuristic Analysis of Feature Changes Under Adam

For Adam, the gradient of each individual parameter entry is approximately normalized by its scale [45]. Thus the learning rate $\eta$ sets the size of the updates. This is why we scale the learning rate as $\eta = \frac{1}{L^{1-\alpha_L}\sqrt{N\mathcal{H}}}$ which gives the same scale updates to the weights as SGD

$$\delta W_{O\mathfrak{h}}^{\ell} \approx \eta \, g^{\ell+1} v_{\mathfrak{h}}^{\ell\top} = \frac{1}{L^{1-\alpha_L}\sqrt{N\mathcal{H}}} g^{\ell+1} v_{\mathfrak{h}}^{\ell\top} \tag{17}$$

Again computing the correction to the forward pass we find

$$\delta h^{\ell+1} = \delta h^{\ell} + \frac{1}{L^{\alpha_L}\sqrt{N\mathcal{H}}} \sum_{\mathfrak{h}=1}^{\mathcal{H}} \left( \frac{1}{L^{1-\alpha_L}\sqrt{N\mathcal{H}}} g^{\ell+1} v_{\mathfrak{h}}^{\ell\top} \right) v_{\mathfrak{h}}^{\ell}$$

$$= \delta h^{\ell} + \frac{1}{L} \left[ \frac{1}{N\mathcal{H}} \sum_{\mathfrak{h}} v_{\mathfrak{h}}^{\ell} \cdot v_{\mathfrak{h}}^{\ell} \right] g^{\ell+1} = \Theta(1) \tag{18}$$

Similarly our update generates the same scale of weight updates to the key and query weight matrices

$$\delta W_{K\mathfrak{h}}^{\ell} \sim \frac{1}{L^{1-\alpha_L}\sqrt{N\mathcal{H}}} q_{\mathfrak{h}}^{\ell} h^{\ell\top} \;,\; \delta W_{Q\mathfrak{h}}^{\ell} \sim \frac{1}{L^{1-\alpha_L}\sqrt{N\mathcal{H}}} k_{\mathfrak{h}}^{\ell} h^{\ell\top} \tag{19}$$

We can therefore follow the identical argument to identify the scale of the change to the pre-attention variables $\delta \mathcal{A}_{\mathfrak{h}}^{\ell} = \Theta(L^{1-\alpha_L})$.

## C.4 What Counts as Feature Learning for Attention Layers?

Any parameterization with $\alpha_N \in [\frac{1}{2}, 1]$ will cause all updates to $\mathcal{A}_{\mathfrak{h}}^{\ell}$ and entries of $\boldsymbol{h}^{\ell+1}$ to be $\Theta_{N,H,L}(1)$ across finite $N$. The entries of $\boldsymbol{q}$ and $\boldsymbol{k}$ only move by $\Theta_N(1)$ if $\alpha_{\mathcal{A}} = 1$ ($\mu$P scaling). However, we argue that this criterion is not strictly necessary. Rather, feature learning could alternatively be defined in terms of evolution of macroscopic variables ($H$, $\mathcal{A}$, $f$, etc) rather than preactivation or key/query vector entries themselves. Table 3 summarizes two example values of $\alpha_{\mathcal{A}}$ which are of special interest for their $N \to \infty$ limits.

| | Variance of $\mathcal{A}(0)$ | Update to $\mathcal{A}$ | Update to $\boldsymbol{k}, \boldsymbol{q}$ Entries |
|---|---|---|---|
| $\alpha_{\mathcal{A}} = 1$ ($\mu$P) | $\Theta(N^{-1})$ | $\Theta(1)$ | $\Theta(1)$ |
| $\alpha_{\mathcal{A}} = \frac{1}{2}$ | $\Theta(1)$ | $\Theta(1)$ | $\Theta(N^{-\frac{1}{2}})$ |

Table 3: Two interesting choices of scaling for the attention layer exponent $\alpha_{\mathcal{A}}$ which give approximately constant updates to the attention matrices $\mathcal{A}_{\mathfrak{h}}$. The $\mu$P scaling $\alpha_{\mathcal{A}} = 1$ causes the entries of the key/query vector entries to move non-negligibly but causes all heads to be identical (and all $\mathcal{A} = 0$) at initialization. Scaling instead with $\alpha_{\mathcal{A}} = \frac{1}{2}$ causes the $\mathcal{A}$ variables to be random but still non-negligibly updated under training.

The choice $\alpha_{\mathcal{A}} = \frac{1}{2}$ allows the variance of $\mathcal{A}_{\mathfrak{h}}^{\ell}$ to be constant size as a function of $N$ while also enabling learning of these variables. We verify these scalings in Figure 13.

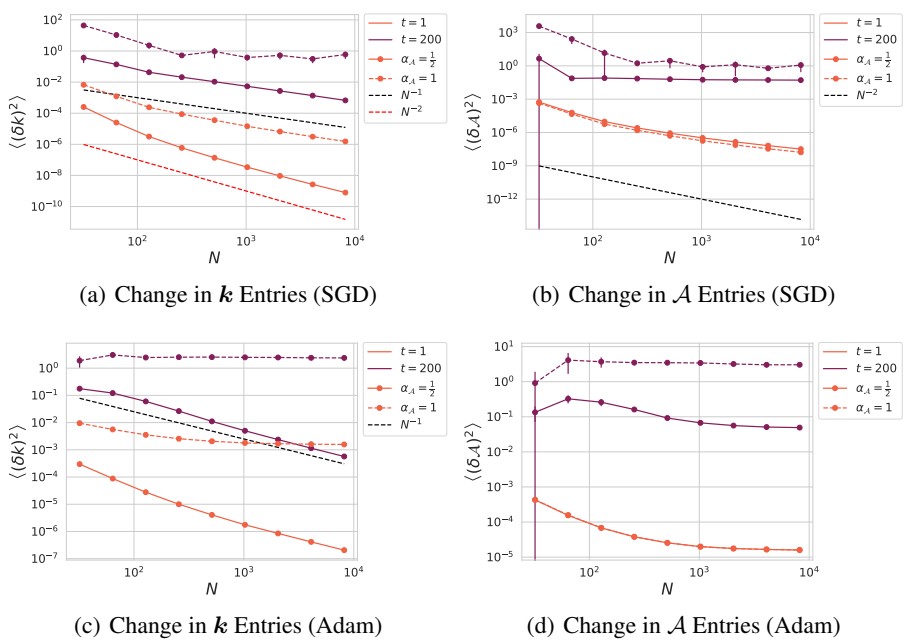

(a) Change in $\boldsymbol{k}$ Entries (SGD)

(b) Change in $\mathcal{A}$ Entries (SGD)

(c) Change in $\boldsymbol{k}$ Entries (Adam)

(d) Change in $\mathcal{A}$ Entries (Adam)

Figure 13: The update to (a) key $\boldsymbol{k}_{\mathfrak{h}}$ entries and (b) pre-attention variables $\mathcal{A}_{\mathfrak{h}}$ after $t$ steps of gradient descent for scaling exponents $\alpha_{\mathcal{A}} \in \{1, \frac{1}{2}\}$. At the first step of SGD, the updates to the keys and attention variables are suppressed due to a lack of correlation between $\boldsymbol{W}_O$ and the gradient $\frac{\partial f}{\partial \boldsymbol{h}}$. After training for multiple steps, this correlation increases and non-negligible updates to the attention variables occur. (c)-(d) The same but for the Adam optimizer with our proposed parameterization.

# D   DMFT Primer and Simple Examples

## D.1   Main Conceptual Idea of the Approach

Dynamical mean field theory is a method that was developed in the physics of spin glasses for dealing with dynamical systems that depend on a **fixed source of disorder**. The disorder could be random couplings between sites in a spin glass model [34], random connections between neurons in a random recurrent neural network [35], random data drawn from a distribution [37, 39] or the random initial weights in a deep neural network [15, 11]. In our case, we are interested in the last example, where the feature learning dynamics of a randomly initialized transformer is a function of the initial weights in each layer. In what follows, we will give a primer on the main objects which typically appear in a DMFT analysis (the correlation and response functions) to illustrate the main ideas of the approach.

## D.2   Example 1: Linear Dynamics with GOE Matrix

In this section, we discuss and derive the DMFT equations for the simplest possible example, a linear dynamical system with a Gaussian symmetric coupling matrix.

In this example we show that the DMFT path integral is computing something non-trivial about the kinds of dynamics induced by a linear dynamical system with a random matrix. In this linear example, the DMFT path integral encodes spectral properties of the random matrix.

Let's consider the simplest possible example: $\frac{d}{dt}h_i(t) = \frac{1}{\sqrt{N}}\sum_{j=1}^{N}W_{ij}h_j(t)$ where $W_{ij} = W_{ji}$ is a Gaussian symmetric matrix (GOE). This matrix is **fixed** while the state $\boldsymbol{h}(t) \in \mathbb{R}^N$ evolves. The path integral appraoch would tell you that in the $N \to \infty$ limit, every neuron $i$ has identical statistics given by the stochastic integro-differential equation

$$\partial_t h(t) = u(t) + \int_0^t ds R(t,s)h(s) \, , \; u(t) \sim \mathcal{GP}(0, C(t,s))$$

$$C(t,s) = \langle h(t)h(s)\rangle \, , \; R(t,s) = \left\langle \frac{\delta h(t)}{\delta u(s)} \right\rangle \tag{20}$$

where $\langle \cdot \rangle$ denotes an average over the random variables $u(t)$. In this picture, the averages $\langle \rangle$ over the noise can also be interpreted as averages over all $N$ neurons in the system, each of which are independent. This stochastic equation can be used to close the evolution equations for the correlation $C(t,s)$ and linear response function $R(t,s)$.

A generic result of this path integral DMFT picture is

1. All neurons (all variables $h_i$) decouple statistically. The presence of all other neurons only enters through "macroscopic" quantities $C(t,s)$ and $R(t,s)$ known as the correlation and response functions. The distribution of these functions over random realizations satisfies a large deviations principle where the distribution over $C, R$ has the form $p(C, R) \sim e^{-NS(C,R)}$ where $S$ is the DMFT action obtained from the path integral method.

2. Extra *memory terms* like $\int_0^t R(t,s)h(s)$ appear which depend on the state at earlier times $s < t$. The Markovian (deterministic) system for $p(\boldsymbol{h}|\boldsymbol{W})$ becomes stochastic and non-markovian after marginalizing $p(\boldsymbol{h}) = \int d\boldsymbol{W} p(\boldsymbol{h}|\boldsymbol{W})p(\boldsymbol{W})$. I would argue these memory terms are not obvious apriori but are systematic to compute in this framework.

Since this toy example is a **linear dynamical system**, one can also identify a connection between the DMFT correlation and response and spectral properties of the random matrix $\boldsymbol{W}$. We note that the response has the form

$$R(t,s) = \frac{1}{N}\text{Tr}\exp\left(W(t-s)\right) = \int d\lambda \rho(\lambda)e^{\lambda(t-s)} \tag{21}$$

where $\rho(\lambda)$ is the eigenvalue density of $W$. In fact a Fourier transform of our DMFT equation recovers the semicircle law $\rho(\lambda) = \frac{1}{\pi}\text{Im}R(i\lambda) = \frac{1}{2\pi}\sqrt{[4-\lambda^2]_+}$ for the eigenvalues.

In general, one can think of DMFT as a more powerful version of this method that can also handle nonlinearities.

### D.3 Example 2: Deep Linear Network Updates

In this section I will try showing how this DMFT approach can give useful insights into reasoning about learning updates which are not obvious apriori. While our paper advocates for taking depth $L \to \infty$ in a residual network, we first thought about simply scaling depth in a standard MLP. Below we show how the proliferation of response terms gives a different predicted scaling with $L$ than if we naively disregarded response terms.

Consider a non-residual linear MLP network with $\mu$P/mean-field scaling with $L$ hidden layers with $N \to \infty$. Train the model for a single step of gradient descent with learning rate $\eta$ on a data point $(x, y)$ with $|x|^2 = 1$ and $y = 1$ and output multiplier $1/\gamma_0$. The forward pass variables $\boldsymbol{h}^\ell(t)$ and the backward pass variables $\boldsymbol{g}^\ell(t)$ are defined recursively as

$$\boldsymbol{h}^{\ell+1}(t) = \frac{1}{\sqrt{N}}\boldsymbol{W}^\ell(t)\boldsymbol{h}^\ell(t) = \frac{1}{\sqrt{N}}\boldsymbol{W}^\ell(0)\boldsymbol{h}^\ell(t) + \eta\gamma_0 \sum_{s<t} H^\ell(t,s)\boldsymbol{g}^{\ell+1}(s) \tag{22}$$

$$\boldsymbol{g}^\ell(t) = \frac{1}{\sqrt{N}}\boldsymbol{W}^\ell(t)^\top \boldsymbol{g}^{\ell+1}(t) = \frac{1}{\sqrt{N}}\boldsymbol{W}^\ell(0)^\top \boldsymbol{g}^{\ell+1}(t) + \eta\gamma_0 \sum_{s<t} G^{\ell+1}(t,s)\boldsymbol{h}^{\ell+1}(s) \tag{23}$$

Now, naively, one may think that $\frac{1}{\sqrt{N}}\boldsymbol{W}^\ell(0)\boldsymbol{h}^\ell(t)$ has entries that are independently Gaussian with covariance $H^\ell(t,t')$. However, this is incorrect and the **DMFT response functions** give an additional correction.

**The DMFT Equations for this Model** Following the approach of [15, 11], we find the following DMFT equations for the preactivations $h^\ell$ after 1 step of training

$$h^\ell(0) = u^\ell(0) , \; g^\ell(0) = r^\ell(0)$$
$$h^\ell(1) = u^\ell(1) + A^{\ell-1}(1,0)g^\ell(0) + \eta\gamma_0 H^{\ell-1}(1,0)g^\ell(0) \tag{24}$$

where the random variables $u^\ell(0), u^\ell(1)$ and $r^\ell(0)$ have the following covariance structure

$$\langle u^\ell(0)u^\ell(0)\rangle = H^{\ell-1}(0,0) , \; \langle u^\ell(1)u^\ell(0)\rangle = H^{\ell-1}(1,0) , \; \langle u^\ell(1)u^\ell(1)\rangle = H^{\ell-1}(1,1)$$
$$\langle r^\ell(0)r^\ell(0)\rangle = G^{\ell+1}(0,0), \tag{25}$$

and the feature kernels $H^\ell(t,t')$, $G^\ell(t,t')$ and response functions $A^\ell(t,t')$ have the form

$$H^\ell(t,t') = \langle h^\ell(t)h^\ell(t')\rangle \; , \; G^\ell(t,t') = \langle g^\ell(t)g^\ell(t')\rangle$$
$$A^\ell(t,t') = \left\langle \frac{\delta h^\ell(t)}{\delta r^\ell(t')} \right\rangle \tag{26}$$

These recursions can be solved with the initial conditions $H^\ell(0,0) = 1$ and $G^\ell(0,0) = 1$ which implies that $H^\ell(1,0) = 1$ so that

$$A^\ell(1,0) = \eta\gamma_0 + A^{\ell-1}(1,0) = \ell\eta\gamma_0 \tag{27}$$

Using this equation, we find

$$H^\ell(1,1) = \langle h^\ell(1)h^\ell(1)\rangle = H^{\ell-1}(1,1) + \eta^2\gamma_0^2\ell^2 = 1 + \eta^2\gamma_0^2 \sum_{k=1}^\ell k^2 \tag{28}$$

This is the DMFT prediction for the scale of the feature kernels after a step of training.

**Neglecting the DMFT Response Gives Incorrect Depth Scalings for MLPs** However, if we had neglected the DMFT response functions and approximated the dynamics as

$$H^\ell(1,1) = H^{\ell-1}(1,1) + \eta^2\gamma_0^2 \implies H^\ell(1,1) = 1 + \eta^2\gamma_0^2 \ell \tag{29}$$

The feature variance after $t = 1$ step of gradient descent $H^\ell = \langle h^\ell(1)^2\rangle$ after $t = 1$ step, the final layer

$$H^L \sim \begin{cases} 1 + \frac{1}{3}\eta^2\gamma_0^2 L^3 & \text{DMFT Response Included (Full DMFT)} \\ 1 + \eta^2\gamma_0^2 L & \text{DMFT Response Neglected} \end{cases} \tag{30}$$

We see that without the response terms we get a completely different scaling prediction with $L$!

**Scaled Residual Networks**   For $\frac{1}{\sqrt{L}}$ residual block scaling ($\alpha_L = \frac{1}{2}$), the response functions are still important to accurately characterize the dynamics and contribute $\Theta_L(1)$ corrections to the feature learning dynamics as $L \to \infty$. However, for the $1/L$ block multiplier scaling, the response functions do not contribute in the limit. These facts are *not-apriori obvious* but follow from the DMFT analysis (either path integral or cavity approach).

# E   DMFT Analysis for Transformers

In this section, we derive the limiting equations for the infinite head limit $\mathcal{H} \to \infty$ of training with SGD. The results can be easily extended to SGD with momentum following the methods of [11, 15].

## E.1   Deriving the DMFT Action

In this section we will derive the limiting equations of motion for stochastic gradient descent in the $\mathcal{H} \to \infty$ limit. We start by defining the loss function which we aim to minimize

$$\mathcal{L} = \int d\boldsymbol{x} \, p(\boldsymbol{x}) \, \ell[f(\boldsymbol{x})] \tag{31}$$

where $p(\boldsymbol{x})$ is the data distribution of interest. We note that this can be the population loss or the empirical loss on a finite collection of points. We let $\Delta(x, t) \equiv -\frac{\partial \ell[f(\boldsymbol{x}, t)]}{\partial f(\boldsymbol{x}, t)}$ represent the error signal on datapoint $\boldsymbol{x}$. At each step of training $t$ a batch of examples $\mathfrak{B}_t = \{\boldsymbol{x}_1(t), ..., \boldsymbol{x}_{|\mathfrak{B}_t|}(t)\}$ is generated and used to estimate a gradient for SGD. In what follows, we let $\mathbb{E}_{\boldsymbol{x} \sim \mathfrak{B}_t}$ represent averages over the minibatch at time $t$. We emphasize here that the batches $\mathfrak{B}_t$ are assumed to be given or fixed and are not averaged over as random draws from $p(\boldsymbol{x})$, but rather our expectation simply denotes the empirical mean over the minibatch at time $t$

$$\mathbb{E}_{\boldsymbol{x} \sim \mathfrak{B}_t}[f(\boldsymbol{x})] = \frac{1}{|\mathfrak{B}_t|} \sum_{\boldsymbol{x} \in \mathfrak{B}_t} f(\boldsymbol{x}). \tag{32}$$

We start by expressing again the forward pass equations for each layer. To make this analysis more compressed while still capturing all of the interesting aspects, we will first compute the equations of motion in the absence of MLP layers (which were analyzed in prior works, see Appendix E) and layernorm (which in the limit it will only apply a deterministic affine transformation to each of the entries of the residual stream and the backward pass gradient variables as we will show explicitly in Appendix E.6). In the absence of layernorm, our forward pass has the form

$$\boldsymbol{h}_{\mathfrak{s}}^{\ell+1}(\boldsymbol{x}, t) = \boldsymbol{h}_{\mathfrak{s}}^{\ell}(\boldsymbol{x}, t) + \frac{\beta_0}{L^{\alpha_L}} \text{MHSA}\left(\boldsymbol{h}^{\ell}(\boldsymbol{x}, t)\right)_{\mathfrak{s}} \tag{33}$$

where the MHSA layer is

$$\text{MHSA}\left(\boldsymbol{h}^{\ell}(\boldsymbol{x}, t)\right)_{\mathfrak{s}} = \frac{1}{\sqrt{\mathcal{H}}} \sum_{\mathfrak{h} \in [\mathcal{H}]} \sum_{\mathfrak{s}' \in [\mathcal{S}]} \boldsymbol{o}_{\mathfrak{h}\mathfrak{s}'}^{\ell}(\boldsymbol{x}, t) \sigma_{\mathfrak{h}\mathfrak{s}\mathfrak{s}'}^{\ell}(\boldsymbol{x}, t)$$

$$\boldsymbol{o}_{\mathfrak{h}\mathfrak{s}}^{\ell}(\boldsymbol{x}, t) = \frac{1}{\sqrt{N}} \boldsymbol{W}_{O\mathfrak{h}}^{\ell}(t) \boldsymbol{v}_{\mathfrak{h}\mathfrak{s}}^{\ell}(\boldsymbol{x}, t) \tag{34}$$

$$\boldsymbol{v}_{\mathfrak{h}\mathfrak{s}}^{\ell}(\boldsymbol{x}, t) = \frac{1}{\sqrt{N\mathcal{H}}} \boldsymbol{W}_{V\mathfrak{h}}^{\ell}(t) \boldsymbol{h}_{\mathfrak{s}}^{\ell}(\boldsymbol{x}, t)$$

$$\sigma_{\mathfrak{h}\mathfrak{s}\mathfrak{s}'}^{\ell}(\boldsymbol{x}, t) = \sigma\left(\boldsymbol{\mathcal{A}}_{\mathfrak{h}}^{\ell}(\boldsymbol{x}, t)\right)_{\mathfrak{s}\mathfrak{s}'} \, , \, \boldsymbol{\mathcal{A}}_{\mathfrak{h}\mathfrak{s}\mathfrak{s}'}^{\ell}(\boldsymbol{x}, t) = \frac{1}{N^{\alpha_{\mathcal{A}}}} \boldsymbol{k}_{\mathfrak{h}\mathfrak{s}}^{\ell}(\boldsymbol{x}, t) \cdot \boldsymbol{q}_{\mathfrak{h}\mathfrak{s}'}^{\ell}(\boldsymbol{x}, t)$$

$$\boldsymbol{k}_{\mathfrak{h}\mathfrak{s}}^{\ell}(\boldsymbol{x}, t) = \frac{1}{N^{\frac{3}{2} - \alpha_{\mathcal{A}}} \sqrt{\mathcal{H}}} \boldsymbol{W}_{K\mathfrak{h}}^{\ell}(t) \boldsymbol{h}_{\mathfrak{s}}^{\ell}(\boldsymbol{x}, t)$$

$$\boldsymbol{q}_{\mathfrak{h}\mathfrak{s}}^{\ell}(\boldsymbol{x}, t) = \frac{1}{N^{\frac{3}{2} - \alpha_{\mathcal{A}}} \sqrt{\mathcal{H}}} \boldsymbol{W}_{Q\mathfrak{h}}^{\ell}(t) \boldsymbol{h}_{\mathfrak{s}}^{\ell}(\boldsymbol{x}, t) \, , \tag{35}$$

To compute the weight dynamics we again introduce the necessary gradient fields which we previously argued have $\Theta(1)$ entries

$$\boldsymbol{g}_{\mathfrak{s}}^{\ell}(\boldsymbol{x}, t) \equiv \gamma_0 N \mathcal{H} \frac{\partial f(\boldsymbol{x}, t)}{\partial \boldsymbol{h}_{\mathfrak{s}}^{\ell}(\boldsymbol{x}, t)} \tag{36}$$

We also introduce the following intermediate quantities which are necessary to characterize the backward pass through the attention layer

$$M_{\mathfrak{h}\mathfrak{s}\mathfrak{s}'}^{\ell}(\boldsymbol{x},t) \equiv \frac{1}{N\sqrt{\mathcal{H}}}\, \boldsymbol{g}_{\mathfrak{s}}^{\ell+1}(\boldsymbol{x},t) \cdot \boldsymbol{o}_{\mathfrak{h}\mathfrak{s}'}^{\ell}(\boldsymbol{x},t)$$

$$\dot{\sigma}_{\mathfrak{h}\mathfrak{s}\mathfrak{s}'\mathfrak{s}''}^{\ell}(\boldsymbol{x},t) \equiv \frac{\partial \sigma_{\mathfrak{h}\mathfrak{s}\mathfrak{s}'}^{\ell}(\boldsymbol{x},t)}{\partial \mathcal{A}_{\mathfrak{h}\mathfrak{s}\mathfrak{s}''}^{\ell}(\boldsymbol{x},t)} \tag{37}$$

We need to break up each of the weight matrices into their initial component and their update from SGD

$$\boldsymbol{W}_{O\mathfrak{h}}^{\ell}(t) = \boldsymbol{W}_{O\mathfrak{h}}^{\ell}(0) + \frac{\beta_0\eta_0\gamma_0}{L^{1-\alpha_L}\sqrt{N\mathcal{H}}}\sum_{t'<t}\mathbb{E}_{\boldsymbol{x}\sim\mathfrak{B}_{t'}}\sum_{\mathfrak{s}}\Delta(\boldsymbol{x},t')\tilde{\boldsymbol{g}}_{\mathfrak{s}}^{\ell}(\boldsymbol{x},t')\boldsymbol{v}_{\mathfrak{h}\mathfrak{s}}^{\ell\sigma}(\boldsymbol{x},t')^{\top}$$

$$\boldsymbol{W}_{V\mathfrak{h}}^{\ell}(t) = \boldsymbol{W}_{V\mathfrak{h}}^{\ell}(0) + \frac{\beta_0\eta_0\gamma_0}{L^{1-\alpha_L}\sqrt{N\mathcal{H}}}\sum_{t'<t}\mathbb{E}_{\boldsymbol{x}\sim\mathfrak{B}_{t'}}\sum_{\mathfrak{s}\mathfrak{s}'}\Delta(\boldsymbol{x},t')\sigma_{\mathfrak{s}\mathfrak{s}'}^{\ell}\tilde{\boldsymbol{g}}_{O\mathfrak{h}\mathfrak{s}}^{\ell}(\boldsymbol{x},t')\boldsymbol{h}_{\mathfrak{s}'}^{\ell}(\boldsymbol{x},t')^{\top}$$

$$\boldsymbol{W}_{K\mathfrak{h}}^{\ell}(t) = \boldsymbol{W}_{K\mathfrak{h}}^{\ell}(0) + \frac{\beta_0\eta_0\gamma_0}{L^{1-\alpha_L}\sqrt{N\mathcal{H}}}\sum_{t'<t}\mathbb{E}_{\boldsymbol{x}\sim\mathfrak{B}_{t'}}\sum_{\mathfrak{s}\mathfrak{s}'\mathfrak{s}''}\Delta(\boldsymbol{x},t')M_{\mathfrak{h}\mathfrak{s}\mathfrak{s}'}^{\ell}(\boldsymbol{x},t')\dot{\sigma}_{\mathfrak{s}\mathfrak{s}'\mathfrak{s}''}^{\ell}\boldsymbol{q}_{\mathfrak{h}\mathfrak{s}''}^{\ell}(\boldsymbol{x},t')\boldsymbol{h}_{\mathfrak{s}}^{\ell}(\boldsymbol{x},t')^{\top}$$

$$\boldsymbol{W}_{Q\mathfrak{h}}^{\ell}(t) = \boldsymbol{W}_{Q\mathfrak{h}}^{\ell}(0) + \frac{\beta_0\eta_0\gamma_0}{L^{1-\alpha_L}\sqrt{N\mathcal{H}}}\sum_{t'<t}\mathbb{E}_{\boldsymbol{x}\sim\mathfrak{B}_{t'}}\sum_{\mathfrak{s}\mathfrak{s}'\mathfrak{s}''}\Delta(\boldsymbol{x},t')M_{\mathfrak{h}\mathfrak{s}\mathfrak{s}'}^{\ell}(\boldsymbol{x},t')\dot{\sigma}_{\mathfrak{s}\mathfrak{s}'\mathfrak{s}''}^{\ell}\boldsymbol{k}_{\mathfrak{h}\mathfrak{s}}^{\ell}(\boldsymbol{x},t')\boldsymbol{h}_{\mathfrak{s}''}^{\ell}(\boldsymbol{x},t')^{\top}$$

We can now express the residual stream as

$$\boldsymbol{h}_{\mathfrak{s}}^{\ell+1}(\boldsymbol{x},t) = \boldsymbol{h}_{\mathfrak{s}}^{\ell}(\boldsymbol{x},t) + \beta_0 L^{-\alpha_L}\,\bar{\boldsymbol{\chi}}_{O\mathfrak{s}}^{\ell+1}(\boldsymbol{x},t)$$
$$+ \eta_0\gamma_0\beta_0^2 L^{-1}\sum_{t'<t}\mathbb{E}_{\boldsymbol{x}'\sim\mathfrak{B}_{t'}}\Delta(\boldsymbol{x}',t')\sum_{\mathfrak{s}'}\boldsymbol{g}_{\mathfrak{s}'}^{\ell+1}(\boldsymbol{x}',t')V_{\mathfrak{s}\mathfrak{s}'}^{\ell\sigma}(\boldsymbol{x},\boldsymbol{x}',t,s) \tag{38}$$

where we introduced the fields

$$\bar{\boldsymbol{\chi}}_{O\mathfrak{s}}^{\ell}(\boldsymbol{x},t) = \frac{1}{\sqrt{\mathcal{H}}}\sum_{\mathfrak{h}=1}^{\mathcal{H}}\sum_{\mathfrak{s}'}\boldsymbol{\chi}_{O\mathfrak{h}\mathfrak{s}'}^{\ell}(\boldsymbol{x},t)\sigma_{\mathfrak{s}\mathfrak{s}'}^{\ell}(\boldsymbol{x},t)\,,\quad \boldsymbol{\chi}_{O\mathfrak{h}\mathfrak{s}}^{\ell}(\boldsymbol{x},t) = \frac{1}{\sqrt{N}}\boldsymbol{W}_{O\mathfrak{h}}^{\ell}(0)\boldsymbol{v}_{\mathfrak{h}\mathfrak{s}}^{\ell}(\boldsymbol{x},t)$$

and the kernel

$$V_{\mathfrak{s}\mathfrak{s}'}^{\ell\sigma}(\boldsymbol{x},\boldsymbol{x}',t,s) = \frac{1}{\mathcal{H}N}\sum_{\mathfrak{h}=1}^{\mathcal{H}}\boldsymbol{v}_{\mathfrak{h}\mathfrak{s}}^{\ell\sigma}(\boldsymbol{x},t)\cdot\boldsymbol{v}_{\mathfrak{h}\mathfrak{s}'}^{\ell\sigma}(\boldsymbol{x}',t') = \frac{1}{\mathcal{H}}\sum_{\mathfrak{h}=1}^{\mathcal{H}}\sum_{\mathfrak{s}''\mathfrak{s}'''}\sigma_{\mathfrak{s}\mathfrak{s}''}^{\ell}\sigma_{\mathfrak{s}'\mathfrak{s}'''}^{\ell}V_{\mathfrak{h}\mathfrak{s}''\mathfrak{s}'''}^{\ell}(\boldsymbol{x},\boldsymbol{x}',t,t')$$
$$\tag{39}$$

We see that, regardless of the choice of $\alpha_L$, the update to the residual stream due to feature learning will scale as $1/L$ which is necessary for a stable infinite depth limit [11]. To track the dynamics of the value vectors, we must also track the variables

$$\boldsymbol{\chi}_{V\mathfrak{h}\mathfrak{s}}^{\ell}(\boldsymbol{x},t) = \frac{1}{\sqrt{N\mathcal{H}}}\boldsymbol{W}_{V\mathfrak{h}}^{\ell}(0)\boldsymbol{h}_{\mathfrak{s}}^{\ell}(\boldsymbol{x},t) \tag{40}$$

We similarly find the following for the key dynamics

$$\boldsymbol{k}_{\mathfrak{h}\mathfrak{s}}^{\ell}(\boldsymbol{x},t) = \boldsymbol{\chi}_{K\mathfrak{h}\mathfrak{s}}^{\ell}(\boldsymbol{x},t)$$
$$+ \frac{\beta_0\eta_0\gamma_0}{L^{1-\alpha_L}N^{1-\alpha_{\mathcal{A}}}}\sum_{t'<t}\mathbb{E}_{\boldsymbol{x}\sim\mathfrak{B}_{t'}}\sum_{\mathfrak{s}'\mathfrak{s}''\mathfrak{s}'''}\Delta(\boldsymbol{x},t')M_{\mathfrak{h}\mathfrak{s}'\mathfrak{s}''}^{\ell}(\boldsymbol{x},t')\dot{\sigma}_{\mathfrak{s}'\mathfrak{s}''\mathfrak{s}'''}^{\ell}\boldsymbol{q}_{\mathfrak{h}\mathfrak{s}'''}^{\ell}(\boldsymbol{x},t')H_{\mathfrak{s}\mathfrak{s}'}^{\ell}(\boldsymbol{x},\boldsymbol{x}',t,t')$$
$$\boldsymbol{\chi}_{K\mathfrak{h}\mathfrak{s}}^{\ell}(\boldsymbol{x},t) = \frac{1}{N^{\frac{3}{2}-\alpha_{\mathcal{A}}}\sqrt{\mathcal{H}}}\boldsymbol{W}_{K\mathfrak{h}}^{\ell}(0)\boldsymbol{h}_{\mathfrak{s}}^{\ell}(\boldsymbol{x},t) \tag{41}$$

and an analogous update equation holds for the query dynamics $\boldsymbol{q}$

$$\boldsymbol{q}_{\mathfrak{h}\mathfrak{s}}^{\ell}(\boldsymbol{x},t) = \boldsymbol{\chi}_{Q\mathfrak{h}\mathfrak{s}}^{\ell}(\boldsymbol{x},t)$$
$$+ \frac{\beta_0\eta_0\gamma_0}{L^{1-\alpha_L}N^{1-\alpha_{\mathcal{A}}}}\sum_{t'<t}\mathbb{E}_{\boldsymbol{x}'\sim\mathfrak{B}_{t'}}\sum_{\mathfrak{s}'\mathfrak{s}''}\Delta(\boldsymbol{x}',t')M_{\mathfrak{h}\mathfrak{s}\mathfrak{s}'}^{\ell}(\boldsymbol{x}',t')\dot{\sigma}_{\mathfrak{s}\mathfrak{s}'\mathfrak{s}''}^{\ell}(\boldsymbol{x}',t')\boldsymbol{k}_{\mathfrak{h}\mathfrak{s}''}^{\ell}(\boldsymbol{x}',t')H_{\mathfrak{s}\mathfrak{s}'}^{\ell}(\boldsymbol{x},\boldsymbol{x}',t,t')$$
$$\boldsymbol{\chi}_{Q\mathfrak{h}\mathfrak{s}}^{\ell}(\boldsymbol{x},t) = \frac{1}{N^{\frac{3}{2}-\alpha_{\mathcal{A}}}\sqrt{\mathcal{H}}}\boldsymbol{W}_{Q\mathfrak{h}}^{\ell}(0)\boldsymbol{h}_{\mathfrak{s}}^{\ell}(\boldsymbol{x},t) \tag{42}$$

We see that if the residual stream is frozen so that $\chi_K, \chi_Q$ are static, the keys and queries will only evolve in the $N, L \to \infty$ limits if $\alpha_L = \alpha_A = 1$ as we argued in the main text. From these equations we can deduce the pre-attention values $\mathcal{A}^\ell_{\mathfrak{h}\mathfrak{s}\mathfrak{s}'}(\boldsymbol{x}, t)$.

Next, we examine the dynamics of the $M^\ell_{\mathfrak{h}\mathfrak{s}}(\boldsymbol{x}, t)$ variables which are defined as

$$M^\ell_{\mathfrak{h}\mathfrak{s}\mathfrak{s}'}(\boldsymbol{x}, t) = \frac{1}{N\sqrt{\mathcal{H}}} \boldsymbol{g}^{\ell+1}_{\mathfrak{s}}(\boldsymbol{x}, t) \cdot \boldsymbol{o}^\ell_{\mathfrak{h}\mathfrak{s}'}(\boldsymbol{x}, t).$$

We can verify that $M^\ell_{\mathfrak{h}}$ are all $\Theta_{N,\mathcal{H}}(1)$ throughout training by expanding the dynamics of the attention output $\boldsymbol{o}^\ell_{\mathfrak{h}\mathfrak{s}'}(\boldsymbol{x}, t)$.

### E.1.1 Backward Pass

Next, we need to work out the recursions for the backward pass variables. After these equations have been worked out, we can isolate the dependence of the full dynamics on all of the initial weights.

**MHSA Layer**  We will start by differentiating through the MHSA layer

$$\boldsymbol{g}^\ell_{\mathfrak{s}}(\boldsymbol{x}, t) = \sum_{\mathfrak{s}'} \left( \frac{\partial \tilde{\boldsymbol{h}}^\ell_{\mathfrak{s}'}(\boldsymbol{x}, t)}{\partial \boldsymbol{h}^\ell_{\mathfrak{s}}(\boldsymbol{x}, t)^\top} \right)^\top \boldsymbol{g}^{\ell+1}_{\mathfrak{s}'}$$

$$= \boldsymbol{g}^{\ell+1}_{\mathfrak{s}}(\boldsymbol{x}, t) + \frac{\beta_0}{L^{\alpha_L}} \sum_{\mathfrak{s}'} \left( \frac{\partial}{\partial \boldsymbol{h}^\ell_{\mathfrak{s}}(\boldsymbol{x}, t)^\top} \mathrm{MHSA}(\boldsymbol{h}^\ell(\boldsymbol{x}, t))_{\mathfrak{s}'} \right)^\top \boldsymbol{g}^{\ell+1}_{\mathfrak{s}'}(\boldsymbol{x}, t)$$

$$= \boldsymbol{g}^{\ell+1}_{\mathfrak{s}}(\boldsymbol{x}, t) + \frac{\beta_0}{L^{\alpha_L}\sqrt{N\mathcal{H}}} \sum_{\mathfrak{s}'} \sum_{\mathfrak{h}} \boldsymbol{W}^\ell_{V\mathfrak{h}}(t)^\top \boldsymbol{g}^{\ell\sigma}_{O\mathfrak{h}\mathfrak{s}}(\boldsymbol{x}, t)$$

$$+ \frac{\beta_0}{L^{\alpha_L}\sqrt{N\mathcal{H}}} \sum_{\mathfrak{h}} \boldsymbol{W}^\ell_{Q\mathfrak{h}}(t)^\top \boldsymbol{k}^{\ell M \dot\sigma}_{\mathfrak{h}\mathfrak{s}}(\boldsymbol{x}, t) + \frac{\beta_0}{L^{\alpha_L}\sqrt{N\mathcal{H}}} \sum_{\mathfrak{h}} \boldsymbol{W}^\ell_{K\mathfrak{h}}(t)^\top \boldsymbol{q}^{\ell M \dot\sigma}_{\mathfrak{h}\mathfrak{s}}(\boldsymbol{x}, t) \quad (43)$$

$$= \boldsymbol{g}^{\ell+1}_{\mathfrak{s}}(\boldsymbol{x}, t) + \frac{\beta_0}{L^{\alpha_L}} \left[ \bar{\boldsymbol{\xi}}^\ell_{Q\mathfrak{s}}(\boldsymbol{x}, t) + \bar{\boldsymbol{\xi}}^\ell_{K\mathfrak{s}}(\boldsymbol{x}, t) + \bar{\boldsymbol{\xi}}^\ell_{V\mathfrak{s}}(\boldsymbol{x}, t) \right]$$

$$+ \frac{\beta_0^2 \eta_0 \gamma_0}{L} \sum_{t < t'} \mathbb{E}_{\boldsymbol{x}' \sim t'} \Delta(\boldsymbol{x}', t') G^{\ell\sigma}_{O\mathfrak{s}\mathfrak{s}'}(\boldsymbol{x}, \boldsymbol{x}', t, t') \boldsymbol{h}^\ell_{\mathfrak{s}'}(\boldsymbol{x}', t')$$

$$+ \frac{\beta_0^2 \eta_0 \gamma_0}{L} \sum_{t < t'} \mathbb{E}_{\boldsymbol{x}' \sim t'} \Delta(\boldsymbol{x}', t') \left[ K^{\ell M \dot\sigma}_{\mathfrak{s}\mathfrak{s}'}(\boldsymbol{x}, \boldsymbol{x}', t, t') + Q^{\ell M \dot\sigma}_{\mathfrak{s}\mathfrak{s}'}(\boldsymbol{x}, \boldsymbol{x}', t, t') \right] \boldsymbol{h}^\ell_{\mathfrak{s}'}(\boldsymbol{x}', t') \quad (44)$$

where we introduced the variables

$$\boldsymbol{g}^{\ell\sigma}_{O\mathfrak{h}\mathfrak{s}}(\boldsymbol{x}, t) = \sum_{\mathfrak{s}'} \sigma^\ell_{\mathfrak{s}\mathfrak{s}'}(\boldsymbol{x}, t) \boldsymbol{g}^\ell_{O\mathfrak{h}\mathfrak{s}'}(\boldsymbol{x}, t)$$

$$\boldsymbol{k}^{\ell M \dot\sigma}_{\mathfrak{h}\mathfrak{s}}(\boldsymbol{x}, t) = \sum_{\mathfrak{s}'\mathfrak{s}''} M^\ell_{\mathfrak{h}\mathfrak{s}'\mathfrak{s}''}(\boldsymbol{x}, t) \dot\sigma^\ell_{\mathfrak{h}\mathfrak{s}'\mathfrak{s}'''\mathfrak{s}}(\boldsymbol{x}, t) \boldsymbol{k}^\ell_{\mathfrak{h}\mathfrak{s}'}(\boldsymbol{x}, t)$$

$$\boldsymbol{q}^{\ell M \dot\sigma}_{\mathfrak{h}\mathfrak{s}}(\boldsymbol{x}, t) = \sum_{\mathfrak{s}''\mathfrak{s}'''} M^\ell_{\mathfrak{h}\mathfrak{s}\mathfrak{s}''}(\boldsymbol{x}, t) \dot\sigma^\ell_{\mathfrak{h}\mathfrak{s}\mathfrak{s}''\mathfrak{s}'''}(\boldsymbol{x}, t) \boldsymbol{q}^\ell_{\mathfrak{h}\mathfrak{s}'''}(\boldsymbol{x}, t)$$

and their associated kernels

$$G^{\ell\sigma}_{O\mathfrak{s}\mathfrak{s}'}(\boldsymbol{x}, \boldsymbol{x}', t, t') = \frac{1}{N\mathcal{H}} \sum_{\mathfrak{h}=1}^{\mathcal{H}} \boldsymbol{g}^{\ell\sigma}_{O\mathfrak{h}}(\boldsymbol{x}, t) \cdot \boldsymbol{g}^{\ell\sigma}_{O\mathfrak{h}\mathfrak{s}'}(\boldsymbol{x}, t)$$

$$K^{\ell M \dot\sigma}_{\mathfrak{s}\mathfrak{s}'}(\boldsymbol{x}, \boldsymbol{x}', t, t') = \frac{1}{N\mathcal{H}} \sum_{\mathfrak{h}=1}^{\mathcal{H}} \boldsymbol{k}^{\ell M \dot\sigma}_{\mathfrak{h}\mathfrak{s}}(\boldsymbol{x}, t) \cdot \boldsymbol{k}^{\ell M \dot\sigma}_{\mathfrak{h}\mathfrak{s}'}(\boldsymbol{x}', t')$$

$$Q^{\ell M \dot\sigma}_{\mathfrak{s}\mathfrak{s}'}(\boldsymbol{x}, \boldsymbol{x}', t, t') = \frac{1}{N\mathcal{H}} \sum_{\mathfrak{h}=1}^{\mathcal{H}} \boldsymbol{q}^{\ell M \dot\sigma}_{\mathfrak{h}\mathfrak{s}}(\boldsymbol{x}, t) \cdot \boldsymbol{q}^{\ell M \dot\sigma}_{\mathfrak{h}\mathfrak{s}'}(\boldsymbol{x}', t') \quad (45)$$

where we introduced the following random fields

$$\bar{\xi}^\ell_{Vs}(\boldsymbol{x},t) = \frac{1}{\sqrt{\mathcal{H}}} \sum_{\mathfrak{h}s'} \xi^\ell_{V\mathfrak{h}s'}(\boldsymbol{x},t)\sigma^\ell_{\mathfrak{h}s's}(\boldsymbol{x},t)$$

$$\xi^\ell_{V\mathfrak{h}s'}(\boldsymbol{x},t) = \frac{1}{\sqrt{N}} \boldsymbol{W}^\ell_{V\mathfrak{h}}(0)^\top \boldsymbol{g}^\ell_{O\mathfrak{h}s'}(\boldsymbol{x},t)$$

$$\bar{\xi}^\ell_{Qs}(\boldsymbol{x},t) = \frac{1}{\sqrt{\mathcal{H}}} \sum_{\mathfrak{h}s's''} \xi^\ell_{Q\mathfrak{h}s'}(\boldsymbol{x},t)\dot{\sigma}^\ell_{\mathfrak{h}s's''s}(\boldsymbol{x},t)M^\ell_{\mathfrak{h}s's''}(\boldsymbol{x},t)$$

$$\xi^\ell_{V\mathfrak{h}s'}(\boldsymbol{x},t) = \frac{1}{\sqrt{N}} \boldsymbol{W}^\ell_{V\mathfrak{h}}(0)^\top \boldsymbol{g}^\ell_{O\mathfrak{h}s'}(\boldsymbol{x},t)$$

$$\bar{\xi}^\ell_{Ks}(\boldsymbol{x},t) = \frac{1}{\sqrt{\mathcal{H}}} \sum_{\mathfrak{h}s''s'''} \xi^\ell_{K\mathfrak{h}s'''}(\boldsymbol{x},t)\dot{\sigma}^\ell_{\mathfrak{h}ss''s'''}(\boldsymbol{x},t)M^\ell_{\mathfrak{h}ss''}(\boldsymbol{x},t)$$

$$\xi^\ell_{K\mathfrak{h}s'''}(\boldsymbol{x},t) = \frac{1}{\sqrt{N}} \boldsymbol{W}^\ell_{K\mathfrak{h}}(0)^\top \boldsymbol{q}^\ell_{\mathfrak{h}s'''}(\boldsymbol{x},t) \tag{46}$$

### E.1.2  Why we need $\alpha_\mathcal{A} = 1$ for gradient stability

From the previous equation we see that regardless of the choice of $\alpha_\mathcal{A}$ we need to choose the variance of $\boldsymbol{W}^\ell_{K\mathfrak{h}}(0)$ and $\boldsymbol{W}^\ell_{Q\mathfrak{h}}(0)$ so that $\frac{1}{\sqrt{N}}\boldsymbol{W}^\ell_{K\mathfrak{h}}(0)\boldsymbol{q}^\ell_\mathfrak{h}(\boldsymbol{x},t)$ has $\mathcal{O}(1)$ entries. This means that the entries can be at most $\Theta(1)$ and not $\Theta(N^{1-\alpha_\mathcal{A}})$ as we originally stipulated in order to obtain $\boldsymbol{k}$ and $\boldsymbol{q}$ with $\Theta(1)$ entries at initialization. Thus, with this required scaling, we must have either $\alpha_\mathcal{A} = 1$ and $\Theta(1)$ variance of the weights, which leads to attention variables which are $\Theta(N^{-1/2})$ at initialization or we choose $\alpha_\mathcal{A} = \frac{1}{2}$ and choose $\boldsymbol{k}, \boldsymbol{q}$ to have entries of scale $\Theta(N^{-1/2})$ at initialization. These both lead to the same vanishing initial condition for the pre-attention variables. It thus suffices to consider $\mu$P scaling $\alpha_\mathcal{A} = 1$ to study the $N \to \infty$ limit. We stress that this effect is not visible from a simple heuristic analysis of the forward pass variables after an update like we perform in Appendix C.

**Isolating all Dependence on Initial Conditions**    To summarize the previous sections, we begin by collecting all of the stochastic fields which show up in the dynamics and depend on the initial weight matrices. These quantities all come in pairs since for each matrix we need to consider the forward and backward passes through the initial matrix.

The following variables are necessary to characterize the dynamics of the $N\mathcal{H}$-dimensional residual stream

$$\chi^\ell_{O\mathfrak{h}s}(\boldsymbol{x},t) = \frac{1}{\sqrt{N}} \boldsymbol{W}^\ell_{O\mathfrak{h}}(0)\boldsymbol{v}^\ell_{\mathfrak{h}s}(\boldsymbol{x},t) \,,\ \xi^\ell_{O\mathfrak{h}s}(\boldsymbol{x},t) = \frac{1}{\sqrt{N\mathcal{H}}} \boldsymbol{W}^\ell_{O\mathfrak{h}}(0)^\top \boldsymbol{g}^{\ell+1}_s(\boldsymbol{x},t)$$

$$\chi^\ell_{V\mathfrak{h}s}(\boldsymbol{x},t) = \frac{1}{\sqrt{N\mathcal{H}}} \boldsymbol{W}^\ell_{V\mathfrak{h}}(0)\boldsymbol{h}^\ell_s(\boldsymbol{x},t) \,,\ \xi^\ell_{V\mathfrak{h}s}(\boldsymbol{x},t) = \frac{1}{\sqrt{N}} \boldsymbol{W}^\ell_{V\mathfrak{h}}(0)^\top \boldsymbol{g}^\ell_{O\mathfrak{h}s}(\boldsymbol{x},t)$$

$$\chi^\ell_{Q\mathfrak{h}s}(\boldsymbol{x},t) = \frac{1}{\sqrt{N\mathcal{H}}} \boldsymbol{W}^\ell_{Q\mathfrak{h}}(0)\boldsymbol{h}^\ell_s(\boldsymbol{x},t) \,,\ \xi^\ell_{Q\mathfrak{h}s}(\boldsymbol{x},t) = \frac{1}{\sqrt{N}} \boldsymbol{W}^\ell_{Q\mathfrak{h}}(0)^\top \boldsymbol{k}^\ell_{\mathfrak{h}s}(\boldsymbol{x},t)$$

$$\chi^\ell_{K\mathfrak{h}s}(\boldsymbol{x},t) = \frac{1}{\sqrt{N\mathcal{H}}} \boldsymbol{W}^\ell_{K\mathfrak{h}}(0)\boldsymbol{h}^\ell_s(\boldsymbol{x},t) \,,\ \xi^\ell_{K\mathfrak{h}s}(\boldsymbol{x},t) = \frac{1}{\sqrt{N}} \boldsymbol{W}^\ell_{K\mathfrak{h}}(0)^\top \boldsymbol{q}^\ell_{\mathfrak{h}s}(\boldsymbol{x},t) \tag{47}$$

From these variables, we can reconstruct the entire dynamics for the $\boldsymbol{h}, \boldsymbol{g}$ fields. We therefore study the moment generating functional for the above primitive random variables. Let

$$\boldsymbol{\theta}_0 = \{\boldsymbol{W}_{O\mathfrak{h}}^\ell, \boldsymbol{W}_{V\mathfrak{h}}^\ell(0), \boldsymbol{W}_{K\mathfrak{h}}^\ell(0), \boldsymbol{W}_{Q\mathfrak{h}}^\ell(0)\}$$

$$Z[\{\boldsymbol{j}\}] = \mathbb{E}_{\boldsymbol{\theta}_0} \exp\left(\sum_{\ell\mathfrak{h}st} \int d\boldsymbol{x} \left[\boldsymbol{j}_{\mathfrak{h}s}^{\chi_O^\ell}(\boldsymbol{x},t) \cdot \boldsymbol{\chi}_{O\mathfrak{h}s}^\ell(\boldsymbol{x},t) + \boldsymbol{j}_{\mathfrak{h}s}^{\xi_O^\ell}(\boldsymbol{x},t) \cdot \boldsymbol{\xi}_{O\mathfrak{h}s}^\ell(\boldsymbol{x},t)\right]\right)$$

$$\times \exp\left(\sum_{\ell\mathfrak{h}st} \int d\boldsymbol{x} \left[\boldsymbol{j}_{\mathfrak{h}s}^{\chi_K^\ell}(\boldsymbol{x},t) \cdot \boldsymbol{\chi}_{K\mathfrak{h}s}^\ell(\boldsymbol{x},t) + \boldsymbol{j}_{\mathfrak{h}s}^{\xi_K^\ell}(\boldsymbol{x},t) \cdot \boldsymbol{\xi}_{K\mathfrak{h}s}^\ell(\boldsymbol{x},t)\right]\right)$$

$$\times \exp\left(\sum_{\ell\mathfrak{h}st} \int d\boldsymbol{x} \left[\boldsymbol{j}_{\mathfrak{h}s}^{\chi_Q^\ell}(\boldsymbol{x},t) \cdot \boldsymbol{\chi}_{Q\mathfrak{h}s}^\ell(\boldsymbol{x},t) + \boldsymbol{j}_{\mathfrak{h}s}^{\xi_Q^\ell}(\boldsymbol{x},t) \cdot \boldsymbol{\xi}_{Q\mathfrak{h}s}^\ell(\boldsymbol{x},t)\right]\right)$$

$$\times \exp\left(\sum_{\ell\mathfrak{h}st} \int d\boldsymbol{x} \left[\boldsymbol{j}_{\mathfrak{h}s}^{\chi_V^\ell}(\boldsymbol{x},t) \cdot \boldsymbol{\chi}_{V\mathfrak{h}s}^\ell(\boldsymbol{x},t) + \boldsymbol{j}_{\mathfrak{h}s}^{\xi_V^\ell}(\boldsymbol{x},t) \cdot \boldsymbol{\xi}_{V\mathfrak{h}s}^\ell(\boldsymbol{x},t)\right]\right) \quad (48)$$

We multiply by the identity to enforce the definition of these random variables in terms of the initial weights. As an example, we would have for the first random variable $\boldsymbol{\chi}_{\mathfrak{s}}^{\ell+1}(\boldsymbol{x},t)$ and its corresponding pair $\boldsymbol{\xi}_{\mathfrak{s}}^\ell(\boldsymbol{x},t)$

**Attention Output Matrices** In this section we integrate over $\boldsymbol{W}_{O\mathfrak{h}}^\ell, \boldsymbol{W}_{K\mathfrak{h}}^\ell, \boldsymbol{W}_{Q\mathfrak{h}}^\ell$.

$$\ln \mathbb{E}_{\boldsymbol{W}_{O\mathfrak{h}}^\ell(0)} \exp\left(-\frac{i}{\sqrt{N}} \sum_{ts} \int d\boldsymbol{x} \, \mathrm{Tr}\boldsymbol{W}_{O\mathfrak{h}}^\ell(0)^\top \left[\hat{\boldsymbol{\chi}}_{O\mathfrak{h}s}^\ell(\boldsymbol{x},t)\boldsymbol{v}_{\mathfrak{h}s}^\ell(\boldsymbol{x},t)^\top + \frac{1}{\sqrt{\mathcal{H}}}\boldsymbol{g}_{\mathfrak{s}}^{\ell+1}(\boldsymbol{x},t)\hat{\boldsymbol{\xi}}_{O\mathfrak{h}s}^\ell(\boldsymbol{x},t)^\top\right]\right)$$

$$= -\frac{1}{2} \sum_{tt'ss'} \int d\boldsymbol{x}d\boldsymbol{x}' \hat{\boldsymbol{\chi}}_{O\mathfrak{h}s}^\ell(\boldsymbol{x},t) \cdot \hat{\boldsymbol{\chi}}_{O\mathfrak{h}s'}^\ell(\boldsymbol{x}',t')V_{\mathfrak{h}ss'}^\ell(\boldsymbol{x},\boldsymbol{x}',t,t')$$

$$= -\frac{1}{2} \sum_{tt'ss'} \int d\boldsymbol{x}d\boldsymbol{x}' \hat{\boldsymbol{\xi}}_{O\mathfrak{h}s}^\ell(\boldsymbol{x},t) \cdot \hat{\boldsymbol{\xi}}_{O\mathfrak{h}s'}^\ell(\boldsymbol{x}',t')G_{ss'}^{\ell+1}(\boldsymbol{x},\boldsymbol{x}',t,t')$$

$$- i \sum_{tt'ss'} \int d\boldsymbol{x}d\boldsymbol{x}' \left[\hat{\boldsymbol{\xi}}_{O\mathfrak{h}s}^\ell(\boldsymbol{x},t) \cdot \boldsymbol{v}_{\mathfrak{h}s}^\ell(\boldsymbol{x},t)R_{\mathfrak{h}ss'}^{g^{\ell+1},\chi_O^\ell}(\boldsymbol{x},\boldsymbol{x}',t,t')\right] \quad (49)$$

where we introduced the response function

$$R_{\mathfrak{h}ss'}^{g^{\ell+1},\chi_O^\ell}(\boldsymbol{x},\boldsymbol{x}',t,t') \equiv -\frac{i}{N\sqrt{\mathcal{H}}}\boldsymbol{g}_{\mathfrak{s}}^{\ell+1}(\boldsymbol{x},t) \cdot \hat{\boldsymbol{\chi}}_{O\mathfrak{h}s'}^\ell(\boldsymbol{x}',t'). \quad (50)$$

**Value Matrices** Next, we average over the value matrices $\boldsymbol{W}_{V\mathfrak{h}}^\ell(0)$ which gives

$$\ln \mathbb{E}_{\boldsymbol{W}_{V\mathfrak{h}}^\ell(0)} \exp\left(-\frac{i}{\sqrt{N}} \sum_{ts} \int d\boldsymbol{x} \, \mathrm{Tr}\boldsymbol{W}_{V\mathfrak{h}}^\ell(0)^\top \left[\frac{1}{\sqrt{\mathcal{H}}}\hat{\boldsymbol{\chi}}_{V\mathfrak{h}s}^\ell(\boldsymbol{x},t)\boldsymbol{h}_{\mathfrak{s}}^\ell(\boldsymbol{x},t)^\top + \boldsymbol{g}_{O\mathfrak{h}s}^\ell(\boldsymbol{x},t)\hat{\boldsymbol{\xi}}_{V\mathfrak{h}s}^\ell(\boldsymbol{x},t)^\top\right]\right)$$

$$= -\frac{1}{2} \sum_{tt'ss'} \int d\boldsymbol{x}d\boldsymbol{x}' \left[\hat{\boldsymbol{\chi}}_{V\mathfrak{h}s}^\ell(\boldsymbol{x},t) \cdot \hat{\boldsymbol{\chi}}_{V\mathfrak{h}s}^\ell(\boldsymbol{x},t)H_{ss'}^\ell(\boldsymbol{x},\boldsymbol{x}',t,t') + \hat{\boldsymbol{\xi}}_{V\mathfrak{h}s}^\ell(\boldsymbol{x},t) \cdot \hat{\boldsymbol{\xi}}_{V\mathfrak{h}s'}^\ell(\boldsymbol{x}',t')G_{O\mathfrak{h}ss'}^\ell(\boldsymbol{x},\boldsymbol{x}',t,t')\right]$$

$$- i \sum_{tt'ss'} \int d\boldsymbol{x}d\boldsymbol{x}' \left[\hat{\boldsymbol{\chi}}_{V\mathfrak{h}s}^\ell(\boldsymbol{x},t) \cdot \boldsymbol{g}_{O\mathfrak{h}s'}^\ell(\boldsymbol{x}',t')R_{\mathfrak{h}ss'}^{h^\ell,\xi_V^\ell}(\boldsymbol{x},\boldsymbol{x}',t,t')\right]$$

$$R_{\mathfrak{h}ss'}^{h^\ell,\xi_V^\ell}(\boldsymbol{x},\boldsymbol{x}',t,t') = -\frac{i}{N\sqrt{\mathcal{H}}}\boldsymbol{h}_{\mathfrak{s}}^\ell(\boldsymbol{x},t) \cdot \hat{\boldsymbol{\xi}}_{V\mathfrak{h}s'}^\ell(\boldsymbol{x}',t') \quad (51)$$

**Key/Query Matrices**   Next, we need to perform the averages involving $\boldsymbol{W}_{K\mathfrak{h}}(0)$ which has entries resulting in

$$
\ln \mathbb{E}_{\boldsymbol{W}_{K\mathfrak{h}}^{\ell}(0)} \exp\left( -\frac{i}{\sqrt{N}} \sum_{ts} \int d\boldsymbol{x}\, \mathrm{Tr}\, \boldsymbol{W}_{K\mathfrak{h}}^{\ell}(0)^{\top} \left[ \frac{1}{\sqrt{\mathcal{H}}} \hat{\boldsymbol{\chi}}_{K\mathfrak{h}s}^{\ell}(\boldsymbol{x},t) \boldsymbol{h}_{\mathfrak{s}}^{\ell}(\boldsymbol{x},t)^{\top} + \boldsymbol{q}_{\mathfrak{h}s}^{\ell}(\boldsymbol{x},t) \hat{\boldsymbol{\xi}}_{K\mathfrak{h}s}^{\ell}(\boldsymbol{x},t)^{\top} \right] \right)
$$

$$
= -\frac{1}{2} \sum_{tt'\mathfrak{s}s'} \int d\boldsymbol{x} d\boldsymbol{x}' \left[ \hat{\boldsymbol{\chi}}_{K\mathfrak{h}s}^{\ell}(\boldsymbol{x},t) \cdot \hat{\boldsymbol{\chi}}_{K\mathfrak{h}s}^{\ell}(\boldsymbol{x},t) H_{\mathfrak{s}s'}^{\ell}(\boldsymbol{x},\boldsymbol{x}',t,t') + \hat{\boldsymbol{\xi}}_{K\mathfrak{h}s}^{\ell}(\boldsymbol{x},t) \cdot \hat{\boldsymbol{\xi}}_{K\mathfrak{h}s'}^{\ell}(\boldsymbol{x}',t') Q_{\mathfrak{h}\mathfrak{s}s'}^{\ell}(\boldsymbol{x},\boldsymbol{x}',t,t') \right]
$$

$$
- i \sum_{tt'\mathfrak{s}s'} \int d\boldsymbol{x} d\boldsymbol{x}' \left[ \hat{\boldsymbol{\chi}}_{K\mathfrak{h}s}^{\ell}(\boldsymbol{x},t) \cdot \boldsymbol{q}_{\mathfrak{h}s'}^{\ell}(\boldsymbol{x}',t') R_{\mathfrak{h}\mathfrak{s}s'}^{h^{\ell},\xi_K^{\ell}}(\boldsymbol{x},\boldsymbol{x}',t,t') \right]
$$

$$
R_{\mathfrak{h}\mathfrak{s}s'}^{h^{\ell},\xi_K^{\ell}}(\boldsymbol{x},\boldsymbol{x}',t,t') \equiv -\frac{i}{N\sqrt{\mathcal{H}}} \boldsymbol{h}_{\mathfrak{s}}^{\ell}(\boldsymbol{x},t) \cdot \hat{\boldsymbol{\xi}}_{K\mathfrak{h}s'}^{\ell}(\boldsymbol{x}',t') \tag{52}
$$

We follow an identical procedure for the query matrices $\boldsymbol{W}_{Q}^{\ell}(0)$.

$$
\ln \mathbb{E}_{\boldsymbol{W}_{Q\mathfrak{h}}^{\ell}(0)} \exp\left( -\frac{i}{\sqrt{N}} \sum_{ts} \int d\boldsymbol{x}\, \mathrm{Tr}\, \boldsymbol{W}_{Q\mathfrak{h}}^{\ell}(0)^{\top} \left[ \frac{1}{\sqrt{\mathcal{H}}} \hat{\boldsymbol{\chi}}_{Q\mathfrak{h}s}^{\ell}(\boldsymbol{x},t) \boldsymbol{h}_{\mathfrak{s}}^{\ell}(\boldsymbol{x},t)^{\top} + \boldsymbol{k}_{\mathfrak{h}s}^{\ell}(\boldsymbol{x},t) \hat{\boldsymbol{\xi}}_{Q\mathfrak{h}s}^{\ell}(\boldsymbol{x},t)^{\top} \right] \right)
$$

$$
= -\frac{1}{2} \sum_{tt'\mathfrak{s}s'} \int d\boldsymbol{x} d\boldsymbol{x}' \left[ \hat{\boldsymbol{\chi}}_{Q\mathfrak{h}s}^{\ell}(\boldsymbol{x},t) \cdot \hat{\boldsymbol{\chi}}_{Q\mathfrak{h}s}^{\ell}(\boldsymbol{x},t) H_{\mathfrak{s}s'}^{\ell}(\boldsymbol{x},\boldsymbol{x}',t,t') + \hat{\boldsymbol{\xi}}_{Q\mathfrak{h}s}^{\ell}(\boldsymbol{x},t) \cdot \hat{\boldsymbol{\xi}}_{Q\mathfrak{h}s'}^{\ell}(\boldsymbol{x}',t') K_{\mathfrak{h}\mathfrak{s}s'}^{\ell}(\boldsymbol{x},\boldsymbol{x}',t,t') \right]
$$

$$
- i \sum_{\mathfrak{h}tt'\mathfrak{s}s'} \int d\boldsymbol{x} d\boldsymbol{x}' \left[ \hat{\boldsymbol{\chi}}_{Q\mathfrak{h}s}^{\ell}(\boldsymbol{x},t) \cdot \boldsymbol{k}_{\mathfrak{h}s'}^{\ell}(\boldsymbol{x}',t') R_{\mathfrak{h}\mathfrak{s}s'}^{h^{\ell},\xi_Q^{\ell}}(\boldsymbol{x},\boldsymbol{x}',t,t') \right]
$$

$$
R_{\mathfrak{h}\mathfrak{s}s'}^{h^{\ell},\xi_Q^{\ell}}(\boldsymbol{x},\boldsymbol{x}',t,t') \equiv -\frac{i}{N\sqrt{\mathcal{H}}} \boldsymbol{h}_{\mathfrak{s}}^{\ell}(\boldsymbol{x},t) \cdot \hat{\boldsymbol{\xi}}_{Q\mathfrak{h}s'}^{\ell}(\boldsymbol{x}',t') \tag{53}
$$

**Enforce Kernel Definitions**   After this step, we can introduce new resolutions of the identity for each of the kernels that appeared in the above computation

$$
1 = \int \frac{dH_{\mathfrak{s}s'}^{\ell}(\boldsymbol{x},\boldsymbol{x}',t,t') d\hat{H}_{\mathfrak{s}s'}^{\ell}(\boldsymbol{x},\boldsymbol{x}',t,t')}{2\pi i N^{-1}\mathcal{H}^{-1}}
$$

$$
\exp\left( \hat{H}_{\mathfrak{s}s'}^{\ell}(\boldsymbol{x},\boldsymbol{x}',t,t') \left[ N\mathcal{H} H_{\mathfrak{s}s'}^{\ell}(\boldsymbol{x},\boldsymbol{x}',t,t') - \boldsymbol{h}_{\mathfrak{s}}^{\ell}(\boldsymbol{x},t) \cdot \boldsymbol{h}_{\mathfrak{s}'}^{\ell}(\boldsymbol{x}',t') \right] \right)
$$

$$
1 = \int \frac{dG_{\mathfrak{s}s'}^{\ell}(\boldsymbol{x},\boldsymbol{x}',t,t') d\hat{G}_{\mathfrak{s}s'}^{\ell}(\boldsymbol{x},\boldsymbol{x}',t,t')}{2\pi i N^{-1}\mathcal{H}^{-1}}
$$

$$
\exp\left( \hat{G}_{\mathfrak{s}s'}^{\ell}(\boldsymbol{x},\boldsymbol{x}',t,t') \left[ N\mathcal{H} G_{\mathfrak{s}s'}^{\ell}(\boldsymbol{x},\boldsymbol{x}',t,t') - \boldsymbol{g}_{\mathfrak{s}}^{\ell}(\boldsymbol{x},t) \cdot \boldsymbol{g}_{\mathfrak{s}'}^{\ell}(\boldsymbol{x}',t') \right] \right) \tag{54}
$$

This is repeated for all of the response functions which involve sums over $N\mathcal{H}$ variables

$$1 = \int \frac{dR_{\mathfrak{h}ss'}^{g^{\ell+1},\chi_O^\ell}(\boldsymbol{x},\boldsymbol{x}',t,t')dR_{s's}^{v^\ell,\xi_O^\ell}(\boldsymbol{x}',\boldsymbol{x},t',t)}{2\pi i N^{-1}}$$

$$\exp\left(-R_{\mathfrak{h}s's}^{v^\ell,\xi_O^\ell}(\boldsymbol{x}',\boldsymbol{x},t',t)\left[NR_{\mathfrak{h}ss'}^{g^{\ell+1},\xi_O^\ell}(\boldsymbol{x},\boldsymbol{x}',t,t') + \frac{i}{\sqrt{\mathcal{H}}}g_s^{\ell+1}(\boldsymbol{x},t)\cdot\hat{\boldsymbol{\chi}}_{O\mathfrak{h}s'}^\ell(\boldsymbol{x}',t')\right]\right)$$

$$1 = \int \frac{dR_{\mathfrak{h}ss'}^{h^\ell\xi_V^\ell}(\boldsymbol{x},\boldsymbol{x}',t,t')dR_{\mathfrak{h}s's}^{g_O^\ell\chi_V^\ell}(\boldsymbol{x}',\boldsymbol{x},t',t)}{2\pi i N^{-1}}$$

$$\exp\left(-R_{\mathfrak{h}s's}^{g_O^\ell\chi_V^\ell}(\boldsymbol{x}',\boldsymbol{x},t',t)\left[NR_{\mathfrak{h}ss'}^{h^\ell\xi_V^\ell}(\boldsymbol{x},\boldsymbol{x}',t,t') + \frac{i}{\sqrt{\mathcal{H}}}h_s^\ell(\boldsymbol{x},t)\cdot\hat{\boldsymbol{\xi}}_{V\mathfrak{h}s'}^\ell(\boldsymbol{x}',t')\right]\right)$$

$$1 = \int \frac{dR_{\mathfrak{h}ss'}^{h^\ell\xi_K^\ell}(\boldsymbol{x},\boldsymbol{x}',t,t')dR_{\mathfrak{h}s's}^{q^\ell\chi_K^\ell}(\boldsymbol{x}',\boldsymbol{x},t',t)}{2\pi i N^{-1}}$$

$$\exp\left(-R_{\mathfrak{h}s's}^{q^\ell\chi_K^\ell}(\boldsymbol{x}',\boldsymbol{x},t',t)\left[NR_{\mathfrak{h}ss'}^{h^\ell\xi_K^\ell}(\boldsymbol{x},\boldsymbol{x}',t,t') + \frac{i}{\sqrt{\mathcal{H}}}h_s^\ell(\boldsymbol{x},t)\cdot\hat{\boldsymbol{\xi}}_{K\mathfrak{h}s'}^\ell(\boldsymbol{x}',t')\right]\right)$$

$$1 = \int \frac{dR_{\mathfrak{h}ss'}^{h^\ell\xi_Q^\ell}(\boldsymbol{x},\boldsymbol{x}',t,t')dR_{\mathfrak{h}s's}^{k^\ell\chi_Q^\ell}(\boldsymbol{x}',\boldsymbol{x},t',t)}{2\pi i N^{-1}}$$

$$\exp\left(-R_{\mathfrak{h}s's}^{k^\ell\chi_Q^\ell}(\boldsymbol{x}',\boldsymbol{x},t',t)\left[NR_{\mathfrak{h}ss'}^{h^\ell\xi_Q^\ell}(\boldsymbol{x},\boldsymbol{x}',t,t') + \frac{i}{\sqrt{\mathcal{H}}}h_s^\ell(\boldsymbol{x},t)\cdot\hat{\boldsymbol{\xi}}_{Q\mathfrak{h}s'}^\ell(\boldsymbol{x}',t')\right]\right)$$

There are other kernels which are only relevant within a single head including $\{Q_\mathfrak{h}^\ell, K_\mathfrak{h}^\ell, A_\mathfrak{h}^\ell, M_\mathfrak{h}^\ell\}$. These

$$1 = \int \frac{dQ_{\mathfrak{h}ss'}^\ell(\boldsymbol{x},\boldsymbol{x}',t,t')d\hat{Q}_{\mathfrak{h}ss'}^\ell(\boldsymbol{x},\boldsymbol{x}',t,t')}{2\pi i N^{-1}}$$

$$\exp\left(\hat{Q}_{\mathfrak{h}ss'}^\ell(\boldsymbol{x},\boldsymbol{x}',t,t')\left[NQ_{\mathfrak{h}ss'}^\ell(\boldsymbol{x},\boldsymbol{x}',t,t') - \boldsymbol{q}_{\mathfrak{h}s}^\ell(\boldsymbol{x},t)\cdot\boldsymbol{q}_{\mathfrak{h}s'}^\ell(\boldsymbol{x}',t')\right]\right)$$

$$1 = \int \frac{dK_{\mathfrak{h}ss'}^\ell(\boldsymbol{x},\boldsymbol{x}',t,t')d\hat{K}_{\mathfrak{h}ss'}^\ell(\boldsymbol{x},\boldsymbol{x}',t,t')}{2\pi i N^{-1}}$$

$$\exp\left(\hat{K}_{\mathfrak{h}ss'}^\ell(\boldsymbol{x},\boldsymbol{x}',t,t')\left[NK_{\mathfrak{h}ss'}^\ell(\boldsymbol{x},\boldsymbol{x}',t,t') - \boldsymbol{k}_{\mathfrak{h}s}^\ell(\boldsymbol{x},t)\cdot\boldsymbol{k}_{\mathfrak{h}s'}^\ell(\boldsymbol{x}',t')\right]\right)$$

$$1 = \int \frac{dM_{\mathfrak{h}ss'}^\ell(\boldsymbol{x},t)d\hat{M}_{\mathfrak{h}ss'}^\ell(\boldsymbol{x},t)}{2\pi i N^{-1}}$$

$$\exp\left(\hat{M}_{\mathfrak{h}ss'}^\ell(\boldsymbol{x},t)\left[NM_{\mathfrak{h}ss'}^\ell(\boldsymbol{x},t) - \frac{1}{\sqrt{\mathcal{H}}}\tilde{\boldsymbol{g}}_s^\ell(\boldsymbol{x},t)\cdot\boldsymbol{o}_{\mathfrak{h}s'}^\ell(\boldsymbol{x},t)\right]\right)$$

$$1 = \int \frac{d\mathcal{A}_{\mathfrak{h}ss'}^\ell(\boldsymbol{x},t)d\hat{\mathcal{A}}_{\mathfrak{h}ss'}^\ell(\boldsymbol{x},t)}{2\pi i N^{-1}}$$

$$\exp\left(\hat{\mathcal{A}}_{\mathfrak{h}ss'}^\ell(\boldsymbol{x},t)\left[N\mathcal{A}_{\mathfrak{h}ss'}^\ell(\boldsymbol{x},t) - \boldsymbol{k}_{\mathfrak{h}s}^\ell(\boldsymbol{x},t)\cdot\boldsymbol{q}_{\mathfrak{h}s'}^\ell(\boldsymbol{x},t)\right]\right)$$

We now combine all of the order parameters into a large collection $\boldsymbol{Q}$ which are vectorized over all layer, time, spatial and sample indices

$$\boldsymbol{Q} = \text{Vec}\{H^\ell, G^\ell, V^\ell, K^\ell, Q^\ell\}$$
$$\cup\{\hat{H}^\ell, \hat{G}^\ell, \hat{V}^\ell, \hat{K}^\ell, \hat{Q}^\ell\}$$
$$\cup\{R^{v^\ell\xi_O^\ell}, R^{g^{\ell+1},\chi_O^\ell}, R^{k^\ell\chi_Q^\ell}, R^{q^\ell,\chi_K^\ell}\}$$
$$\cup\{R^{h^\ell,\xi_V^\ell}, R^{h^\ell,\xi_K^\ell}, R^{h^\ell,\xi_Q^\ell}, R^{h^\ell,\xi_V^\ell}\}. \tag{55}$$

After introducing this collection of order parameters, our original MGF satisfies a large deviation principle with

$$Z \propto \int d\boldsymbol{Q}\ \exp\left(N\mathcal{H}L\,S(\boldsymbol{Q})\right) \tag{56}$$

where the DMFT action $S(\boldsymbol{Q})$ has the form

$$
\begin{aligned}
S = {} & \frac{1}{L}\sum_{\ell=1}^{L}\sum_{tt'\mathfrak{s}\mathfrak{s}'}\int d\boldsymbol{x}d\boldsymbol{x}'[H_{\mathfrak{s}\mathfrak{s}'}^{\ell}(\boldsymbol{x},\boldsymbol{x}',t,t')\hat{H}_{\mathfrak{s}\mathfrak{s}'}^{\ell}(\boldsymbol{x},\boldsymbol{x}',t,t') + G_{\mathfrak{s}\mathfrak{s}'}^{\ell}(\boldsymbol{x},\boldsymbol{x}',t,t')\hat{G}_{\mathfrak{s}\mathfrak{s}'}^{\ell}(\boldsymbol{x},\boldsymbol{x}',t,t')] \\
& + \frac{1}{\mathcal{H}L}\sum_{\mathfrak{h}=1}^{\mathcal{H}}\sum_{\ell=1}^{L}\sum_{tt'\mathfrak{s}\mathfrak{s}'}\int d\boldsymbol{x}d\boldsymbol{x}'\left[\hat{Q}_{\mathfrak{h}\mathfrak{s}\mathfrak{s}'}^{\ell}(\boldsymbol{x},\boldsymbol{x}',t,t')Q_{\mathfrak{h}\mathfrak{s}\mathfrak{s}'}^{\ell}(\boldsymbol{x},\boldsymbol{x}',t,t') + \hat{K}_{\mathfrak{h}\mathfrak{s}\mathfrak{s}'}^{\ell}(\boldsymbol{x},\boldsymbol{x}',t,t')K_{\mathfrak{h}\mathfrak{s}\mathfrak{s}'}^{\ell}(\boldsymbol{x},\boldsymbol{x}',t,t')\right] \\
& + \frac{1}{\mathcal{H}L}\sum_{\mathfrak{h}=1}^{\mathcal{H}}\sum_{\ell=1}^{L}\sum_{tt'\mathfrak{s}\mathfrak{s}'}\int d\boldsymbol{x}d\boldsymbol{x}'\left[\hat{V}_{\mathfrak{h}\mathfrak{s}\mathfrak{s}'}^{\ell}(\boldsymbol{x},\boldsymbol{x}',t,t')V_{\mathfrak{h}\mathfrak{s}\mathfrak{s}'}^{\ell}(\boldsymbol{x},\boldsymbol{x}',t,t')\right] \\
& + \frac{1}{\mathcal{H}L}\sum_{\mathfrak{h}=1}^{\mathcal{H}}\sum_{\ell=1}^{L}\sum_{t\mathfrak{s}\mathfrak{s}'}\int d\boldsymbol{x}\left[\hat{\mathcal{A}}_{\mathfrak{h}\mathfrak{s}\mathfrak{s}'}^{\ell}(\boldsymbol{x},t)\mathcal{A}_{\mathfrak{h}\mathfrak{s}\mathfrak{s}'}^{\ell}(\boldsymbol{x},t) + \hat{M}_{\mathfrak{h}\mathfrak{s}\mathfrak{s}'}^{\ell}(\boldsymbol{x},t)M_{\mathfrak{h}\mathfrak{s}\mathfrak{s}'}^{\ell}(\boldsymbol{x},t)\right] \\
& - \frac{1}{\mathcal{H}L}\sum_{\mathfrak{h}=1}^{\mathcal{H}}\sum_{\ell=1}^{L}\sum_{tt'\mathfrak{s}\mathfrak{s}'}\int d\boldsymbol{x}d\boldsymbol{x}'\left[R_{\mathfrak{h}\mathfrak{s}'\mathfrak{s}}^{v^{\ell},\xi_{O}^{\ell}}(\boldsymbol{x}',\boldsymbol{x},t',t)R_{\mathfrak{h}\mathfrak{s}\mathfrak{s}'}^{g^{\ell+1},\xi_{O}^{\ell}}(\boldsymbol{x},\boldsymbol{x}',t,t') + R_{\mathfrak{h}\mathfrak{s}'\mathfrak{s}}^{g_{O}^{\ell}\chi_{V}^{\ell}}(\boldsymbol{x}',\boldsymbol{x},t',t)R_{\mathfrak{h}\mathfrak{s}\mathfrak{s}'}^{h^{\ell}\xi_{V}^{\ell}}(\boldsymbol{x},\boldsymbol{x}',t,t')\right] \\
& - \frac{1}{\mathcal{H}L}\sum_{\mathfrak{h}=1}^{\mathcal{H}}\sum_{\ell=1}^{L}\sum_{tt'\mathfrak{s}\mathfrak{s}'}\int d\boldsymbol{x}d\boldsymbol{x}'\left[R_{\mathfrak{h}\mathfrak{s}'\mathfrak{s}}^{q^{\ell}\chi_{K}^{\ell}}(\boldsymbol{x}',\boldsymbol{x},t',t)R_{\mathfrak{h}\mathfrak{s}\mathfrak{s}'}^{h^{\ell}\xi_{K}^{\ell}}(\boldsymbol{x},\boldsymbol{x}',t,t') + R_{\mathfrak{h}\mathfrak{s}'\mathfrak{s}}^{k^{\ell}\chi_{Q}^{\ell}}(\boldsymbol{x}',\boldsymbol{x},t',t)R_{\mathfrak{h}\mathfrak{s}\mathfrak{s}'}^{h^{\ell}\xi_{Q}^{\ell}}(\boldsymbol{x},\boldsymbol{x}',t,t')\right] \\
& + \frac{1}{L}\ln\mathcal{Z}_{\text{res}} + \frac{1}{L\mathcal{H}}\sum_{\mathfrak{h}=1}^{\mathcal{H}}\sum_{\ell=1}^{L}\ln\mathcal{Z}_{\text{MHSA},\mathfrak{h}}^{\ell} \qquad\qquad (57)
\end{aligned}
$$

The residual stream single site moment generating functional has the form

$$
\mathcal{Z}_{\text{res}} = \int \prod_{\ell \mathfrak{h} s \boldsymbol{x} t} \frac{d\hat{\chi}_{O\mathfrak{h}s}(\boldsymbol{x},t)d\chi_{O\mathfrak{h}s}(\boldsymbol{x},t)}{2\pi} \frac{d\hat{\xi}_{V\mathfrak{h}s}(\boldsymbol{x},t)d\xi_{V\mathfrak{h}s}(\boldsymbol{x},t)}{2\pi} \frac{d\hat{\xi}_{K\mathfrak{h}s}(\boldsymbol{x},t)d\xi_{K\mathfrak{h}s}(\boldsymbol{x},t)}{2\pi} \frac{d\hat{\xi}_{Q\mathfrak{h}s}(\boldsymbol{x},t)d\xi_{Q\mathfrak{h}s}(\boldsymbol{x},t)}{2\pi}
$$

$$
\exp\left( -\sum_{\ell=1}^{L} \sum_{tt'ss'} \int d\boldsymbol{x}d\boldsymbol{x}' \hat{H}_{ss'}^{\ell}(\boldsymbol{x},\boldsymbol{x}',t,t') h_s^{\ell}(\boldsymbol{x},t) h_{s'}^{\ell}(\boldsymbol{x}',t') \right)
$$

$$
\exp\left( -\sum_{\ell=1}^{L} \sum_{tt'ss'} \int d\boldsymbol{x}d\boldsymbol{x}' \hat{G}_{ss'}^{\ell}(\boldsymbol{x},\boldsymbol{x}',t,t') g_s^{\ell}(\boldsymbol{x},t) g_{s'}^{\ell}(\boldsymbol{x}',t') \right)
$$

$$
\exp\left( -\sum_{\ell=1}^{L} \sum_{\mathfrak{h}} \sum_{tss'} \int d\boldsymbol{x} \hat{M}_{\mathfrak{h}ss'}^{\ell}(\boldsymbol{x},t) g_s^{\ell+1}(\boldsymbol{x},t) o_{\mathfrak{h}s'}^{\ell}(\boldsymbol{x},t) \right)
$$

$$
\exp\left( -\frac{1}{2} \sum_{\ell\mathfrak{h}} \sum_{tt'ss'} \int d\boldsymbol{x}d\boldsymbol{x}' \hat{\chi}_{O\mathfrak{h}s}^{\ell}(\boldsymbol{x},t) \hat{\chi}_{O\mathfrak{h}s'}^{\ell}(\boldsymbol{x}',t') V_{\mathfrak{h}ss'}^{\ell}(\boldsymbol{x},\boldsymbol{x}',t,t') \right)
$$

$$
\exp\left( -\frac{1}{2} \sum_{\ell\mathfrak{h}} \sum_{tt'ss'} \int d\boldsymbol{x}d\boldsymbol{x}' \hat{\xi}_{V\mathfrak{h}s}^{\ell}(\boldsymbol{x},t) \hat{\xi}_{V\mathfrak{h}s'}^{\ell}(\boldsymbol{x}',t') G_{O\mathfrak{h}ss'}^{\ell}(\boldsymbol{x},\boldsymbol{x}',t,t') \right)
$$

$$
\exp\left( -\frac{1}{2} \sum_{\ell\mathfrak{h}} \sum_{tt'ss'} \int d\boldsymbol{x}d\boldsymbol{x}' \hat{\xi}_{K\mathfrak{h}s}^{\ell}(\boldsymbol{x},t) \hat{\xi}_{K\mathfrak{h}s'}^{\ell}(\boldsymbol{x}',t') Q_{\mathfrak{h}ss'}^{\ell}(\boldsymbol{x},\boldsymbol{x}',t,t') \right)
$$

$$
\exp\left( -\frac{1}{2} \sum_{\ell\mathfrak{h}} \sum_{tt'ss'} \int d\boldsymbol{x}d\boldsymbol{x}' \hat{\xi}_{Q\mathfrak{h}s}^{\ell}(\boldsymbol{x},t) \hat{\xi}_{Q\mathfrak{h}s'}^{\ell}(\boldsymbol{x}',t') K_{\mathfrak{h}ss'}^{\ell}(\boldsymbol{x},\boldsymbol{x}',t,t') \right)
$$

$$
\exp\left( i \sum_{\ell\mathfrak{h}} \sum_{ts} \int d\boldsymbol{x} \hat{\chi}_{O\mathfrak{h}s}^{\ell}(\boldsymbol{x},t) \chi_{O\mathfrak{h}s}^{\ell}(\boldsymbol{x},t) + \hat{\xi}_{V\mathfrak{h}s}^{\ell}(\boldsymbol{x},t) \xi_{V\mathfrak{h}s}^{\ell}(\boldsymbol{x},t) \right)
$$

$$
\exp\left( i \sum_{\ell\mathfrak{h}} \sum_{ts} \int d\boldsymbol{x} \hat{\xi}_{Q\mathfrak{h}s}^{\ell}(\boldsymbol{x},t) \xi_{Q\mathfrak{h}s}^{\ell}(\boldsymbol{x},t) + \hat{\xi}_{K\mathfrak{h}s}^{\ell}(\boldsymbol{x},t) \xi_{K\mathfrak{h}s}^{\ell}(\boldsymbol{x},t) \right)
$$

$$
\exp\left( -\frac{i}{\sqrt{\mathcal{H}}} \sum_{\ell\mathfrak{h}} \sum_{tt'ss'} \int d\boldsymbol{x}d\boldsymbol{x}' R_{\mathfrak{h}ss'}^{v^{\ell},\xi_O^{\ell}}(\boldsymbol{x},\boldsymbol{x}',t,t') \hat{\chi}_{O\mathfrak{h}s}^{\ell}(\boldsymbol{x},t) g_{s'}^{\ell+1}(\boldsymbol{x}',t') \right)
$$

$$
\exp\left( -\frac{i}{\sqrt{\mathcal{H}}} \sum_{\ell\mathfrak{h}} \sum_{tt'ss'} \int d\boldsymbol{x}d\boldsymbol{x}' R_{\mathfrak{h}ss'}^{g_O^{\ell}\chi_V^{\ell}}(\boldsymbol{x},\boldsymbol{x}',t,t') \hat{\xi}_{V\mathfrak{h}s}^{\ell}(\boldsymbol{x},t) h_{s'}^{\ell}(\boldsymbol{x}',t') \right)
$$

$$
\exp\left( -\frac{i}{\sqrt{\mathcal{H}}} \sum_{\ell\mathfrak{h}} \sum_{tt'ss'} \int d\boldsymbol{x}d\boldsymbol{x}' R_{\mathfrak{h}ss'}^{q^{\ell}\chi_K^{\ell}}(\boldsymbol{x},\boldsymbol{x}',t,t') \hat{\xi}_{K\mathfrak{h}s}^{\ell}(\boldsymbol{x},t) h_{s'}^{\ell}(\boldsymbol{x}',t') \right)
$$

$$
\exp\left( -\frac{i}{\sqrt{\mathcal{H}}} \sum_{\ell\mathfrak{h}} \sum_{tt'ss'} \int d\boldsymbol{x}d\boldsymbol{x}' R_{\mathfrak{h}ss'}^{k^{\ell}\chi_Q^{\ell}}(\boldsymbol{x},\boldsymbol{x}',t,t') \hat{\xi}_{Q\mathfrak{h}s}^{\ell}(\boldsymbol{x},t) h_{s'}^{\ell}(\boldsymbol{x}',t') \right) \tag{58}
$$

Though this expression is cumbersome, we will show that, since $\hat{H}, \hat{G}$ vanish at their saddle point this MGF merely encodes the following statistical description of the fields of interest such as

$$
\chi_{O\mathfrak{h}s}^{\ell}(\boldsymbol{x},t) = u_{O\mathfrak{h}s}^{\ell}(\boldsymbol{x},t) + \frac{1}{\sqrt{\mathcal{H}}} \sum_{t's'} \int d\boldsymbol{x}' R_{\mathfrak{h}ss'}^{v^{\ell},\xi_O^{\ell}}(\boldsymbol{x},\boldsymbol{x}',t,t') g_{s'}^{\ell+1}(\boldsymbol{x}',t')
$$

$$
u_{O\mathfrak{h}s}^{\ell}(\boldsymbol{x},t) \sim \mathcal{GP}\left( 0, V_{\mathfrak{h}ss'}^{\ell}(\boldsymbol{x},\boldsymbol{x}',t,t') \right) \tag{59}
$$

We note that $h$ only depends on $\bar{\chi}_O^\ell = \frac{1}{\sqrt{\mathcal{H}}} \sum_{\mathfrak{h}=1}^{\mathcal{H}} \chi_{O\mathfrak{h}}^\ell \sigma_{\mathfrak{h}}^\ell$ so it has the form

$$\bar{\chi}_O^\ell = \frac{1}{\sqrt{\mathcal{H}}} \sum_{\mathfrak{h}=1}^{\mathcal{H}} u_{O\mathfrak{h}}^\ell \sigma_{\mathfrak{h}}^\ell + \bar{R}^{v^\ell \xi_O^\ell} g^\ell \;,\;\; \bar{R}^{v^\ell \xi_O^\ell} = \frac{1}{\mathcal{H}} \sum_{\mathfrak{h}=1}^{\mathcal{H}} R_{\mathfrak{h}}^{v^\ell, \xi_O^\ell} \sigma_{\mathfrak{h}}^\ell \tag{60}$$

This fact will be important when we take the large head limit with $N$ fixed in Appendix E.3.

Next, we analyze the single site MGF for the hidden MHSA layers

$$\mathcal{Z}_{\text{MHSA},\mathfrak{h}}^\ell = \int \prod_{\ell \mathfrak{h} s \boldsymbol{x} t} \frac{d\hat{\xi}_{O\mathfrak{h}s}(\boldsymbol{x},t) d\xi_{O\mathfrak{h}s}(\boldsymbol{x},t)}{2\pi} \frac{d\hat{\chi}_{V\mathfrak{h}s}(\boldsymbol{x},t) d\chi_{V\mathfrak{h}s}(\boldsymbol{x},t)}{2\pi} \frac{d\hat{\chi}_{K\mathfrak{h}s}(\boldsymbol{x},t) d\chi_{K\mathfrak{h}s}(\boldsymbol{x},t)}{2\pi} \frac{d\hat{\chi}_{Q\mathfrak{h}s}(\boldsymbol{x},t) d\chi_{Q\mathfrak{h}s}(\boldsymbol{x},t)}{2\pi}$$

$$\exp\left( -\sum_{tt'ss'} \int d\boldsymbol{x}d\boldsymbol{x}' \hat{Q}_{\mathfrak{h}ss'}^\ell(\boldsymbol{x},\boldsymbol{x}',t,t') q_{\mathfrak{h}s}^\ell(\boldsymbol{x},t) q_{\mathfrak{h}s'}^\ell(\boldsymbol{x}',t') \right)$$

$$\exp\left( -\sum_{tt'ss'} \int d\boldsymbol{x}d\boldsymbol{x}' \hat{K}_{\mathfrak{h}ss'}^\ell(\boldsymbol{x},\boldsymbol{x}',t,t') k_{\mathfrak{h}s}^\ell(\boldsymbol{x},t) k_{\mathfrak{h}s'}^\ell(\boldsymbol{x}',t') \right)$$

$$\exp\left( -\sum_{tt'ss'} \int d\boldsymbol{x}d\boldsymbol{x}' \hat{V}_{\mathfrak{h}ss'}^\ell(\boldsymbol{x},\boldsymbol{x}',t,t') v_{\mathfrak{h}s}^\ell(\boldsymbol{x},t) v_{\mathfrak{h}s'}^\ell(\boldsymbol{x}',t') \right)$$

$$\exp\left( -\sum_{tss'} \int d\boldsymbol{x} \hat{A}_{\mathfrak{h}ss'}^\ell(\boldsymbol{x},t) k_{\mathfrak{h}s}^\ell(\boldsymbol{x},t) q_{\mathfrak{h}s'}^\ell(\boldsymbol{x},t) \right)$$

$$\exp\left( -\frac{1}{2} \sum_{tt'ss'} \int d\boldsymbol{x}d\boldsymbol{x}' \hat{\xi}_{O\mathfrak{h}s}^\ell(\boldsymbol{x},t) \hat{\xi}_{O\mathfrak{h}s'}^\ell(\boldsymbol{x}',t') G_{ss'}^{\ell+1}(\boldsymbol{x},\boldsymbol{x}',t,t') \right)$$

$$\exp\left( -\frac{1}{2} \sum_{tt'ss'} \int d\boldsymbol{x}d\boldsymbol{x}' \hat{\chi}_{V\mathfrak{h}s}^\ell(\boldsymbol{x},t) \hat{\chi}_{V\mathfrak{h}s}^\ell(\boldsymbol{x},t) H_{ss'}^\ell(\boldsymbol{x},\boldsymbol{x}',t,t') \right)$$

$$\exp\left( -\frac{1}{2} \sum_{tt'ss'} \int d\boldsymbol{x}d\boldsymbol{x}' \hat{\chi}_{K\mathfrak{h}s}^\ell(\boldsymbol{x},t) \hat{\chi}_{K\mathfrak{h}s}^\ell(\boldsymbol{x},t) H_{ss'}^\ell(\boldsymbol{x},\boldsymbol{x}',t,t') \right)$$

$$\exp\left( -\frac{1}{2} \sum_{tt'ss'} \int d\boldsymbol{x}d\boldsymbol{x}' \hat{\chi}_{Q\mathfrak{h}s}^\ell(\boldsymbol{x},t) \hat{\chi}_{Q\mathfrak{h}s}^\ell(\boldsymbol{x},t) H_{ss'}^\ell(\boldsymbol{x},\boldsymbol{x}',t,t') \right)$$

$$\exp\left( i\sum_{ts} \int d\boldsymbol{x} \; \hat{\xi}_{O\mathfrak{h}s}^\ell(\boldsymbol{x},t) \xi_{O\mathfrak{h}s}^\ell(\boldsymbol{x},t) + \hat{\chi}_{V\mathfrak{h}s}^\ell(\boldsymbol{x},t) \chi_{V\mathfrak{h}s}^\ell(\boldsymbol{x},t) \right)$$

$$\exp\left( i\sum_{ts} \int d\boldsymbol{x} \; \hat{\chi}_{Q\mathfrak{h}s}^\ell(\boldsymbol{x},t) \chi_{Q\mathfrak{h}s}^\ell(\boldsymbol{x},t) + \hat{\chi}_{K\mathfrak{h}s}^\ell(\boldsymbol{x},t) \chi_{K\mathfrak{h}s}^\ell(\boldsymbol{x},t) \right)$$

$$\exp\left( -i\sum_{tt'ss'} \int d\boldsymbol{x}d\boldsymbol{x}' R_{\mathfrak{h}ss'}^{g^{\ell+1}, \chi_O^\ell}(\boldsymbol{x},\boldsymbol{x}',t,t') \hat{\xi}_{O\mathfrak{h}s}^\ell(\boldsymbol{x},t) v_{\mathfrak{h}s'}^\ell(\boldsymbol{x}',t') \right)$$

$$\exp\left( -i\sum_{tt'ss'} \int d\boldsymbol{x}d\boldsymbol{x}' R_{\mathfrak{h}ss'}^{h^\ell, \xi_V^\ell}(\boldsymbol{x},\boldsymbol{x}',t,t') \hat{\chi}_{V\mathfrak{h}s}^\ell(\boldsymbol{x},t) g_{O\mathfrak{h}s'}^\ell(\boldsymbol{x}',t') \right)$$

$$\exp\left( -i\sum_{\mathfrak{h}tt'ss'} \int d\boldsymbol{x}d\boldsymbol{x}' R_{\mathfrak{h}ss'}^{h^\ell, \xi_Q^\ell}(\boldsymbol{x},\boldsymbol{x}',t,t') \hat{\chi}_{Q\mathfrak{h}s}^\ell(\boldsymbol{x},t) k_{\mathfrak{h}s'}^\ell(\boldsymbol{x}',t') \right)$$

$$\exp\left( -i\sum_{\mathfrak{h}tt'ss'} \int d\boldsymbol{x}d\boldsymbol{x}' R_{\mathfrak{h}ss'}^{h^\ell, \xi_K^\ell}(\boldsymbol{x},\boldsymbol{x}',t,t') \hat{\chi}_{K\mathfrak{h}s}^\ell(\boldsymbol{x},t) q_{\mathfrak{h}s'}^\ell(\boldsymbol{x}',t') \right) \tag{61}$$

## E.2 Infinite $N$ (Key/Query dimension) Limit

First, we take the $N \to \infty$ limit with $\mathcal{H}, L$ fixed. This can be obtained with a simple saddle point procedure using the action in the form written in the previous section. This calculation exactly mimics prior works [15, 11] where all of the order parameters $\boldsymbol{Q}$ take on their values at the saddle point $\boldsymbol{Q}^\star$.

$$\frac{\partial S(\boldsymbol{Q})}{\partial \boldsymbol{Q}}\Big|_{\boldsymbol{Q}^\star} = 0 \tag{62}$$

**Saddle Point Values for Order Parameters** Under this saddle point, all of the order parameters presented will concentrate and all neurons will become statistically independent. The governing equations for the order parameters $\boldsymbol{Q}^\star$ are given in terms of averages $\langle \cdot \rangle$ over the single site densities defined by the moment generating functionals $\mathcal{Z}$ and have the form

$$H^\ell_{\mathfrak{s}\mathfrak{s}'}(\boldsymbol{x}, \boldsymbol{x}', t, t') = \left\langle h^\ell_{\mathfrak{s}}(\boldsymbol{x}, t) h^\ell_{\mathfrak{s}'}(\boldsymbol{x}', t') \right\rangle \;,\; G^\ell_{\mathfrak{s}\mathfrak{s}'}(\boldsymbol{x}, \boldsymbol{x}', t, t') = \left\langle g^\ell_{\mathfrak{s}}(\boldsymbol{x}, t) g^\ell_{\mathfrak{s}'}(\boldsymbol{x}', t') \right\rangle$$

$$M^\ell_{\mathfrak{h}\mathfrak{s}\mathfrak{s}'}(\boldsymbol{x}, t) = \left\langle g^{\ell+1}_{\mathfrak{s}}(\boldsymbol{x}, t) o^\ell_{\mathfrak{h}\mathfrak{s}'}(\boldsymbol{x}, t) \right\rangle \;,\; V^\ell_{\mathfrak{h}\mathfrak{s}\mathfrak{s}'}(\boldsymbol{x}, \boldsymbol{x}', t, t') = \left\langle v^\ell_{\mathfrak{h}\mathfrak{s}}(\boldsymbol{x}, t) v^\ell_{\mathfrak{h}\mathfrak{s}'}(\boldsymbol{x}', t') \right\rangle$$

$$Q^\ell_{\mathfrak{h}\mathfrak{s}\mathfrak{s}'}(\boldsymbol{x}, \boldsymbol{x}', t, t') = \left\langle q^\ell_{\mathfrak{h}\mathfrak{s}}(\boldsymbol{x}, t) q^\ell_{\mathfrak{h}\mathfrak{s}'}(\boldsymbol{x}', t') \right\rangle \;,\; K^\ell_{\mathfrak{h}\mathfrak{s}\mathfrak{s}'}(\boldsymbol{x}, \boldsymbol{x}', t, t') = \left\langle k^\ell_{\mathfrak{h}\mathfrak{s}}(\boldsymbol{x}, t) k^\ell_{\mathfrak{h}\mathfrak{s}'}(\boldsymbol{x}', t') \right\rangle$$

$$\mathcal{A}^\ell_{\mathfrak{h}\mathfrak{s}\mathfrak{s}'}(\boldsymbol{x}, t) = \left\langle k^\ell_{\mathfrak{h}\mathfrak{s}}(\boldsymbol{x}, t) q^\ell_{\mathfrak{h}\mathfrak{s}'}(\boldsymbol{x}, t) \right\rangle$$

$$R^{g^{\ell+1}, \chi^\ell_O}_{\mathfrak{h}\mathfrak{s}\mathfrak{s}'}(\boldsymbol{x}, \boldsymbol{x}', t, t') = \sqrt{\mathcal{H}} \left\langle \frac{\delta g^{\ell+1}_{\mathfrak{s}}(\boldsymbol{x}, t)}{\delta u^\ell_{O\mathfrak{h}\mathfrak{s}'}(\boldsymbol{x}', t')} \right\rangle$$

$$R^{h^\ell, \xi^\ell_V}_{\mathfrak{h}\mathfrak{s}\mathfrak{s}'}(\boldsymbol{x}, \boldsymbol{x}', t, t') = \sqrt{\mathcal{H}} \left\langle \frac{\delta h^\ell_{\mathfrak{s}}(\boldsymbol{x}, t)}{\delta r^\ell_{V\mathfrak{h}\mathfrak{s}'}(\boldsymbol{x}', t')} \right\rangle$$

$$R^{h^\ell, \xi^\ell_K}_{\mathfrak{h}\mathfrak{s}\mathfrak{s}'}(\boldsymbol{x}, \boldsymbol{x}', t, t') = \sqrt{\mathcal{H}} \left\langle \frac{\delta h^\ell_{\mathfrak{s}}(\boldsymbol{x}, t)}{\delta r^\ell_{K\mathfrak{h}\mathfrak{s}'}(\boldsymbol{x}', t')} \right\rangle$$

$$R^{h^\ell, \xi^\ell_Q}_{\mathfrak{h}\mathfrak{s}\mathfrak{s}'}(\boldsymbol{x}, \boldsymbol{x}', t, t') = \sqrt{\mathcal{H}} \left\langle \frac{\delta h^\ell_{\mathfrak{s}}(\boldsymbol{x}, t)}{\delta r^\ell_{Q\mathfrak{h}\mathfrak{s}'}(\boldsymbol{x}', t')} \right\rangle$$

$$R^{v^\ell \hat{\xi}^\ell_O}_{\mathfrak{h}\mathfrak{s}\mathfrak{s}'}(\boldsymbol{x}, \boldsymbol{x}', t, t') = \left\langle \frac{\delta v^\ell_{\mathfrak{h}\mathfrak{s}}(\boldsymbol{x}, t)}{\delta r^\ell_{O\mathfrak{h}\mathfrak{s}'}(\boldsymbol{x}', t')} \right\rangle$$

$$R^{g^\ell_O \chi^\ell_V}_{\mathfrak{s}\mathfrak{s}'}(\boldsymbol{x}, \boldsymbol{x}', t, t') = \left\langle \frac{\delta g^\ell_{O\mathfrak{h}\mathfrak{s}}(\boldsymbol{x}, t)}{\delta u^\ell_{V\mathfrak{h}\mathfrak{s}'}(\boldsymbol{x}', t')} \right\rangle$$

$$R^{k^\ell \chi^\ell_Q}_{\mathfrak{s}\mathfrak{s}'}(\boldsymbol{x}, \boldsymbol{x}', t, t') = \left\langle \frac{\delta k^\ell_{\mathfrak{h}\mathfrak{s}}(\boldsymbol{x}, t)}{\delta u^\ell_{Q\mathfrak{h}\mathfrak{s}'}(\boldsymbol{x}', t')} \right\rangle$$

$$R^{q^\ell \chi^\ell_K}_{\mathfrak{s}\mathfrak{s}'}(\boldsymbol{x}, \boldsymbol{x}', t, t') = \left\langle \frac{\delta q^\ell_{\mathfrak{h}\mathfrak{s}}(\boldsymbol{x}, t)}{\delta u^\ell_{K\mathfrak{h}\mathfrak{s}'}(\boldsymbol{x}', t')} \right\rangle \tag{63}$$

**Single Site Stochastic Processes**   Our fields of interest will obey the stochastic dynamics

$$\chi_{O\mathfrak{h}\mathfrak{s}}^{\ell}(\boldsymbol{x},t) = u_{\mathfrak{h}\mathfrak{s}}^{\ell}(\boldsymbol{x},t) + \frac{1}{\sqrt{\mathcal{H}}}\sum_{t'\mathfrak{s}'}\int d\boldsymbol{x}' R_{\mathfrak{h}\mathfrak{s}\mathfrak{s}'}^{v^{\ell},\xi_{O}^{\ell}}(\boldsymbol{x},\boldsymbol{x}',t,t')g_{\mathfrak{s}'}^{\ell+1}(\boldsymbol{x}',t')$$

$$u_{O\mathfrak{h}\mathfrak{s}}^{\ell}(\boldsymbol{x},t) \sim \mathcal{GP}\left(0, V_{\mathfrak{h}\mathfrak{s}\mathfrak{s}'}^{\ell}(\boldsymbol{x},\boldsymbol{x}',t,t')\right)$$

$$\xi_{V\mathfrak{h}\mathfrak{s}}^{\ell}(\boldsymbol{x},t) = r_{V\mathfrak{h}\mathfrak{s}}^{\ell}(\boldsymbol{x},t) + \frac{1}{\sqrt{\mathcal{H}}}\sum_{t'\mathfrak{s}'}\int d\boldsymbol{x}' R_{\mathfrak{h}\mathfrak{s}\mathfrak{s}'}^{g_{O}^{\ell},\chi_{V}^{\ell}}(\boldsymbol{x},\boldsymbol{x}',t,t')h_{\mathfrak{s}'}^{\ell}(\boldsymbol{x}',t')$$

$$r_{V\mathfrak{h}\mathfrak{s}}^{\ell}(\boldsymbol{x},t) \sim \mathcal{GP}\left(0, G_{O\mathfrak{h}\mathfrak{s}\mathfrak{s}'}^{\ell}(\boldsymbol{x},\boldsymbol{x}',t,t')\right)$$

$$\xi_{K\mathfrak{h}\mathfrak{s}}^{\ell}(\boldsymbol{x},t) = r_{K\mathfrak{h}\mathfrak{s}}^{\ell}(\boldsymbol{x},t) + \frac{1}{\sqrt{\mathcal{H}}}\sum_{t'\mathfrak{s}'}\int d\boldsymbol{x}' R_{\mathfrak{h}\mathfrak{s}\mathfrak{s}'}^{q^{\ell},\chi_{K}^{\ell}}(\boldsymbol{x},\boldsymbol{x}',t,t')h_{\mathfrak{s}'}^{\ell}(\boldsymbol{x}',t')$$

$$r_{K\mathfrak{h}\mathfrak{s}}^{\ell}(\boldsymbol{x},t) \sim \mathcal{GP}\left(0, Q_{\mathfrak{h}\mathfrak{s}\mathfrak{s}'}^{\ell}(\boldsymbol{x},\boldsymbol{x}',t,t')\right)$$

$$\xi_{K\mathfrak{h}\mathfrak{s}}^{\ell}(\boldsymbol{x},t) = r_{K\mathfrak{h}\mathfrak{s}}^{\ell}(\boldsymbol{x},t) + \frac{1}{\sqrt{\mathcal{H}}}\sum_{t'\mathfrak{s}'}\int d\boldsymbol{x}' R_{\mathfrak{h}\mathfrak{s}\mathfrak{s}'}^{q^{\ell},\chi_{K}^{\ell}}(\boldsymbol{x},\boldsymbol{x}',t,t')h_{\mathfrak{s}'}^{\ell}(\boldsymbol{x}',t')$$

$$r_{K\mathfrak{h}\mathfrak{s}}^{\ell}(\boldsymbol{x},t) \sim \mathcal{GP}\left(0, Q_{\mathfrak{h}\mathfrak{s}\mathfrak{s}'}^{\ell}(\boldsymbol{x},\boldsymbol{x}',t,t')\right)$$

$$\xi_{O\mathfrak{h}\mathfrak{s}}^{\ell}(\boldsymbol{x},t) = r_{O\mathfrak{h}\mathfrak{s}}^{\ell}(\boldsymbol{x},t) + \sum_{t'\mathfrak{s}'}\int d\boldsymbol{x}' R_{\mathfrak{h}\mathfrak{s}\mathfrak{s}'}^{g^{\ell+1}\chi_{O}^{\ell}}(\boldsymbol{x},\boldsymbol{x}',t,t')v_{\mathfrak{h}\mathfrak{s}'}^{\ell}(\boldsymbol{x}',t')$$

$$r_{O\mathfrak{h}\mathfrak{s}}^{\ell}(\boldsymbol{x},t) \sim \mathcal{GP}\left(0, G_{\mathfrak{s}\mathfrak{s}'}^{\ell+1}(\boldsymbol{x},\boldsymbol{x}',t,t')\right)$$

$$\chi_{V\mathfrak{h}\mathfrak{s}}^{\ell}(\boldsymbol{x},t) = u_{V\mathfrak{h}\mathfrak{s}}^{\ell}(\boldsymbol{x},t) + \sum_{t'\mathfrak{s}'}\int d\boldsymbol{x}' R_{\mathfrak{h}\mathfrak{s}\mathfrak{s}'}^{h^{\ell}\xi_{V}^{\ell}}(\boldsymbol{x},\boldsymbol{x}',t,t')g_{O\mathfrak{h}\mathfrak{s}'}^{\ell}(\boldsymbol{x}',t')$$

$$u_{O\mathfrak{h}\mathfrak{s}}^{\ell}(\boldsymbol{x},t) \sim \mathcal{GP}\left(0, H_{\mathfrak{s}\mathfrak{s}'}^{\ell}(\boldsymbol{x},\boldsymbol{x}',t,t')\right)$$

$$\chi_{K\mathfrak{h}\mathfrak{s}}^{\ell}(\boldsymbol{x},t) = u_{K\mathfrak{h}\mathfrak{s}}^{\ell}(\boldsymbol{x},t) + \sum_{t'\mathfrak{s}'}\int d\boldsymbol{x}' R_{\mathfrak{h}\mathfrak{s}\mathfrak{s}'}^{h^{\ell}\xi_{K}^{\ell}}(\boldsymbol{x},\boldsymbol{x}',t,t')q_{\mathfrak{h}\mathfrak{s}'}^{\ell}(\boldsymbol{x}',t')$$

$$u_{K\mathfrak{h}\mathfrak{s}}^{\ell}(\boldsymbol{x},t) \sim \mathcal{GP}\left(0, H_{\mathfrak{s}\mathfrak{s}'}^{\ell}(\boldsymbol{x},\boldsymbol{x}',t,t')\right)$$

$$\chi_{Q\mathfrak{h}\mathfrak{s}}^{\ell}(\boldsymbol{x},t) = u_{Q\mathfrak{h}\mathfrak{s}}^{\ell}(\boldsymbol{x},t) + \sum_{t'\mathfrak{s}'}\int d\boldsymbol{x}' R_{\mathfrak{h}\mathfrak{s}\mathfrak{s}'}^{h^{\ell}\xi_{Q}^{\ell}}(\boldsymbol{x},\boldsymbol{x}',t,t')k_{\mathfrak{h}\mathfrak{s}'}^{\ell}(\boldsymbol{x}',t')$$

$$u_{Q\mathfrak{h}\mathfrak{s}}^{\ell}(\boldsymbol{x},t) \sim \mathcal{GP}\left(0, H_{\mathfrak{s}\mathfrak{s}'}^{\ell}(\boldsymbol{x},\boldsymbol{x}',t,t')\right) \tag{64}$$

**Residual Stream**   The forward pass residual variables obey the following stochastic process

$$h_{\mathfrak{s}}^{\ell+1}(\boldsymbol{x},t) = h_{\mathfrak{s}}^{\ell}(\boldsymbol{x},t) + \beta_0 L^{-\alpha_L}\bar{\chi}_{O\mathfrak{h}\mathfrak{s}}^{\ell}(\boldsymbol{x},t)$$
$$+ \eta_0\gamma_0\beta_0^2 L^{-1}\sum_{t'<t}\mathbb{E}_{\boldsymbol{x}'\sim\mathfrak{B}_{t'}}\Delta(\boldsymbol{x}',t')\sum_{\mathfrak{s}'}g_{\mathfrak{s}'}^{\ell+1}(\boldsymbol{x}',t')V_{\mathfrak{s}\mathfrak{s}'}^{\ell\sigma}(\boldsymbol{x},\boldsymbol{x}',t,s) \tag{65}$$

where

$$\bar{\chi}_{O\mathfrak{h}\mathfrak{s}}^{\ell}(\boldsymbol{x},t) = \frac{1}{\sqrt{\mathcal{H}}}\sum_{\mathfrak{h}=1}^{\mathcal{H}}\sum_{\mathfrak{s}'\in[\mathcal{S}]}\sigma_{\mathfrak{h}\mathfrak{s}\mathfrak{s}'}^{\ell}(\boldsymbol{x},t)\chi_{O\mathfrak{h}\mathfrak{s}'}^{\ell}(\boldsymbol{x},t) \tag{66}$$

the backward pass satisfies

$$g_{\mathfrak{s}}^{\ell}(\boldsymbol{x},t) = g_{\mathfrak{s}}^{\ell+1}(\boldsymbol{x},t) + \frac{\beta_0}{L^{\alpha_L}}\left[\bar{\xi}_{Q\mathfrak{s}}^{\ell}(\boldsymbol{x},t) + \bar{\xi}_{K\mathfrak{s}}^{\ell}(\boldsymbol{x},t) + \bar{\xi}_{V\mathfrak{s}}^{\ell}(\boldsymbol{x},t)\right]$$
$$+ \frac{\beta_0^2\eta_0\gamma_0}{L}\sum_{t<t'}\mathbb{E}_{\boldsymbol{x}'\sim t'}\Delta(\boldsymbol{x}',t')G_{O\mathfrak{s}\mathfrak{s}'}^{\ell\sigma}(\boldsymbol{x},\boldsymbol{x}',t,t')h_{\mathfrak{s}'}^{\ell}(\boldsymbol{x}',t')$$
$$+ \frac{\beta_0^2\eta_0\gamma_0}{L}\sum_{t<t'}\mathbb{E}_{\boldsymbol{x}'\sim t'}\Delta(\boldsymbol{x}',t')\left[K_{\mathfrak{s}\mathfrak{s}'}^{\ell M\dot{\sigma}}(\boldsymbol{x},\boldsymbol{x}',t,t') + Q_{\mathfrak{s}\mathfrak{s}'}^{\ell M\dot{\sigma}}(\boldsymbol{x},\boldsymbol{x}',t,t')\right]h_{\mathfrak{s}'}^{\ell}(\boldsymbol{x}',t')$$

$$\tag{67}$$

The keys and queries have dynamics

$$
\begin{aligned}
k_{\mathfrak{h}\mathfrak{s}}^{\ell}(\boldsymbol{x},t) &= \chi_{K\mathfrak{h}\mathfrak{s}}^{\ell}(\boldsymbol{x},t) \\
&\quad + \frac{\beta_0\eta_0\gamma_0}{L^{1-\alpha_L}N^{1-\alpha_{\mathcal{A}}}}\sum_{t'<t}\mathbb{E}_{\boldsymbol{x}\sim\mathcal{B}_{t'}}\sum_{\mathfrak{s}'\mathfrak{s}''\mathfrak{s}'''}\Delta(\boldsymbol{x},t')M_{\mathfrak{h}\mathfrak{s}'\mathfrak{s}''}^{\ell}(\boldsymbol{x},t')\dot{\sigma}_{\mathfrak{s}'\mathfrak{s}''\mathfrak{s}'''}^{\ell}q_{\mathfrak{h}\mathfrak{s}'''}^{\ell}(\boldsymbol{x},t')H_{\mathfrak{s}\mathfrak{s}'}^{\ell}(\boldsymbol{x},\boldsymbol{x}',t,t')
\end{aligned}
$$

$$
\begin{aligned}
q_{\mathfrak{h}\mathfrak{s}}^{\ell}(\boldsymbol{x},t) &= \chi_{Q\mathfrak{h}\mathfrak{s}}^{\ell}(\boldsymbol{x},t) \\
&\quad + \frac{\beta_0\eta_0\gamma_0}{L^{1-\alpha_L}}\sum_{t'<t}\mathbb{E}_{\boldsymbol{x}'\sim\mathcal{B}_{t'}}\sum_{\mathfrak{s}'\mathfrak{s}''}\Delta(\boldsymbol{x}',t')M_{\mathfrak{h}\mathfrak{s}\mathfrak{s}'}^{\ell}(\boldsymbol{x}',t')\dot{\sigma}_{\mathfrak{s}\mathfrak{s}'\mathfrak{s}''}^{\ell}(\boldsymbol{x}',t')k_{\mathfrak{h}\mathfrak{s}''}^{\ell}(\boldsymbol{x}',t')H_{\mathfrak{s}\mathfrak{s}'}^{\ell}(\boldsymbol{x},\boldsymbol{x}',t,t')
\end{aligned}
\tag{68}
$$

### E.2.1 Multi-head Attention is Single-Head Attention as $N \to \infty$

In this section, we use the derived saddle point equations for the $N \to \infty$ limit and argue that they imply that all heads in the MHSA layer learn identical attention matrices and contribute the same feature updates to the residual stream. To do so, we proceed in a three step inductive argument.

1. First, we show that at initialization, all key, query, value and attention matrices $\{Q_{\mathfrak{h}}, K_{\mathfrak{h}}, V_{\mathfrak{h}}, \mathcal{A}_{\mathfrak{h}}, M_{\mathfrak{h}}\}$ are equal across heads.

2. Next, we show inductively that if these quantities $\{Q_{\mathfrak{h}}, K_{\mathfrak{h}}, V_{\mathfrak{h}}, \mathcal{A}_{\mathfrak{h}}, M_{\mathfrak{h}}\}$ are identical across heads up to some time, then that implies that the response functions $R_{\mathfrak{h}}$ are also identical across heads up to that time.

3. Lastly, we show that if the response functions $R_{\mathfrak{h}}$ are identical up to some time, then that implies that the MHSA kernels $\{Q_{\mathfrak{h}}, K_{\mathfrak{h}}, V_{\mathfrak{h}}, \mathcal{A}_{\mathfrak{h}}, M_{\mathfrak{h}}\}$ will also be identical across heads at future times.

First, we note that, at initialization, all of the MHSA kernels are identical across heads since

$$
\forall\, \mathfrak{h} \in [\mathcal{H}] \quad Q_{\mathfrak{h}\mathfrak{s}}^{\ell}(\boldsymbol{x},\boldsymbol{x}',0,0) = K_{\mathfrak{h}\mathfrak{s}}^{\ell}(\boldsymbol{x},\boldsymbol{x}',0,0) = V_{\mathfrak{h}\mathfrak{s}}^{\ell}(\boldsymbol{x},\boldsymbol{x}',0,0) = H_{\mathfrak{s}\mathfrak{s}'}^{\ell}(\boldsymbol{x},\boldsymbol{x}',0,0)
$$
$$
M_{\mathfrak{h}\mathfrak{s}\mathfrak{s}'}^{\ell}(\boldsymbol{x},0) = A_{\mathfrak{h}\mathfrak{s}\mathfrak{s}'}^{\ell}(\boldsymbol{x},0) = 0.
\tag{69}
$$

Next, we need to analyze the response functions under an inductive hypothesis on the equality of the MHSA kernels. We start by noting that all response functions are causal so we can group the response functions that arise from $\chi_{O\mathfrak{h}}^{\ell}$ with the feature learning update to the residual stream, writing the following compressed equation for the forward and backward passes

$$
\begin{aligned}
h_{\mathfrak{s}}^{\ell+1}(\boldsymbol{x},t) &= h_{\mathfrak{s}}^{\ell}(\boldsymbol{x},t) + \frac{1}{\sqrt{\mathcal{H}}}\sum_{\mathfrak{h}=1}^{\mathcal{H}}\sum_{\mathfrak{s}'}u_{O\mathfrak{h}\mathfrak{s}'}^{\ell}(\boldsymbol{x},t)\sigma_{\mathfrak{h}\mathfrak{s}\mathfrak{s}'}^{\ell}(\boldsymbol{x},t) \\
&\quad + \eta_0\gamma_0\beta_0^2 L^{-1}\sum_{t'<t}\sum_{\mathfrak{s}'}\int d\boldsymbol{x}'\, C_{\mathfrak{s}\mathfrak{s}'}^{h^{\ell}}(\boldsymbol{x},\boldsymbol{x}',t,t')g_{\mathfrak{s}'}^{\ell+1}(\boldsymbol{x}',t') \\[6pt]
g_{\mathfrak{s}}^{\ell}(\boldsymbol{x},t) &= g_{\mathfrak{s}}^{\ell}(\boldsymbol{x},t) + \frac{1}{\sqrt{\mathcal{H}}}\sum_{\mathfrak{h}\mathfrak{s}'}r_{V\mathfrak{h}\mathfrak{s}'}^{\ell}(\boldsymbol{x},t)\sigma_{\mathfrak{h}\mathfrak{s}'\mathfrak{s}}^{\ell}(\boldsymbol{x},t) \\
&\quad + \frac{1}{\sqrt{\mathcal{H}}}\sum_{\mathfrak{h}\mathfrak{s}'\mathfrak{s}''}r_{Q\mathfrak{h}\mathfrak{s}'}^{\ell}(\boldsymbol{x},t)\dot{\sigma}_{\mathfrak{h}\mathfrak{s}'\mathfrak{s}''\mathfrak{s}}^{\ell}(\boldsymbol{x},t)M_{\mathfrak{h}\mathfrak{s}'\mathfrak{s}''}^{\ell}(\boldsymbol{x},t) \\
&\quad + \frac{1}{\sqrt{\mathcal{H}}}\sum_{\mathfrak{h}\mathfrak{s}''\mathfrak{s}'''}r_{K\mathfrak{h}\mathfrak{s}'''}^{\ell}(\boldsymbol{x},t)\dot{\sigma}_{\mathfrak{h}\mathfrak{s}\mathfrak{s}''\mathfrak{s}'''}^{\ell}(\boldsymbol{x},t)M_{\mathfrak{h}\mathfrak{s}\mathfrak{s}''}^{\ell}(\boldsymbol{x},t) \\
&\quad + \eta_0\gamma_0\beta_0^2 L^{-1}\sum_{t'<t}\sum_{\mathfrak{s}'}\int d\boldsymbol{x}'\, C_{\mathfrak{s}\mathfrak{s}'}^{g^{\ell}}(\boldsymbol{x},\boldsymbol{x}',t,t')h_{\mathfrak{s}'}^{\ell}(\boldsymbol{x}',t')
\end{aligned}
$$

where $C^{h^\ell}$ and $C^{g^\ell}$ only involve deterministic *head-averaged* kernels and thus do not carry a $\mathfrak{h}$ index. We now derive useful response function identities

$$\sqrt{\mathcal{H}}\frac{\delta h_{\mathfrak{s}}^\ell(\boldsymbol{x},t)}{\delta u_{O\mathfrak{h}\mathfrak{s}'}^{\ell'}(\boldsymbol{x}',t')} = \Theta(\ell-\ell')\delta(t-t')\delta(\boldsymbol{x}-\boldsymbol{x}')\sigma_{\mathfrak{h}\mathfrak{s}\mathfrak{s}'}^\ell(\boldsymbol{x},t)$$

$$+\eta_0\gamma_0\beta_0^2 L^{-1}\sum_{k=1}^\ell\sum_{t''<t}\sum_{\mathfrak{s}''}\int d\boldsymbol{x}''C_{\mathfrak{s}\mathfrak{s}''}^{h^k}(\boldsymbol{x},\boldsymbol{x}'',t,t'')\left(\sqrt{\mathcal{H}}\frac{\delta g_{\mathfrak{s}''}^{k+1}(\boldsymbol{x}'',t'')}{\delta u_{O\mathfrak{h}\mathfrak{s}'}^{\ell'}(\boldsymbol{x}',t')}\right)$$

$$\sqrt{\mathcal{H}}\frac{\delta g_{\mathfrak{s}''}^\ell(\boldsymbol{x}'',t'')}{\delta u_{O\mathfrak{h}\mathfrak{s}'}^{\ell'}(\boldsymbol{x}',t')} = \eta_0\gamma_0\beta_0^2 L^{-1}\sum_{k=\ell}^L\sum_{t''<t}\sum_{\mathfrak{s}''}\int d\boldsymbol{x}''C_{\mathfrak{s}\mathfrak{s}''}^{g^k}(\boldsymbol{x},\boldsymbol{x}'',t,t'')\left(\sqrt{\mathcal{H}}\frac{\delta h_{\mathfrak{s}''}^k(\boldsymbol{x}'',t'')}{\delta u_{O\mathfrak{h}\mathfrak{s}'}^{\ell'}(\boldsymbol{x}',t')}\right)$$

$$\tag{70}$$

These equations give the needed response function $R^{g^{\ell+1}\chi_O^\ell}$. From these equations we immediately see that if $\sigma_{\mathfrak{h}\mathfrak{s}\mathfrak{s}'}^\ell(\boldsymbol{x},t) = \sigma_{\mathfrak{h}'\mathfrak{s}\mathfrak{s}'}^\ell(\boldsymbol{x},t)$ (which holds under our inductive hypothesis) then $R_\mathfrak{h}^{g^{\ell+1}\chi_O^\ell} = R_{\mathfrak{h}'}^{g^{\ell+1}\chi_O^\ell}$. This same argument is repeated for all other response functions that are computed as derivatives of residual stream variables. We have thus found that

$$\forall \mathfrak{h},\mathfrak{h}'\in[\mathcal{H}], \ R_\mathfrak{h}^{g^{\ell+1}\chi_O^\ell} = R_{\mathfrak{h}'}^{g^{\ell+1}\chi_O^\ell}, \ R_\mathfrak{h}^{h^\ell\xi_V^\ell} = R_{\mathfrak{h}'}^{h^\ell\xi_V^\ell}, \ R_\mathfrak{h}^{h^\ell\xi_K^\ell} = R_{\mathfrak{h}'}^{h^\ell\xi_K^\ell}, \ R_\mathfrak{h}^{h^\ell\xi_Q^\ell} = R_{\mathfrak{h}'}^{h^\ell\xi_Q^\ell}$$

Now, we can analyze the dynamics of the keys, queries and values within a head using the above property and the original inductive hypothesis that $\mathcal{A}_\mathfrak{h} = \mathcal{A}_{\mathfrak{h}'}, M_\mathfrak{h} = M_{\mathfrak{h}'}...$ for times less than $t$. We will now prove that this implies that these variables will remain the same. We start by examining the keys and queries which have the form

$$k_{\mathfrak{h}\mathfrak{s}}^\ell(\boldsymbol{x},t) = u_{K\mathfrak{h}\mathfrak{s}}^\ell(\boldsymbol{x},t) + \sum_{t'<t}\sum_{\mathfrak{s}'}\int d\boldsymbol{x}'C_{\mathfrak{s}\mathfrak{s}'}^{k^\ell}(\boldsymbol{x},\boldsymbol{x}',t,t')q_{\mathfrak{h}\mathfrak{s}'}^\ell(\boldsymbol{x}',t')\tag{71}$$

$$q_{\mathfrak{h}\mathfrak{s}}^\ell(\boldsymbol{x},t) = u_{Q\mathfrak{h}\mathfrak{s}}^\ell(\boldsymbol{x},t) + \sum_{t'<t}\sum_{\mathfrak{s}'}\int d\boldsymbol{x}'C_{\mathfrak{s}\mathfrak{s}'}^{q^\ell}(\boldsymbol{x},\boldsymbol{x}',t,t')k_{\mathfrak{h}\mathfrak{s}'}^\ell(\boldsymbol{x}',t')\tag{72}$$

where $C_{\mathfrak{s}\mathfrak{s}'}^{k^\ell}(\boldsymbol{x},\boldsymbol{x}',t,t')$ and $C_{\mathfrak{s}\mathfrak{s}'}^{q^\ell}(\boldsymbol{x},\boldsymbol{x}',t,t')$ are two operators that do not carry a $\mathfrak{h}$ (are the same across all heads). These relations can be viewed as a linear system of equations. We let $\boldsymbol{k}_\mathfrak{h}^\ell = \text{Vec}\{k_{\mathfrak{h}\mathfrak{s}}^\ell(\boldsymbol{x},t)\}_{\mathfrak{s}\boldsymbol{x}t}$ and analogously $\boldsymbol{C}^{k^\ell} = \text{Mat}\{C_{\mathfrak{s}\mathfrak{s}'}^{k^\ell}(\boldsymbol{x},\boldsymbol{x}',t,t')\}$ as in the calculations of Bordelon and Pehlevan [15, 41]. Using this shorthand, we can express the key/query and attention kernels as

$$\boldsymbol{k}_\mathfrak{h}^\ell = \left[\boldsymbol{I}-\boldsymbol{C}^{k^\ell}\boldsymbol{C}^{q^\ell}\right]^{-1}\left[\boldsymbol{u}_{K\mathfrak{h}}^\ell+\boldsymbol{C}^{k^\ell}\boldsymbol{u}_{Q\mathfrak{h}}^\ell\right], \ \boldsymbol{q}_\mathfrak{h}^\ell = \left[\boldsymbol{I}-\boldsymbol{C}^{q^\ell}\boldsymbol{C}^{k^\ell}\right]^{-1}\left[\boldsymbol{u}_{Q\mathfrak{h}}^\ell+\boldsymbol{C}^{q^\ell}\boldsymbol{u}_{K\mathfrak{h}}^\ell\right]$$

$$\boldsymbol{K}_\mathfrak{h}^\ell = \left[\boldsymbol{I}-\boldsymbol{C}^{k^\ell}\boldsymbol{C}^{q^\ell}\right]^{-1}\left[\boldsymbol{H}^\ell+\boldsymbol{C}^{k^\ell}\boldsymbol{H}^\ell\boldsymbol{C}^{k^\ell\top}\right]\left[\boldsymbol{I}-\boldsymbol{C}^{k^\ell}\boldsymbol{C}^{q^\ell}\right]^{-1\top}$$

$$\boldsymbol{Q}_\mathfrak{h}^\ell = \left[\boldsymbol{I}-\boldsymbol{C}^{q^\ell}\boldsymbol{C}^{k^\ell}\right]^{-1}\left[\boldsymbol{H}^\ell+\boldsymbol{C}^{q^\ell}\boldsymbol{H}^\ell\boldsymbol{C}^{q^\ell\top}\right]\left[\boldsymbol{I}-\boldsymbol{C}^{q^\ell}\boldsymbol{C}^{k^\ell}\right]^{-1\top}$$

$$\boldsymbol{\mathcal{A}}_\mathfrak{h}^\ell = \left[\boldsymbol{I}-\boldsymbol{C}^{k^\ell}\boldsymbol{C}^{q^\ell}\right]^{-1}\boldsymbol{C}^{k^\ell}\boldsymbol{H}^\ell\boldsymbol{C}^{q^\ell\top}\left[\boldsymbol{I}-\boldsymbol{C}^{q^\ell}\boldsymbol{C}^{k^\ell}\right]^{-1\top}\tag{73}$$

We thus see that the final kernels $\boldsymbol{K}_\mathfrak{h}^\ell, \boldsymbol{Q}_\mathfrak{h}^\ell, \boldsymbol{\mathcal{A}}_\mathfrak{h}^\ell$ are all identical across heads. An identical argument can be carried out for the value kernel $V_\mathfrak{h}^\ell$ and the $M_\mathfrak{h}^\ell$ order parameter.

### E.3 Infinite $\mathcal{H}$ Limit

In this section, we compute the infinite head limit with $N, L$ fixed. This limit is more technically involved than the $N\to\infty$ limit which required only a simple saddle point of the full DMFT action over all kernels. At finite $N$ we cannot use this technique since the kernels within MHSA blocks are *random variables*. However, as was shown in Bordelon and Pehlevan [17], the DMFT action still contains the necessary information to characterize the distribution over order parameters at finite $N$. In the case of transformers with infinitely many heads, a subset of the order parameters introduced in

the previous section will still concentrate as $\mathcal{H} \to \infty$ including

$$H_{\mathfrak{s}\mathfrak{s}'}^{\ell}(\boldsymbol{x}, \boldsymbol{x}', t, t') = \frac{1}{N\mathcal{H}} \boldsymbol{h}_{\mathfrak{s}}^{\ell}(\boldsymbol{x}, t) \cdot \boldsymbol{h}_{\mathfrak{s}}^{\ell}(\boldsymbol{x}, t)$$

$$G_{\mathfrak{s}\mathfrak{s}'}^{\ell}(\boldsymbol{x}, \boldsymbol{x}', t, t') = \frac{1}{N\mathcal{H}} \boldsymbol{g}_{\mathfrak{s}}^{\ell}(\boldsymbol{x}, t) \cdot \boldsymbol{g}_{\mathfrak{s}}^{\ell}(\boldsymbol{x}, t)$$

$$V_{\mathfrak{s}\mathfrak{s}'}^{\ell\sigma}(\boldsymbol{x}, \boldsymbol{x}', t, t') = \frac{1}{\mathcal{H}} \sum_{\mathfrak{h}=1}^{\mathcal{H}} \sum_{\mathfrak{s}''\mathfrak{s}'''} V_{\mathfrak{h}\mathfrak{s}''\mathfrak{s}'''}^{\ell\sigma}(\boldsymbol{x}, \boldsymbol{x}', t, t') \sigma_{\mathfrak{h}\mathfrak{s}\mathfrak{s}''}^{\ell}(\boldsymbol{x}, t) \sigma_{\mathfrak{h}\mathfrak{s}\mathfrak{s}'''}^{\ell}(\boldsymbol{x}', t')$$

and many more correlation and response functions. We will call the full collection of all the necessary head-averaged order parameters $\boldsymbol{Q}_{\text{global}}$. Further, not all of the stochastic fields will be relevant to characterize the residual stream. Specifically, only head-averaged fields $\bar{\chi}_O^{\ell}, \bar{\xi}_V^{\ell}, \bar{\xi}_K^{\ell}, \bar{\xi}_Q^{\ell}$ are relevant. For example, the first of these is defined as

$$\bar{\chi}_{O\mathfrak{s}}^{\ell}(\boldsymbol{x}, t) = \frac{1}{\sqrt{\mathcal{H}}} \sum_{\mathfrak{h}=1}^{\mathcal{H}} \sum_{\mathfrak{s}' \in [\mathcal{S}]} \sigma_{\mathfrak{h}\mathfrak{s}\mathfrak{s}'}^{\ell}(\boldsymbol{x}, t) \chi_{O\mathfrak{h}\mathfrak{s}'}^{\ell}(\boldsymbol{x}, t)$$

$$= \frac{1}{\sqrt{\mathcal{H}}} \sum_{\mathfrak{h}=1}^{\mathcal{H}} \sum_{\mathfrak{s}' \in [\mathcal{S}]} \sigma_{\mathfrak{h}\mathfrak{s}\mathfrak{s}'}^{\ell}(\boldsymbol{x}, t) \left[ u_{O\mathfrak{h}\mathfrak{s}'}^{\ell}(\boldsymbol{x}, t) + \frac{1}{\sqrt{\mathcal{H}}} \sum_{t'\mathfrak{s}''} \int d\boldsymbol{x}' R_{\mathfrak{h}\mathfrak{s}'\mathfrak{s}''}^{v^{\ell}, \xi_O^{\ell}}(\boldsymbol{x}, \boldsymbol{x}', t, t') g_{\mathfrak{s}''}^{\ell+1}(\boldsymbol{x}', t') \right]$$

$$= \bar{u}_{O\mathfrak{s}'}^{\ell}(\boldsymbol{x}, t) + \sum_{t'\mathfrak{s}'} \int d\boldsymbol{x}' \bar{R}_{\mathfrak{s}\mathfrak{s}'}^{v^{\ell}\xi_O^{\ell}\sigma}(\boldsymbol{x}, \boldsymbol{x}', t, t') g_{\mathfrak{s}'}^{\ell+1}(\boldsymbol{x}', t') \tag{74}$$

We note that the above equation is true at any value of $N$ but the covariance of $u_{\mathfrak{h}}^{\ell}$ and the response functions $R_{\mathfrak{h}}^{v^{\ell}\xi_O^{\ell}}$ are random variables [17]. However, the residual stream only depends upon collective, head-averaged variables

$$\bar{u}_{O\mathfrak{s}'}^{\ell}(\boldsymbol{x}, t) = \frac{1}{\sqrt{\mathcal{H}}} \sum_{\mathfrak{h}=1}^{\mathcal{H}} \sum_{\mathfrak{s}' \in [\mathcal{S}]} \sigma_{\mathfrak{h}\mathfrak{s}\mathfrak{s}'}^{\ell} u_{O\mathfrak{h}\mathfrak{s}'}^{\ell}(\boldsymbol{x}, t) \tag{75}$$

$$\bar{R}_{\mathfrak{s}\mathfrak{s}'}^{v^{\ell}\xi_O^{\ell}\sigma}(\boldsymbol{x}, \boldsymbol{x}', t, t') = \frac{1}{\mathcal{H}} \sum_{\mathfrak{h}=1}^{\mathcal{H}} \sum_{\mathfrak{s}''} \sigma_{\mathfrak{h}\mathfrak{s}\mathfrak{s}''}^{\ell}(\boldsymbol{x}, t) R_{\mathfrak{h}\mathfrak{s}''\mathfrak{s}'}^{v^{\ell}, \xi_O^{\ell}}(\boldsymbol{x}, \boldsymbol{x}', t, t'). \tag{76}$$

The intuition behind the large $\mathcal{H}$ limit is that, even though $\sigma_{\mathfrak{h}}^{\ell}$ and $R_{\mathfrak{h}}^{v^{\ell}\xi_O}$ are random variables, there should be a central limit theorem for $\bar{u}_O^{\ell}$ and a law of large numbers for $\bar{R}_{\mathfrak{s}\mathfrak{s}'}^{v^{\ell}\xi_O^{\ell}\sigma}(\boldsymbol{x}, \boldsymbol{x}', t, t')$.

### E.3.1   Partitioning Order Parameters

Based on the intuition developed in the previous section, we now derive an alternative DMFT action by tracking the moment generating functional for the head-averaged random fields that occur on the residual stream $\bar{\chi}_O^{\ell}, \bar{\xi}_V^{\ell}, \bar{\xi}_K^{\ell}, \bar{\xi}_Q^{\ell}$. To characterize this joint distribution, we must also of course keep track of the random fields within the MHSA blocks such as $\{\chi_{Q\mathfrak{h}}^{\ell}, \chi_{K\mathfrak{h}}^{\ell}, \chi_{V\mathfrak{h}}^{\ell}, \xi_{O\mathfrak{h}}^{\ell}\}_{\mathfrak{h} \in [\mathcal{H}]}$. Repeating the path integral setup of the previous section, we need to performing averages over all initial weights such as

$$\ln \mathbb{E}_{\{\boldsymbol{W}_{O\mathfrak{h}}^{\ell}(0)\}} \exp \left( -\frac{i}{\sqrt{N\mathcal{H}}} \sum_{\mathfrak{h}=1}^{\mathcal{H}} \text{Tr} \boldsymbol{W}_{O\mathfrak{h}}^{\ell}(0)^{\top} \sum_{t\mathfrak{s}} \int d\boldsymbol{x} \left[ \hat{\bar{\chi}}_{O\mathfrak{s}}^{\ell}(\boldsymbol{x}, t) \boldsymbol{v}_{\mathfrak{h}\mathfrak{s}}^{\ell\sigma}(\boldsymbol{x}, t)^{\top} + \boldsymbol{g}_{\mathfrak{s}}^{\ell+1}(\boldsymbol{x}, t) \hat{\boldsymbol{\xi}}_{O\mathfrak{h}\mathfrak{s}}^{\ell}(\boldsymbol{x}, t)^{\top} \right] \right)$$

$$= -\frac{1}{2} \sum_{tt'\mathfrak{s}\mathfrak{s}'} \int d\boldsymbol{x} d\boldsymbol{x}' \hat{\bar{\chi}}_{O\mathfrak{s}}^{\ell}(\boldsymbol{x}, t) \cdot \hat{\bar{\chi}}_{O\mathfrak{s}'}^{\ell}(\boldsymbol{x}', t') V_{\mathfrak{s}\mathfrak{s}'}^{\ell\sigma}(\boldsymbol{x}, \boldsymbol{x}', t, t')$$

$$- \frac{1}{2} \sum_{\mathfrak{h} \in [\mathcal{H}]} \sum_{tt'\mathfrak{s}\mathfrak{s}'} \int d\boldsymbol{x} d\boldsymbol{x}' \hat{\boldsymbol{\xi}}_{O\mathfrak{h}\mathfrak{s}}^{\ell}(\boldsymbol{x}, t) \cdot \hat{\boldsymbol{\xi}}_{O\mathfrak{h}\mathfrak{s}'}^{\ell}(\boldsymbol{x}', t') G_{\mathfrak{s}\mathfrak{s}'}^{\ell+1}(\boldsymbol{x}, \boldsymbol{x}', t, t')$$

$$- \frac{1}{N\mathcal{H}} \sum_{tt'\mathfrak{s}\mathfrak{s}'} \int d\boldsymbol{x} d\boldsymbol{x}' \left( \hat{\bar{\chi}}_{O\mathfrak{s}}^{\ell}(\boldsymbol{x}, t) \cdot \boldsymbol{g}_{\mathfrak{s}'}^{\ell+1}(\boldsymbol{x}', t') \right) \left( \sum_{\mathfrak{h} \in [\mathcal{H}]} \boldsymbol{v}_{\mathfrak{h}\mathfrak{s}}^{\ell\sigma}(\boldsymbol{x}, t) \cdot \hat{\boldsymbol{\xi}}_{O\mathfrak{h}\mathfrak{s}'}^{\ell}(\boldsymbol{x}', t') \right) \tag{77}$$

It is clear that from this integral that the relevant self-averaging order parameters are

$$V_{\mathfrak{s}\mathfrak{s}'}^{\ell\sigma}(\boldsymbol{x},\boldsymbol{x}',t,t') = \frac{1}{N\mathcal{H}}\sum_{\mathfrak{h}=1}^{\mathcal{H}}\boldsymbol{v}_{\mathfrak{h}\mathfrak{s}}^{\ell\sigma}(\boldsymbol{x},t)\cdot\boldsymbol{v}_{\mathfrak{h}\mathfrak{s}'}^{\ell\sigma}(\boldsymbol{x}',t')$$

$$G_{\mathfrak{s}\mathfrak{s}'}^{\ell}(\boldsymbol{x},\boldsymbol{x}',t,t') = \frac{1}{N\mathcal{H}}\sum_{\mathfrak{h}=1}^{\mathcal{H}}\boldsymbol{g}_{\mathfrak{s}}^{\ell}(\boldsymbol{x},t)\cdot\boldsymbol{g}_{\mathfrak{s}'}^{\ell}(\boldsymbol{x}',t')$$

$$R_{\mathfrak{s}\mathfrak{s}'}^{g^{\ell}\bar{\chi}^{O}}(\boldsymbol{x},\boldsymbol{x}',t,t') = -\frac{i}{N\mathcal{H}}\boldsymbol{g}_{\mathfrak{s}}^{\ell+1}(\boldsymbol{x},t)\cdot\hat{\bar{\boldsymbol{\chi}}}_{O\mathfrak{s}'}^{\ell}(\boldsymbol{x}',t')$$

$$\bar{R}_{\mathfrak{s}\mathfrak{s}'}^{v^{\ell}\xi_{O}^{\ell}}(\boldsymbol{x},\boldsymbol{x}',t,t') = -\frac{i}{N\mathcal{H}}\sum_{\mathfrak{h}=1}^{\mathcal{H}}\boldsymbol{v}_{\mathfrak{s}}^{\ell\sigma}(\boldsymbol{x},t)\cdot\hat{\boldsymbol{\xi}}_{O\mathfrak{h}\mathfrak{s}'}^{\ell}(\boldsymbol{x}',t') \tag{78}$$

We repeat this same procedure for all collections of weights and arrive at the following set of order parameters

$$\begin{aligned}
\boldsymbol{Q}_{\text{global}} = \text{Vec}&\{H^{\ell},G^{\ell},V^{\ell\sigma},K^{\ell M\dot{\sigma}},Q^{\ell M\dot{\sigma}},G_{O}^{\ell\sigma}\}\\
&\cup\{\hat{H}^{\ell},\hat{G}^{\ell},\hat{V}^{\ell\sigma},\hat{K}^{\ell M\dot{\sigma}},\hat{Q}^{\ell M\dot{\sigma}},\hat{G}_{O}^{\ell\sigma}\}\\
&\cup\{R^{g^{\ell}\bar{\chi}_{O}},R^{h^{\ell}\bar{\xi}_{V}},R^{h^{\ell}\bar{\xi}_{K}},R^{h^{\ell}\bar{\xi}_{Q}}\}\\
&\cup\{\bar{R}^{v^{\ell}\xi_{O}^{\ell}},\bar{R}^{g_{O}^{\ell}\chi_{V}^{\ell}},\bar{R}^{q^{\ell}\chi_{K}^{\ell}},\bar{R}^{k^{\ell}\chi_{Q}^{\ell}}\}
\end{aligned} \tag{79}$$

We expect that these order parameters will be self-averaging since they involve averages over $\mathcal{H}$ variables. However, the other variables we introduced $\{\mathcal{A}_{\mathfrak{h}},\mathcal{Q}_{\mathfrak{h}},K_{\mathfrak{h}},V_{\mathfrak{h}}\}$ will not concentrate at finite $N$ and will instead behave as random variables.

After introducing these order parameters we find that the moment generating functional has the form

$$Z = \int d\boldsymbol{Q}_{\text{global}}\exp\left(N\mathcal{H}L\,S(\boldsymbol{Q}_{\text{global}})\right) \tag{80}$$

where $S$ has the form

$$\begin{aligned}
S =\ & \frac{1}{L}\sum_{\ell=1}^{L}\sum_{tt'\mathfrak{s}\mathfrak{s}'}\int d\boldsymbol{x}d\boldsymbol{x}'[H_{\mathfrak{s}\mathfrak{s}'}^{\ell}(\boldsymbol{x},\boldsymbol{x}',t,t')\hat{H}_{\mathfrak{s}\mathfrak{s}'}^{\ell}(\boldsymbol{x},\boldsymbol{x}',t,t') + G_{\mathfrak{s}\mathfrak{s}'}^{\ell}(\boldsymbol{x},\boldsymbol{x}',t,t')\hat{G}_{\mathfrak{s}\mathfrak{s}'}^{\ell}(\boldsymbol{x},\boldsymbol{x}',t,t')]\\
& + \frac{1}{L}\sum_{\ell=1}^{L}\sum_{tt'\mathfrak{s}\mathfrak{s}'}\int d\boldsymbol{x}d\boldsymbol{x}'[V_{\mathfrak{s}\mathfrak{s}'}^{\ell\sigma}(\boldsymbol{x},\boldsymbol{x}',t,t')\hat{V}_{\mathfrak{s}\mathfrak{s}'}^{\ell\sigma}(\boldsymbol{x},\boldsymbol{x}',t,t') + K_{\mathfrak{s}\mathfrak{s}'}^{\ell M\dot{\sigma}}(\boldsymbol{x},\boldsymbol{x}',t,t')\hat{K}_{\mathfrak{s}\mathfrak{s}'}^{\ell M\dot{\sigma}}(\boldsymbol{x},\boldsymbol{x}',t,t')]\\
& + \frac{1}{L}\sum_{\ell=1}^{L}\sum_{tt'\mathfrak{s}\mathfrak{s}'}\int d\boldsymbol{x}d\boldsymbol{x}'[G_{O\mathfrak{s}\mathfrak{s}'}^{\ell\sigma}(\boldsymbol{x},\boldsymbol{x}',t,t')\hat{G}_{O\mathfrak{s}\mathfrak{s}'}^{\ell\sigma}(\boldsymbol{x},\boldsymbol{x}',t,t') + Q_{\mathfrak{s}\mathfrak{s}'}^{\ell M\dot{\sigma}}(\boldsymbol{x},\boldsymbol{x}',t,t')\hat{Q}_{\mathfrak{s}\mathfrak{s}'}^{\ell M\dot{\sigma}}(\boldsymbol{x},\boldsymbol{x}',t,t')]\\
& - \frac{1}{L}\sum_{\ell=1}^{L}\sum_{tt'\mathfrak{s}\mathfrak{s}'}\int d\boldsymbol{x}d\boldsymbol{x}'[R_{\mathfrak{s}\mathfrak{s}'}^{g^{\ell}\bar{\chi}_{O}}(\boldsymbol{x},\boldsymbol{x}',t,t')\bar{R}_{\mathfrak{s}'\mathfrak{s}}^{v^{\ell}\xi_{O}^{\ell}}(\boldsymbol{x}',\boldsymbol{x},t',t) + R_{\mathfrak{s}\mathfrak{s}'}^{h^{\ell}\bar{\xi}_{V}^{\ell}}(\boldsymbol{x},\boldsymbol{x}',t,t')\bar{R}_{\mathfrak{s}'\mathfrak{s}}^{g_{O}^{\ell}\chi_{V}^{\ell}}(\boldsymbol{x}',\boldsymbol{x},t',t)]\\
& - \frac{1}{L}\sum_{\ell=1}^{L}\sum_{tt'\mathfrak{s}\mathfrak{s}'}\int d\boldsymbol{x}d\boldsymbol{x}'[R_{\mathfrak{s}\mathfrak{s}'}^{h^{\ell}\bar{\xi}_{K}}(\boldsymbol{x},\boldsymbol{x}',t,t')\bar{R}_{\mathfrak{s}'\mathfrak{s}}^{q^{\ell}\chi_{K}^{\ell}}(\boldsymbol{x}',\boldsymbol{x},t',t) + R_{\mathfrak{s}\mathfrak{s}'}^{h^{\ell}\bar{\xi}_{Q}}(\boldsymbol{x},\boldsymbol{x}',t,t')\bar{R}_{\mathfrak{s}'\mathfrak{s}}^{k^{\ell}\chi_{Q}^{\ell}}(\boldsymbol{x}',\boldsymbol{x},t',t)]\\
& + \frac{1}{L}\ln\mathcal{Z}_{\text{res}} + \frac{1}{NL}\sum_{\ell=1}^{L}\ln\mathcal{Z}_{\text{MHSA}}^{\ell} \tag{81}
\end{aligned}$$

The single site moment generating function for the residual stream has the form

$$
\mathcal{Z}_{\text{res}} = \int \prod_{\ell s x t} \frac{d\bar{\chi}_{Os}^\ell(\boldsymbol{x},t)d\hat{\bar{\chi}}_{Os}^\ell(\boldsymbol{x},t)}{2\pi} \frac{d\bar{\xi}_{Qs}^\ell(\boldsymbol{x},t)d\hat{\bar{\xi}}_{Qs}^\ell(\boldsymbol{x},t)}{2\pi} \frac{d\bar{\xi}_{Ks}^\ell(\boldsymbol{x},t)d\hat{\bar{\xi}}_{Ks}^\ell(\boldsymbol{x},t)}{2\pi} \frac{d\bar{\xi}_{Vs}^\ell(\boldsymbol{x},t)d\hat{\bar{\xi}}_{Vs}^\ell(\boldsymbol{x},t)}{2\pi}
$$

$$
\exp\left(-\sum_{\ell t t' s s'} \int d\boldsymbol{x}d\boldsymbol{x}'\left[h_s^\ell(\boldsymbol{x},t)h_{s'}^\ell(\boldsymbol{x}',t')\hat{H}_{ss'}^\ell(\boldsymbol{x},\boldsymbol{x}',t,t') + g_s^\ell(\boldsymbol{x},t)g_{s'}^\ell(\boldsymbol{x}',t')\hat{G}_{ss'}^\ell(\boldsymbol{x},\boldsymbol{x}',t,t')\right]\right)
$$

$$
\exp\left(-\frac{1}{2}\sum_{\ell t t' s s'} \int d\boldsymbol{x}d\boldsymbol{x}'\left[\hat{\chi}_{Os}^\ell(\boldsymbol{x},t)\hat{\chi}_{Os'}^\ell(\boldsymbol{x}',t')V_{ss'}^{\ell\sigma}(\boldsymbol{x},\boldsymbol{x}',t,t') + \hat{\bar{\xi}}_{Vs}^\ell(\boldsymbol{x},t)\hat{\bar{\xi}}_{Vs'}^\ell(\boldsymbol{x}',t')G_{Oss'}^{\ell\sigma}(\boldsymbol{x},\boldsymbol{x}',t,t')\right]\right)
$$

$$
\exp\left(-\frac{1}{2}\sum_{\ell t t' s s'} \int d\boldsymbol{x}d\boldsymbol{x}'\left[\hat{\bar{\xi}}_{Ks}^\ell(\boldsymbol{x},t)\hat{\bar{\xi}}_{Ks'}^\ell(\boldsymbol{x}',t')Q_{ss'}^{\ell M\dot{\sigma}}(\boldsymbol{x},\boldsymbol{x}',t,t') + \hat{\bar{\xi}}_{Qs}^\ell(\boldsymbol{x},t)\hat{\bar{\xi}}_{Qs'}^\ell(\boldsymbol{x}',t')K_{ss'}^{\ell M\dot{\sigma}}(\boldsymbol{x},\boldsymbol{x}',t,t')\right]\right)
$$

$$
\exp\left(-i\sum_{\ell t t' s s'} \int d\boldsymbol{x}d\boldsymbol{x}'\left[\hat{\chi}_{Os}^\ell(\boldsymbol{x},t)g_{s'}^{\ell+1}(\boldsymbol{x}',t')\bar{R}_{ss'}^{v^\ell \xi_O^\ell}(\boldsymbol{x},\boldsymbol{x}',t,t') + \hat{\bar{\xi}}_{Vs}^\ell(\boldsymbol{x},t)h_{s'}^\ell(\boldsymbol{x}',t')\bar{R}_{ss'}^{g_O^\ell \chi_V^\ell}(\boldsymbol{x},\boldsymbol{x}',t,t')\right]\right)
$$

$$
\exp\left(-i\sum_{\ell t t' s s'} \int d\boldsymbol{x}d\boldsymbol{x}'\left[\hat{\bar{\xi}}_{Ks}^\ell(\boldsymbol{x},t)h_{s'}^{\ell+1}(\boldsymbol{x}',t')\bar{R}_{ss'}^{q^\ell \chi_K^\ell}(\boldsymbol{x},\boldsymbol{x}',t,t') + \hat{\bar{\xi}}_{Qs}^\ell(\boldsymbol{x},t)h_{s'}^\ell(\boldsymbol{x}',t')\bar{R}_{ss'}^{k^\ell \chi_Q^\ell}(\boldsymbol{x},\boldsymbol{x}',t,t')\right]\right)
$$

$$
\exp\left(i\sum_{\ell t s} \int d\boldsymbol{x}\left[\hat{\chi}_{Os}^\ell(\boldsymbol{x},t)\bar{\chi}_{Os}^\ell(\boldsymbol{x},t) + \hat{\bar{\xi}}_{Vs}^\ell(\boldsymbol{x},t)\bar{\xi}_{Vs}^\ell(\boldsymbol{x},t)\right]\right)
$$

$$
\exp\left(i\sum_{\ell t s} \int d\boldsymbol{x}\left[\hat{\bar{\xi}}_{Ks}^\ell(\boldsymbol{x},t)\bar{\xi}_{Ks}^\ell(\boldsymbol{x},t) + \hat{\bar{\xi}}_{Qs}^\ell(\boldsymbol{x},t)\bar{\xi}_{Qs}^\ell(\boldsymbol{x},t)\right]\right) \tag{82}
$$

We can express the MHSA single-head partition functions $\mathcal{Z}_{\text{MHSA}}$ in terms of the remaining order parameters within each head that will no longer concentrate at finite $N$

$$
\boldsymbol{Q}_{\text{MHSA}}^\ell = \{\mathcal{A}^\ell, M^\ell, Q^\ell, K^\ell, V^\ell, G_O^\ell, \hat{\mathcal{A}}^\ell, \hat{M}^\ell, \hat{Q}^\ell, \hat{K}^\ell, \hat{V}^\ell, \hat{G}_O^\ell\} \tag{83}
$$

After introducing these order parameters, we have

$$
\mathcal{Z}_{\text{MHSA}} = \int d\boldsymbol{Q}_{\text{MHSA}}^\ell \exp\left(NS_{\text{MHSA}}(\boldsymbol{Q}_{\text{MHSA}}^\ell)\right)
$$

$$
S_{\text{MHSA}} = \sum_{t t' s s'} \int d\boldsymbol{x}d\boldsymbol{x}'[Q_{ss'}^\ell(\boldsymbol{x},\boldsymbol{x}',t,t')\hat{Q}_{ss'}^\ell(\boldsymbol{x},\boldsymbol{x}',t,t') + K_{ss'}^\ell(\boldsymbol{x},\boldsymbol{x}',t,t')\hat{K}_{ss'}^\ell(\boldsymbol{x},\boldsymbol{x}',t,t')]
$$

$$
+ \sum_{t t' s s'} \int d\boldsymbol{x}d\boldsymbol{x}'[V_{ss'}^\ell(\boldsymbol{x},\boldsymbol{x}',t,t')\hat{V}_{ss'}^\ell(\boldsymbol{x},\boldsymbol{x}',t,t') + G_{Oss'}^\ell(\boldsymbol{x},\boldsymbol{x}',t,t')\hat{G}_{Oss'}^\ell(\boldsymbol{x},\boldsymbol{x}',t,t')]
$$

$$
+ \sum_{t s s'} \int d\boldsymbol{x}[\mathcal{A}_{ss'}^\ell(\boldsymbol{x},t)\hat{\mathcal{A}}_{ss'}^\ell(\boldsymbol{x},t) + M_{ss'}^\ell(\boldsymbol{x},t)\hat{M}_{ss'}^\ell(\boldsymbol{x},t)] + \ln \mathcal{Z}_{qkv}^\ell \tag{84}
$$

where the key-query-value single site partition function has the form

$$\mathcal{Z}_{qkv}^{\ell} = \int \prod_{\mathfrak{s}xt} \frac{d\hat{\xi}_{O\mathfrak{s}}^{\ell}(\boldsymbol{x},t)d\xi_{O\mathfrak{s}}^{\ell}(\boldsymbol{x},t)}{2\pi} \frac{d\hat{\chi}_{V\mathfrak{s}}^{\ell}(\boldsymbol{x},t)d\chi_{V\mathfrak{s}}^{\ell}(\boldsymbol{x},t)}{2\pi} \frac{d\hat{\chi}_{K\mathfrak{s}}^{\ell}(\boldsymbol{x},t)d\chi_{K\mathfrak{s}}^{\ell}(\boldsymbol{x},t)}{2\pi} \frac{d\hat{\chi}_{Q\mathfrak{s}}^{\ell}(\boldsymbol{x},t)d\chi_{Q\mathfrak{s}}^{\ell}(\boldsymbol{x},t)}{2\pi}$$

$$\exp\left(-\sum_{tt'\mathfrak{s}\mathfrak{s}'} \int d\boldsymbol{x}d\boldsymbol{x}'\hat{Q}_{\mathfrak{s}\mathfrak{s}'}^{\ell}(\boldsymbol{x},\boldsymbol{x}',t,t')q_{\mathfrak{s}}^{\ell}(\boldsymbol{x},t)q_{\mathfrak{s}'}^{\ell}(\boldsymbol{x}',t')\right)$$

$$\exp\left(-\sum_{tt'\mathfrak{s}\mathfrak{s}'} \int d\boldsymbol{x}d\boldsymbol{x}'\hat{K}_{\mathfrak{s}\mathfrak{s}'}^{\ell}(\boldsymbol{x},\boldsymbol{x}',t,t')k_{\mathfrak{s}}^{\ell}(\boldsymbol{x},t)k_{\mathfrak{s}'}^{\ell}(\boldsymbol{x}',t')\right)$$

$$\exp\left(-\sum_{tt'\mathfrak{s}\mathfrak{s}'} \int d\boldsymbol{x}d\boldsymbol{x}'\hat{V}_{\mathfrak{s}\mathfrak{s}'}^{\ell}(\boldsymbol{x},\boldsymbol{x}',t,t')v_{\mathfrak{s}}^{\ell}(\boldsymbol{x},t)v_{\mathfrak{s}'}^{\ell}(\boldsymbol{x}',t')\right)$$

$$\exp\left(-\sum_{tt'\mathfrak{s}\mathfrak{s}'} \int d\boldsymbol{x}d\boldsymbol{x}'\hat{V}_{\mathfrak{s}\mathfrak{s}'}^{\ell\sigma}(\boldsymbol{x},\boldsymbol{x}',t,t')v_{\mathfrak{s}}^{\ell\sigma}(\boldsymbol{x},t)v_{\mathfrak{s}'}^{\ell\sigma}(\boldsymbol{x}',t')\right)$$

$$\exp\left(-\sum_{tt'\mathfrak{s}\mathfrak{s}'} \int d\boldsymbol{x}d\boldsymbol{x}'\hat{G}_{O\mathfrak{s}\mathfrak{s}'}^{\ell\sigma}(\boldsymbol{x},\boldsymbol{x}',t,t')g_{O\mathfrak{s}}^{\ell\sigma}(\boldsymbol{x},t)g_{O\mathfrak{s}'}^{\ell\sigma}(\boldsymbol{x}',t')\right)$$

$$\exp\left(-\sum_{tt'\mathfrak{s}\mathfrak{s}'} \int d\boldsymbol{x}d\boldsymbol{x}'\hat{K}_{\mathfrak{s}\mathfrak{s}'}^{\ell M\dot{\sigma}}(\boldsymbol{x},\boldsymbol{x}',t,t')k_{\mathfrak{s}}^{\ell M\dot{\sigma}}(\boldsymbol{x},t)k_{\mathfrak{s}'}^{\ell M\dot{\sigma}}(\boldsymbol{x}',t')\right)$$

$$\exp\left(-\sum_{tt'\mathfrak{s}\mathfrak{s}'} \int d\boldsymbol{x}d\boldsymbol{x}'\hat{Q}_{\mathfrak{s}\mathfrak{s}'}^{\ell M\dot{\sigma}}(\boldsymbol{x},\boldsymbol{x}',t,t')q_{\mathfrak{s}}^{\ell M\dot{\sigma}}(\boldsymbol{x},t)q_{\mathfrak{s}'}^{\ell M\dot{\sigma}}(\boldsymbol{x}',t')\right)$$

$$\exp\left(-\sum_{t\mathfrak{s}\mathfrak{s}'} \int d\boldsymbol{x}\hat{\mathcal{A}}_{\mathfrak{s}\mathfrak{s}'}^{\ell}(\boldsymbol{x},t)k_{\mathfrak{s}}^{\ell}(\boldsymbol{x},t)q_{\mathfrak{s}'}^{\ell}(\boldsymbol{x},t)\right)$$

$$\exp\left(-\frac{1}{2}\sum_{tt'\mathfrak{s}\mathfrak{s}'} \int d\boldsymbol{x}d\boldsymbol{x}'\hat{\xi}_{O\mathfrak{s}}^{\ell}(\boldsymbol{x},t)\hat{\xi}_{O\mathfrak{s}'}^{\ell}(\boldsymbol{x}',t')G_{\mathfrak{s}\mathfrak{s}'}^{\ell+1}(\boldsymbol{x},\boldsymbol{x}',t,t')\right)$$

$$\exp\left(-\frac{1}{2}\sum_{tt'\mathfrak{s}\mathfrak{s}'} \int d\boldsymbol{x}d\boldsymbol{x}'\hat{\chi}_{V\mathfrak{s}}^{\ell}(\boldsymbol{x},t)\hat{\chi}_{V\mathfrak{s}}^{\ell}(\boldsymbol{x},t)H_{\mathfrak{s}\mathfrak{s}'}^{\ell}(\boldsymbol{x},\boldsymbol{x}',t,t')\right)$$

$$\exp\left(-\frac{1}{2}\sum_{tt'\mathfrak{s}\mathfrak{s}'} \int d\boldsymbol{x}d\boldsymbol{x}'\hat{\chi}_{K\mathfrak{s}}^{\ell}(\boldsymbol{x},t)\hat{\chi}_{K\mathfrak{s}}^{\ell}(\boldsymbol{x},t)H_{\mathfrak{s}\mathfrak{s}'}^{\ell}(\boldsymbol{x},\boldsymbol{x}',t,t')\right)$$

$$\exp\left(-\frac{1}{2}\sum_{tt'\mathfrak{s}\mathfrak{s}'} \int d\boldsymbol{x}d\boldsymbol{x}'\hat{\chi}_{Q\mathfrak{s}}^{\ell}(\boldsymbol{x},t)\hat{\chi}_{Q\mathfrak{s}}^{\ell}(\boldsymbol{x},t)H_{\mathfrak{s}\mathfrak{s}'}^{\ell}(\boldsymbol{x},\boldsymbol{x}',t,t')\right)$$

$$\exp\left(i\sum_{t\mathfrak{s}} \int d\boldsymbol{x}\,\hat{\xi}_{O\mathfrak{s}}^{\ell}(\boldsymbol{x},t)\xi_{O\mathfrak{s}}^{\ell}(\boldsymbol{x},t) + \hat{\chi}_{V\mathfrak{s}}^{\ell}(\boldsymbol{x},t)\chi_{V\mathfrak{s}}^{\ell}(\boldsymbol{x},t)\right)$$

$$\exp\left(i\sum_{t\mathfrak{s}} \int d\boldsymbol{x}\,\hat{\chi}_{Q\mathfrak{s}}^{\ell}(\boldsymbol{x},t)\chi_{Q\mathfrak{s}}^{\ell}(\boldsymbol{x},t) + \hat{\chi}_{K\mathfrak{s}}^{\ell}(\boldsymbol{x},t)\chi_{K\mathfrak{s}}^{\ell}(\boldsymbol{x},t)\right)$$

$$\exp\left(-i\sum_{tt'\mathfrak{s}\mathfrak{s}'\mathfrak{s}''} \int d\boldsymbol{x}d\boldsymbol{x}'\bar{R}_{\mathfrak{s}\mathfrak{s}'}^{g^{\ell+1}\bar{\chi}_O^{\ell}}(\boldsymbol{x},\boldsymbol{x}',t,t')\hat{\xi}_{O\mathfrak{s}}^{\ell}(\boldsymbol{x},t)v_{\mathfrak{s}''}^{\ell}(\boldsymbol{x}',t')\sigma_{\mathfrak{s}\mathfrak{s}'\mathfrak{s}''}^{\ell}(\boldsymbol{x}',t')\right)$$

$$\exp\left(-i\sum_{tt'\mathfrak{s}\mathfrak{s}'} \int d\boldsymbol{x}d\boldsymbol{x}'R_{\mathfrak{h}\mathfrak{s}\mathfrak{s}'}^{h^{\ell},\bar{\xi}_V^{\ell}}(\boldsymbol{x},\boldsymbol{x}',t,t')\hat{\chi}_{V\mathfrak{h}\mathfrak{s}}^{\ell}(\boldsymbol{x},t)g_{O\mathfrak{h}\mathfrak{s}'}^{\ell}(\boldsymbol{x}',t')\right)$$

$$\exp\left(-i\sum_{\mathfrak{h}tt'\mathfrak{s}\mathfrak{s}'} \int d\boldsymbol{x}d\boldsymbol{x}'R_{\mathfrak{h}\mathfrak{s}\mathfrak{s}'}^{h^{\ell}\bar{\xi}_Q^{\ell}}(\boldsymbol{x},\boldsymbol{x}',t,t')\hat{\chi}_{Q\mathfrak{h}\mathfrak{s}}^{\ell}(\boldsymbol{x},t)k_{\mathfrak{h}\mathfrak{s}'}^{\ell}(\boldsymbol{x}',t')\right)$$

$$\exp\left(-i\sum_{\mathfrak{h}tt'\mathfrak{s}\mathfrak{s}'} \int d\boldsymbol{x}d\boldsymbol{x}'R_{\mathfrak{h}\mathfrak{s}\mathfrak{s}'}^{h^{\ell}\bar{\xi}_K^{\ell}}(\boldsymbol{x},\boldsymbol{x}',t,t')\hat{\chi}_{K\mathfrak{h}\mathfrak{s}}^{\ell}(\boldsymbol{x},t)q_{\mathfrak{h}\mathfrak{s}'}^{\ell}(\boldsymbol{x}',t')\right) \qquad (85)$$

The saddle point equations for this limit are computed as derivatives with respect to $\boldsymbol{Q}_{\text{global}}$ *only*, reflecting that head-averages will converge as $\mathcal{H} \to \infty$

$$\frac{\partial S}{\partial \boldsymbol{Q}_{\text{global}}} = 0.\tag{86}$$

The final saddle point equations are given in terms of averages over the distribution of heads defined by $\mathcal{Z}_{\text{MHSA}}$ which we denote as $\langle \cdot \rangle_{\text{MHSA}}$ as well as averages over the residual stream which we denote as $\langle \cdot \rangle$.

These equations give the following (we suppress the sequence indices to simplify the final expressions)

$$H^\ell(\boldsymbol{x}, \boldsymbol{x}', t, t') = \left\langle h^\ell(\boldsymbol{x}, t) h^\ell(\boldsymbol{x}', t') \right\rangle \; , \; G^\ell(\boldsymbol{x}, \boldsymbol{x}', t, t') = \left\langle g^\ell(\boldsymbol{x}, t) g^\ell(\boldsymbol{x}', t') \right\rangle$$

$$V^{\ell\sigma}(\boldsymbol{x}, \boldsymbol{x}') = \left\langle V^\ell(\boldsymbol{x}, \boldsymbol{x}', t, t') \sigma^\ell(\boldsymbol{x}, t) \sigma^\ell(\boldsymbol{x}', t') \right\rangle_{\text{MHSA}}$$

$$G^{\ell\sigma}_{O\mathfrak{s}\mathfrak{s}'}(\boldsymbol{x}, \boldsymbol{x}') = \left\langle G^\ell_O(\boldsymbol{x}, \boldsymbol{x}', t, t') \sigma^\ell(\boldsymbol{x}, t) \sigma^\ell(\boldsymbol{x}', t') \right\rangle_{\text{MHSA}}$$

$$K^{\ell M\dot\sigma}(\boldsymbol{x}, \boldsymbol{x}') = \left\langle K^\ell(\boldsymbol{x}, \boldsymbol{x}', t, t') M^\ell(\boldsymbol{x}, t) M^\ell(\boldsymbol{x}', t') \dot\sigma^\ell(\boldsymbol{x}, t) \dot\sigma^\ell(\boldsymbol{x}', t') \right\rangle_{\text{MHSA}}$$

$$Q^{\ell M\dot\sigma}(\boldsymbol{x}, \boldsymbol{x}') = \left\langle Q^\ell(\boldsymbol{x}, \boldsymbol{x}', t, t') M^\ell(\boldsymbol{x}, t) M^\ell(\boldsymbol{x}', t') \dot\sigma^\ell(\boldsymbol{x}, t) \dot\sigma^\ell(\boldsymbol{x}', t') \right\rangle_{\text{MHSA}}$$

$$R^{g^{\ell+1}, \bar\chi^\ell_O}_{\mathfrak{s}\mathfrak{s}'}(\boldsymbol{x}, \boldsymbol{x}', t, t') = \left\langle \frac{\delta g^{\ell+1}_\mathfrak{s}(\boldsymbol{x}, t)}{\delta \bar u^\ell_{O\mathfrak{s}'}(\boldsymbol{x}', t')} \right\rangle$$

$$R^{h^\ell, \bar\xi^\ell_V}_{\mathfrak{s}\mathfrak{s}'}(\boldsymbol{x}, \boldsymbol{x}', t, t') = \left\langle \frac{\delta h^\ell_\mathfrak{s}(\boldsymbol{x}, t)}{\delta \bar r^\ell_{V\mathfrak{s}'}(\boldsymbol{x}', t')} \right\rangle$$

$$R^{h^\ell, \xi^\ell_K}_{\mathfrak{s}\mathfrak{s}'}(\boldsymbol{x}, \boldsymbol{x}', t, t') = \left\langle \frac{\delta h^\ell_\mathfrak{s}(\boldsymbol{x}, t)}{\delta \bar r^\ell_{K\mathfrak{s}'}(\boldsymbol{x}', t')} \right\rangle$$

$$R^{h^\ell, \xi^\ell_Q}_{\mathfrak{s}\mathfrak{s}'}(\boldsymbol{x}, \boldsymbol{x}', t, t') = \left\langle \frac{\delta h^\ell_\mathfrak{s}(\boldsymbol{x}, t)}{\delta \bar r^\ell_{Q\mathfrak{s}'}(\boldsymbol{x}', t')} \right\rangle$$

$$\bar R^{v^\ell \hat\xi^\ell_O}_{\mathfrak{h}\mathfrak{s}\mathfrak{s}'}(\boldsymbol{x}, \boldsymbol{x}', t, t') = \frac{1}{N} \text{Tr} \left\langle \frac{\delta \boldsymbol{v}^{\ell\sigma}_\mathfrak{s}(\boldsymbol{x}, t)}{\delta \boldsymbol{r}^\ell_{O\mathfrak{s}'}(\boldsymbol{x}', t')^\top} \right\rangle_{\text{MHSA}}$$

$$\bar R^{g^\ell_O \chi^\ell_V}_{\mathfrak{s}\mathfrak{s}'}(\boldsymbol{x}, \boldsymbol{x}', t, t') = \frac{1}{N} \text{Tr} \left\langle \frac{\delta \boldsymbol{g}^{\ell\sigma}_{O\mathfrak{h}\mathfrak{s}}(\boldsymbol{x}, t)}{\delta \boldsymbol{u}^\ell_{V\mathfrak{h}\mathfrak{s}'}(\boldsymbol{x}', t')^\top} \right\rangle_{\text{MHSA}}$$

$$\bar R^{k^\ell \chi^\ell_Q}_{\mathfrak{s}\mathfrak{s}'}(\boldsymbol{x}, \boldsymbol{x}', t, t') = \frac{1}{N} \text{Tr} \left\langle \frac{\delta \boldsymbol{k}^{\ell M\sigma}_\mathfrak{s}(\boldsymbol{x}, t)}{\delta \boldsymbol{u}^\ell_{Q\mathfrak{s}'}(\boldsymbol{x}', t')^\top} \right\rangle_{\text{MHSA}}$$

$$\bar R^{q^\ell \chi^\ell_K}_{\mathfrak{s}\mathfrak{s}'}(\boldsymbol{x}, \boldsymbol{x}', t, t') = \frac{1}{N} \text{Tr} \left\langle \frac{\delta \boldsymbol{q}^\ell_{\mathfrak{h}\mathfrak{s}}(\boldsymbol{x}, t)}{\delta \boldsymbol{u}^\ell_{K\mathfrak{h}\mathfrak{s}'}(\boldsymbol{x}', t')^\top} \right\rangle_{\text{MHSA}}\tag{87}$$

**Residual Stream Dynamics**   The residual stream satisfies the following single-site dynamics

$$h^{\ell+1}_\mathfrak{s}(\boldsymbol{x}, t) = h^\ell_\mathfrak{s}(\boldsymbol{x}, t) + \frac{\beta_0}{L^{\alpha_L}} \bar u^\ell_{O\mathfrak{s}}(\boldsymbol{x}, t) + \frac{\beta_0}{L^\alpha} \sum_{t' \mathfrak{s}'} \int d\boldsymbol{x}' \bar R^{v^\ell \xi^\ell_O}_{\mathfrak{s}\mathfrak{s}'}(\boldsymbol{x}, \boldsymbol{x}', t, t') g^{\ell+1}_{\mathfrak{s}'}(\boldsymbol{x}', t')$$

$$+ \frac{\eta_0 \gamma_0 \beta_0^2}{L} \sum_{t' < t} \mathbb{E}_{\boldsymbol{x}' \sim \mathfrak{B}_{t'}} V^{\ell\sigma}_{\mathfrak{s}\mathfrak{s}'}(\boldsymbol{x}, \boldsymbol{x}', t, t') g^{\ell+1}_{\mathfrak{s}'}(\boldsymbol{x}', t') \; , \; \bar u^\ell_{O\mathfrak{s}}(\boldsymbol{x}, t) \sim \mathcal{GP}(0, V^{\ell\sigma})$$

$$g^\ell_\mathfrak{s}(\boldsymbol{x}, t) = g^{\ell+1}_\mathfrak{s}(\boldsymbol{x}, t) + \frac{\beta_0}{L^{\alpha_L}} \left[ \bar r^\ell_{V\mathfrak{s}}(\boldsymbol{x}, t) + \bar r^\ell_{K\mathfrak{s}}(\boldsymbol{x}, t) + \bar r^\ell_{Q\mathfrak{s}}(\boldsymbol{x}, t) \right]$$

$$+ \frac{\beta_0}{L^\alpha} \sum_{t' \mathfrak{s}'} \int d\boldsymbol{x}' [\bar R^{g^\ell_O \chi^\ell_V}_{\mathfrak{s}\mathfrak{s}'}(\boldsymbol{x}, \boldsymbol{x}', t, t') + \bar R^{k^\ell \chi^\ell_Q}_{\mathfrak{s}\mathfrak{s}'}(\boldsymbol{x}, \boldsymbol{x}', t, t') + \bar R^{q^\ell \chi^\ell_K}_{\mathfrak{s}\mathfrak{s}'}(\boldsymbol{x}, \boldsymbol{x}', t, t')] h^\ell_{\mathfrak{s}'}(\boldsymbol{x}', t')$$

$$+ \frac{\eta_0 \gamma_0 \beta_0^2}{L} \sum_{t' < t} \mathbb{E}_{\boldsymbol{x}' \sim \mathfrak{B}_{t'}} [G^{\ell\sigma}_{O\mathfrak{s}\mathfrak{s}'}(\boldsymbol{x}, \boldsymbol{x}', t, t') + K^{\ell M\dot\sigma}_{O\mathfrak{s}\mathfrak{s}'}(\boldsymbol{x}, \boldsymbol{x}', t, t') + Q^{\ell M\dot\sigma}_{O\mathfrak{s}\mathfrak{s}'}(\boldsymbol{x}, \boldsymbol{x}', t, t')] h^\ell_{\mathfrak{s}'}(\boldsymbol{x}', t')$$

$$\bar r^\ell_{V\mathfrak{s}}(\boldsymbol{x}, t) \sim \mathcal{GP}(0, G^{\ell\sigma}_O) \; , \; \bar r^\ell_{Q\mathfrak{s}}(\boldsymbol{x}, t) \sim \mathcal{GP}(0, Q^{\ell M\dot\sigma}) \; , \; \bar r^\ell_{K\mathfrak{s}}(\boldsymbol{x}, t) \sim \mathcal{GP}(0, Q^{\ell M\dot\sigma})\tag{88}$$

This matches the result provided in the main text which introduces a compressed

$$C^\ell_{\mathfrak{s}\mathfrak{s}'}(\boldsymbol{x}, \boldsymbol{x}', t, t') = \frac{1}{\eta_0 \gamma_0 \beta_0} \bar{R}^{v^\ell \xi^\ell_O}_{\mathfrak{s}\mathfrak{s}'}(\boldsymbol{x}, \boldsymbol{x}', t, t') + p_{t'}(\boldsymbol{x}')\Delta(\boldsymbol{x}', t')V^{\ell\sigma}_{\mathfrak{s}\mathfrak{s}'}(\boldsymbol{x}, \boldsymbol{x}', t, t') \qquad (89)$$

where $p_t(\boldsymbol{x}) = \frac{1}{|\mathfrak{B}_t|}\sum_{\boldsymbol{x}'\in\mathfrak{B}_t}\delta(\boldsymbol{x}-\boldsymbol{x}')$ denotes the uniform distribution over the batch $\mathfrak{B}_t$.

### E.4 Infinite $L$ Limits

In this section, we discuss the two large $L$ limits. This can be derived formally in two distinct ways. First, one could start with the initial For this section, it suffices to reason about the scale of the Gaussian noise which appears in the residual stream and the contribution from the response functions.

#### E.4.1 Basic Intuition

Deriving the infinite depth limits To gain intuition for the large $\mathcal{H}, L \to \infty$ limit, we use the fact that the random variables $\chi^\ell = u^\ell + R^\ell g^\ell$ decompose into a Gaussian $u^\ell$ which are uncorrelated across layers and a linear response $R^\ell$ are response functions. This implies that

$$h^L = \frac{\beta_0}{L^{\alpha_L}}\sum_{k=1}^L u^k + \frac{\beta_0}{L^{\alpha_L}}\sum_{k=1}^L R^k g^k + \frac{\eta_0\gamma_0\beta_0^2}{L}\sum_{k=1}^L V^{k\sigma}g^k \qquad (90)$$

We first note that the sum of the Gaussians is a zero-mean random variable with standard deviation

$$\frac{1}{L^\alpha}\sum_{k=1}^L u^k \sim \mathcal{O}(L^{\frac{1}{2}-\alpha_L}) \qquad (91)$$

Thus, this integrated random variable will vanish unless $\alpha_L = \frac{1}{2}$.

Next, we can investigate the scale of the residual stream response functions. For instance

$$\frac{\partial h^\ell}{\partial u^k} = \mathcal{O}\left(L^{-\alpha_L}\right), \quad \frac{\partial h^\ell}{\partial r^k} = \mathcal{O}\left(L^{-\alpha_L}\right), \quad \frac{\partial g^\ell}{\partial u^k} = \mathcal{O}\left(L^{-\alpha_L}\right), \quad \frac{\partial g^\ell}{\partial r^k} = \mathcal{O}\left(L^{-\alpha_L}\right)$$

$$\frac{1}{L^{\alpha_L}}\sum_{k=1}^\ell R^k g^k = \mathcal{O}\left(L^{1-2\alpha_L}\right) \qquad (92)$$

As a consequence, we see that the effect of the Gaussian and linear response terms will vanish as $L \to \infty$ provided that $\alpha_L > \frac{1}{2}$. We will consider first, the case where $\alpha = 1$ which gives an ODE like limit for the residual updates before moving onto the more involved $\alpha_L = \frac{1}{2}$ case.

To formally take the $L \to \infty$ limit, we redefine all of the preactivation fields and kernels in terms of layer time $\tau$ defined as

$$\tau = \lim_{L\to\infty}\frac{\ell}{L} \in [0, 1]. \qquad (93)$$

For example, the residual kernels are defined as

$$H_{\mathfrak{s}\mathfrak{s}'}(\tau, \boldsymbol{x}, \boldsymbol{x}', t, t') \equiv \lim_{L\to\infty} H^{L\tau}_{\mathfrak{s}\mathfrak{s}'}(\boldsymbol{x}, \boldsymbol{x}', t, t'). \qquad (94)$$

The finite difference equations for the residual updates $L(h^{\ell+1} - h^\ell) \sim \frac{\partial}{\partial\tau}h(\tau)$ become differential updates (either SDE-like or ODE-like depending on $\alpha_L$) [46, 11].

#### E.4.2 ODE Limit $\alpha_L = 1$

First, we investigate the case of $\alpha_L = 1$. In this case, the $\mathcal{H}, L \to \infty$ limit results in a complete disappearance of the $\bar{\chi}^\ell_O, \bar{\xi}^\ell_V, \bar{\xi}^\ell_K, \bar{\xi}^\ell_Q$ fields.

$$\partial_\tau h_{\mathfrak{s}}(\tau, \boldsymbol{x}, t) = \eta_0\gamma_0\beta_0^2 \sum_{t'<t}\mathbb{E}_{\boldsymbol{x}'\sim\mathfrak{B}_t}\Delta(\boldsymbol{x}', t')V^\sigma_{\mathfrak{s}\mathfrak{s}'}(\tau, \boldsymbol{x}, \boldsymbol{x}', t, t')g_{\mathfrak{s}'}(\tau', \boldsymbol{x}', t')$$

$$-\partial_\tau g_{\mathfrak{s}}(\tau, \boldsymbol{x}, t) = \eta_0\gamma_0\beta_0^2 \sum_{t'<t}\mathbb{E}_{\boldsymbol{x}'\sim\mathfrak{B}_t}\Delta(\boldsymbol{x}', t')[G^\sigma_{\mathfrak{s}\mathfrak{s}'}(\tau, \boldsymbol{x}, \boldsymbol{x}', t, t') + K^{M\dot{\sigma}}_{\mathfrak{s}\mathfrak{s}'}(\tau, \boldsymbol{x}, \boldsymbol{x}', t, t')]g_{\mathfrak{s}'}(\tau', \boldsymbol{x}', t')$$

$$+\eta_0\gamma_0\beta_0^2 \sum_{t'<t}\mathbb{E}_{\boldsymbol{x}'\sim\mathfrak{B}_t}\Delta(\boldsymbol{x}', t')[Q^{M\dot{\sigma}}_{\mathfrak{s}\mathfrak{s}'}(\tau, \boldsymbol{x}, \boldsymbol{x}', t, t')]g_{\mathfrak{s}'}(\tau', \boldsymbol{x}', t') \qquad (95)$$

### E.4.3 SDE Limit $\alpha_L = \frac{1}{2}$

This $\alpha_L = \frac{1}{2}$ limit is more technically involved since neither the Gaussian terms from the DMFT nor the response functions vanish. For the Gaussian terms, we note that the sums of the independent Gaussians are all multiplied by $\frac{1}{\sqrt{L}} \sim \sqrt{d\tau}$, which can be interpreted as integrated Brownian motion in the limit

$$\lim_{L \to \infty} \frac{1}{\sqrt{L}} \sum_{k=1}^{\ell} u^k \to \int_0^\tau du(\tau')$$

$$\langle du(\tau) du(\tau') \rangle = V^\sigma(\tau) \delta(\tau - \tau') d\tau d\tau' \tag{96}$$

Following the derivation of Bordelon et al. [11], which maintains the exact dependence on the full integrated response and provides the result as an integrated SDE

$$h_{\mathfrak{s}}(\tau, \boldsymbol{x}, t) = \beta_0 \int_0^\tau d\bar{u}_{\mathfrak{s}}(\tau' \boldsymbol{x}, t) + \eta_0 \gamma_0 \beta_0^2 \sum_{t' < t} \mathbb{E}_{\boldsymbol{x}' \sim \mathfrak{B}_{t'}} \int_0^\tau d\tau' C_{\mathfrak{s}\mathfrak{s}'}(\tau, \boldsymbol{x}, \boldsymbol{x}', t, t') g_{\mathfrak{s}'}(\tau', \boldsymbol{x}', t')$$

$$C_{\mathfrak{s}\mathfrak{s}'}(\tau, \boldsymbol{x}, \boldsymbol{x}', t, t') = \frac{1}{\eta_0 \gamma_0 \beta_0} \bar{R}_{\mathfrak{s}\mathfrak{s}'}^{v\xi_O}(\tau, \boldsymbol{x}, \boldsymbol{x}', t, t') + p_{t'}(\boldsymbol{x}') \Delta(\boldsymbol{x}', t') V_{\mathfrak{s}\mathfrak{s}'}^\sigma(\tau, \boldsymbol{x}, \boldsymbol{x}', t, t'). \tag{97}$$

Combining the forward pass equations from the previous two subsection recovers the Result 3 of the main text. This is combined with a complementary equation for the backward pass.

### E.5 Effect of MLP Layers

Adding the MLP block to the residual stream can also be easily handled with the methods of the preceeding sections. The forward pass equations in this case take the form

$$\tilde{\boldsymbol{h}}_{\mathfrak{s}}^\ell(\boldsymbol{x}, t) = \boldsymbol{h}_{\mathfrak{s}}^\ell(\boldsymbol{x}, t) + \frac{\beta_0}{L^{\alpha_L}} \text{MHSA} \left( \boldsymbol{h}^\ell(\boldsymbol{x}, t) \right)_{\mathfrak{s}} \tag{98}$$

$$\boldsymbol{h}^{\ell+1}(\boldsymbol{x}, t) = \tilde{\boldsymbol{h}}_{\mathfrak{s}}^\ell(\boldsymbol{x}, t) + \frac{\beta_0}{L^{\alpha_L}} \text{MLP} \left( \tilde{\boldsymbol{h}}_{\mathfrak{s}}^\ell(\boldsymbol{x}, t) \right), \tag{99}$$

where the MLP layer is

$$\text{MLP}(\tilde{\boldsymbol{h}}_{\mathfrak{s}}^\ell) = \frac{1}{\sqrt{N\mathcal{H}}} \boldsymbol{W}^{\ell,2} \phi \left( \tilde{\boldsymbol{h}}_{\mathfrak{s}}^{\ell,1} \right) , \quad \tilde{\boldsymbol{h}}^{\ell,1} = \frac{1}{\sqrt{N\mathcal{H}}} \boldsymbol{W}^{\ell,1} \tilde{\boldsymbol{h}}_{\mathfrak{s}}^\ell \tag{100}$$

The following gradient fields are necessary

$$\boldsymbol{g}_{\mathfrak{s}}^\ell(\boldsymbol{x}, t) \equiv \gamma_0 N\mathcal{H} \frac{\partial f(\boldsymbol{x}, t)}{\partial \boldsymbol{h}_{\mathfrak{s}}^\ell(\boldsymbol{x}, t)} , \quad \tilde{\boldsymbol{g}}_{\mathfrak{s}}^\ell(\boldsymbol{x}, t) \equiv \gamma_0 N\mathcal{H} \frac{\partial f(\boldsymbol{x}, t)}{\partial \tilde{\boldsymbol{h}}_{\mathfrak{s}}^\ell(\boldsymbol{x}, t)}$$

$$\tilde{\boldsymbol{g}}_{\mathfrak{s}}^{\ell,1}(\boldsymbol{x}, t) \equiv \gamma_0 N\mathcal{H} \frac{\partial f(\boldsymbol{x}, t)}{\partial \tilde{\boldsymbol{h}}_{\mathfrak{s}}^{\ell,1}(\boldsymbol{x}, t)} \tag{101}$$

$$\boldsymbol{W}^{\ell,2}(t) = \boldsymbol{W}^{\ell,2}(0) + \frac{\beta_0 \eta_0 \gamma_0}{L^{1-\alpha_L} \sqrt{N\mathcal{H}}} \sum_{t' < t} \mathbb{E}_{\boldsymbol{x} \sim \mathfrak{B}_{t'}} \sum_{\mathfrak{s}} \Delta(\boldsymbol{x}, t') \boldsymbol{g}_{\mathfrak{s}}^{\ell+1}(\boldsymbol{x}, t') \phi(\tilde{\boldsymbol{h}}_{\mathfrak{s}}^{\ell,1}(\boldsymbol{x}, t'))^\top$$

$$\boldsymbol{W}^{\ell,1}(t) = \boldsymbol{W}^{\ell,1}(0) + \frac{\beta_0 \eta_0 \gamma_0}{L^{1-\alpha_L} \sqrt{N\mathcal{H}}} \sum_{t' < t} \mathbb{E}_{\boldsymbol{x} \sim \mathfrak{B}_{t'}} \sum_{\mathfrak{s}} \Delta(\boldsymbol{x}, t') \tilde{\boldsymbol{g}}_{\mathfrak{s}}^{\ell,1}(\boldsymbol{x}, t') \bar{\boldsymbol{h}}_{\mathfrak{s}}^\ell(\boldsymbol{x}, t')^\top \tag{102}$$

The MLP hidden layer dynamics is much simpler to characterize and resembles the structure analyzed in prior works on infinite width networks [15].

$$\tilde{\boldsymbol{h}}_{\mathfrak{s}}^{\ell,1}(\boldsymbol{x}, t) = \tilde{\boldsymbol{\chi}}_{\mathfrak{s}}^{\ell,1}(\boldsymbol{x}, t) + \frac{\beta_0 \eta_0 \gamma_0}{L^{1-\alpha_L}} \sum_{t' < t} \mathbb{E}_{\boldsymbol{x} \sim \mathfrak{B}_{t'}} \sum_{\mathfrak{s}} \Delta(\boldsymbol{x}, t') \tilde{\boldsymbol{g}}_{\mathfrak{s}}^{\ell,1}(\boldsymbol{x}, t') H_{\mathfrak{s}\mathfrak{s}'}^\ell(\boldsymbol{x}, \boldsymbol{x}', t, t') \tag{103}$$

Again, we see that the inner dynamics for $\tilde{\boldsymbol{h}}^{\ell,1}$ due to the weight updates in this layer scale as $L^{-1+\alpha_L}$, suggesting the need to choose $\alpha_L = 1$ if we desire this hidden layer to contribute to the representational updates.

**MLP Layer Gradients**  For the MLP layer we have the simpler backpropagation equations

$$\tilde{g}_{\mathfrak{s}}^{\ell,1}(\boldsymbol{x},t) = \left(\frac{\partial \boldsymbol{h}_{\mathfrak{s}}^{\ell+1}(\boldsymbol{x},t)}{\partial \tilde{\boldsymbol{h}}_{\mathfrak{s}}^{\ell,1}(\boldsymbol{x},t)}\right)^{\top} \boldsymbol{g}_{\mathfrak{s}}^{\ell+1}(\boldsymbol{x},t)$$

$$= \frac{\beta_0}{L_L^{\alpha}} \dot{\phi}(\tilde{\boldsymbol{h}}_{\mathfrak{s}}^{\ell,1}(\boldsymbol{x},t)) \odot \left[\frac{1}{\sqrt{N\mathcal{H}}} \boldsymbol{W}^{\ell,2}(t)^{\top} \boldsymbol{g}_{\mathfrak{s}}^{\ell+1}(\boldsymbol{x},t)\right]$$

$$\tilde{g}_{\mathfrak{s}}^{\ell}(\boldsymbol{x},t) = \frac{1}{\sqrt{N\mathcal{H}}} \boldsymbol{W}^{\ell,1}(t)^{\top} \tilde{\boldsymbol{g}}_{\mathfrak{s}}^{\ell,1}(\boldsymbol{x},t) \tag{104}$$

The components of these fields that depend on initial conditions are

$$\boldsymbol{\xi}_{\mathfrak{s}}^{\ell,1}(\boldsymbol{x},t) = \frac{1}{\sqrt{N\mathcal{H}}} \boldsymbol{W}^{\ell,2}(0)^{\top} \boldsymbol{g}_{\mathfrak{s}}^{\ell+1}(\boldsymbol{x},t)$$

$$\boldsymbol{\xi}_{\mathfrak{s}}^{\ell}(\boldsymbol{x},t) = \frac{1}{\sqrt{N\mathcal{H}}} \boldsymbol{W}^{\ell,1}(0)^{\top} \boldsymbol{g}_{\mathfrak{s}}^{\ell,1}(\boldsymbol{x},t) \tag{105}$$

**MLP Matrices**  After utilizing these resolutions of the identity for all $\mathfrak{s}, \boldsymbol{x}, t$, we can integrate over the weights $\boldsymbol{W}^{\ell,2}(0)$

$$\ln \mathbb{E}_{\boldsymbol{W}^{\ell,2}(0)} \exp\left(-\frac{i}{\sqrt{N\mathcal{H}}} \sum_{t\mathfrak{s}} \int d\boldsymbol{x}\, \mathrm{Tr} \boldsymbol{W}^{\ell,2}(0)^{\top} \left[\hat{\boldsymbol{\chi}}_{\mathfrak{s}}^{\ell+1}(\boldsymbol{x},t)\phi(\tilde{\boldsymbol{h}}_{\mathfrak{s}}^{\ell,1}(\boldsymbol{x},t))^{\top} + \boldsymbol{g}_{\mathfrak{s}}^{\ell+1}(\boldsymbol{x},t)\hat{\boldsymbol{\xi}}_{\mathfrak{s}}^{\ell,1}(\boldsymbol{x},t)^{\top}\right]\right)$$

$$= -\frac{1}{2} \sum_{tt'\mathfrak{s}\mathfrak{s}'} \int d\boldsymbol{x} d\boldsymbol{x}' \left[\hat{\boldsymbol{\chi}}_{\mathfrak{s}}^{\ell+1}(\boldsymbol{x},t) \cdot \hat{\boldsymbol{\chi}}_{\mathfrak{s}'}^{\ell+1}(\boldsymbol{x}',t')\Phi_{\mathfrak{s}\mathfrak{s}'}^{\ell,1}(\boldsymbol{x},\boldsymbol{x}',t,t') + \hat{\boldsymbol{\xi}}_{\mathfrak{s}}^{\ell,1}(\boldsymbol{x},t) \cdot \hat{\boldsymbol{\xi}}_{\mathfrak{s}'}^{\ell,1}(\boldsymbol{x}',t')G_{\mathfrak{s}\mathfrak{s}'}^{\ell+1}(\boldsymbol{x},\boldsymbol{x}',t,t')\right]$$

$$- i \sum_{tt'\mathfrak{s}\mathfrak{s}'} \int d\boldsymbol{x} d\boldsymbol{x}' \left[\hat{\boldsymbol{\chi}}_{\mathfrak{s}}^{\ell+1}(\boldsymbol{x},t) \cdot \boldsymbol{g}_{\mathfrak{s}'}^{\ell+1}(\boldsymbol{x}',t')R_{\mathfrak{s}\mathfrak{s}'}^{\ell,1}(\boldsymbol{x},\boldsymbol{x}',t,t')\right] \tag{106}$$

where we introduced the response function

$$R_{\mathfrak{s}\mathfrak{s}'}^{\ell,1}(\boldsymbol{x},\boldsymbol{x}',t,t') \equiv -\frac{i}{N\mathcal{H}}\phi\left(\tilde{\boldsymbol{h}}_{\mathfrak{s}}^{\ell,1}(\boldsymbol{x},t)\right) \cdot \hat{\boldsymbol{\xi}}_{\mathfrak{s}'}^{\ell,1}(\boldsymbol{x}',t') \tag{107}$$

We can perform an identical step to integrate over $\boldsymbol{W}^{\ell,1}(0)$. This gives us

$$\ln \mathbb{E}_{\boldsymbol{W}^{\ell,1}(0)} \exp\left(-\frac{i}{\sqrt{N\mathcal{H}}} \sum_{t\mathfrak{s}} \int d\boldsymbol{x}\, \mathrm{Tr} \boldsymbol{W}^{\ell,1}(0)^{\top} \left[\hat{\tilde{\boldsymbol{\chi}}}_{\mathfrak{s}}^{\ell,1}(\boldsymbol{x},t)\tilde{\boldsymbol{h}}_{\mathfrak{s}}^{\ell}(\boldsymbol{x},t)^{\top} + \boldsymbol{g}_{\mathfrak{s}}^{\ell,1}(\boldsymbol{x},t)\hat{\boldsymbol{\xi}}_{\mathfrak{s}}^{\ell}(\boldsymbol{x},t)^{\top}\right]\right)$$

$$= -\frac{1}{2} \sum_{tt'\mathfrak{s}\mathfrak{s}'} \int d\boldsymbol{x} d\boldsymbol{x}' \left[\hat{\tilde{\boldsymbol{\chi}}}_{\mathfrak{s}}^{\ell,1}(\boldsymbol{x},t) \cdot \hat{\tilde{\boldsymbol{\chi}}}_{\mathfrak{s}'}^{\ell,1}(\boldsymbol{x}',t')\tilde{H}_{\mathfrak{s}\mathfrak{s}'}^{\ell}(\boldsymbol{x},\boldsymbol{x}',t,t') + \hat{\boldsymbol{\xi}}_{\mathfrak{s}}^{\ell}(\boldsymbol{x},t) \cdot \hat{\boldsymbol{\xi}}_{\mathfrak{s}'}^{\ell}(\boldsymbol{x}',t')G_{\mathfrak{s}\mathfrak{s}'}^{\ell,1}(\boldsymbol{x},\boldsymbol{x}',t,t')\right]$$

$$- i \sum_{tt'\mathfrak{s}\mathfrak{s}'} \int d\boldsymbol{x} d\boldsymbol{x}' \left[\hat{\boldsymbol{\chi}}_{\mathfrak{s}}^{\ell,1}(\boldsymbol{x},t) \cdot \boldsymbol{g}_{\mathfrak{s}'}^{\ell+1}(\boldsymbol{x}',t')\tilde{R}_{\mathfrak{s}\mathfrak{s}'}^{\ell}(\boldsymbol{x},\boldsymbol{x}',t,t')\right] \tag{108}$$

where we introduced

$$\tilde{R}_{\mathfrak{s}\mathfrak{s}'}^{\ell}(\boldsymbol{x},\boldsymbol{x}',t,t') = -\frac{i}{N\mathcal{H}}\tilde{\boldsymbol{h}}_{\mathfrak{s}}^{\ell}(\boldsymbol{x},t) \cdot \hat{\boldsymbol{\xi}}_{\mathfrak{h}}^{\ell}(\boldsymbol{x}',t') \tag{109}$$

### E.6  Effect of Layer Norm on the Limiting Process

**Layernorm**  The derivative of layer-norm $\frac{\partial \bar{h}_{\mathfrak{s}'}^{\ell}}{\partial h_{\mathfrak{s}'}^{\ell\top}}$ acts as the following in the large $\mathcal{H}$ limit

$$\frac{\partial \bar{\boldsymbol{h}}}{\partial \boldsymbol{h}^{\top}} = \frac{1}{\sqrt{\sigma^2 + \epsilon}}\left(\boldsymbol{I} - \frac{1}{N\mathcal{H}}\boldsymbol{1}\boldsymbol{1}^{\top}\right) - \frac{1}{N\mathcal{H}}\frac{1}{(\sigma^2 + \epsilon)^{3/2}}[\boldsymbol{h} - \mu\boldsymbol{1}][\boldsymbol{h} - \mu\boldsymbol{1}]^{\top} \tag{110}$$

In the limit of $\mathcal{H} \to \infty$ the variables $\mu = \frac{1}{N\mathcal{H}}\boldsymbol{h} \cdot \boldsymbol{1}$ and $\sigma^2 = \frac{1}{N\mathcal{H}}|\boldsymbol{h} - \mu\boldsymbol{1}|^2$ will become deterministic over random initializations. We thus just have to consider how these types of vectors act on gradients

$$\left(\frac{\partial \bar{\boldsymbol{h}}}{\partial \boldsymbol{h}^{\top}}\right)^{\top} \boldsymbol{g} = \frac{1}{\sqrt{\sigma^2 + \epsilon}}(\boldsymbol{g} - \boldsymbol{1}\mu_g) - \frac{1}{(\sigma^2 + \epsilon)^{3/2}}[\boldsymbol{h} - \mu\boldsymbol{1}]\left(\frac{1}{N\mathcal{H}}[\boldsymbol{h} - \mu\boldsymbol{1}]^{\top}\boldsymbol{g}\right). \tag{111}$$

Each of these operations will lead to one inner product that will be self averaging $\mu_g = \frac{1}{N\mathcal{H}}\boldsymbol{1} \cdot \boldsymbol{g}$ or $\left(\frac{1}{N\mathcal{H}}[\boldsymbol{h} - \mu\boldsymbol{1}]^{\top}\boldsymbol{g}\right)$. Thus this operation will not alter the backward pass in terms of scaling.

## F  Compute Resources and Experimental Details

Each of the experimental runs performed in this paper were all performed on single NVIDIA H100 GPU. Each run of the full CIFAR-5M took anywhere from 5 minutes to 1 hour depending on model size. Each run of the C4 training took anywhere from 1 hour to 6 hours depending on model size and total amount of training steps.

The language model trained on C4 used a context length of 256 and the GPT-tokenizer from Huggingface. Sequences that were too short were concatenated with other sentences to reach the full context length rather than padding the end of the sequence. We use trainable positional encodings and separate embedding and decoding parameters as implemented in Appendix F.

