# OpenReview forum: "Infinite Limits of Multi-head Transformer Dynamics"
_NeurIPS.cc/2024/Conference — NeurIPS 2024 poster_

### Official Review · Reviewer_UGyb · 2024-07-02

**Soundness:** 3
**Presentation:** 2
**Contribution:** 3
**Rating:** 6
**Confidence:** 2

**Summary:**

The paper analyzes scaling limits of transformer models w.r.t. key-query dimension $N$, head count $H$ and depth $L$ using dynamical mean-field theory. For the $N\to\infty$ limit it is shown that $1/N$ scaling for the pre-attention scores is required for stable learning and all heads become degenerate. Conversely for the $H\to\infty$ limit the kernel of each head is shown to follow independent stochastic processes. For $L\to\infty$ a branch scaling of $1/L$ is shown to be required for feature evolution. The theoretical results are complemented by experiments on natural language datasets.

**Strengths:**

While I am not familiar with the specifics of DMFT, the paper seems to provide a solid analysis of a problem of practical importance - the effective scaling of various hyperparameters in large-scale limits, extending previous muP type works to the transformer architecture. The results theoretically establish appropriate scaling regimes which guarantee diverse kernels are generated across attention heads and are backed by heuristics and detailed experiments on vision and natural language transformers where necessary. The analysis is able to provide concrete descriptions and recommendations of scaling regimes despite the generality of the framework, for example the effects of MLP layers, learning rate scaling and LayerNorm are also accounted for in the appendix.

**Weaknesses:**

See Questions.

**Questions:**

* Besides specifying the scaling exponents which seem to be somewhat intuitive from the central limit theorem and law of large number type limits, can the explicit limiting kernel derivations (equations 5-10) be used to predict more detailed aspects of the training process?
* There should be discussion on the effectiveness of each scaling with respect to compute. For example in the experiments, which configuration was optimal in terms of flops and how well does the theory corroborate this?
* The distinction between $\alpha_\mathcal{A} = 1$ and $\alpha_\mathcal{A} = 1/2$ is explained in Appendix C.4, what happens if $\alpha_\mathcal{A}$ interpolates between the two extremes?
* What are some previous results or potential approaches towards analyzing limiting dynamics of neural networks under Adam?

**Limitations:**

A Limitations section is provided in Section 5.

---

> ### Author Rebuttal · Authors · 2024-08-06
>
> ### Strengths
>
> We thank the reviewer for their supportive comments and for their detailed reading of our work. Below we try addressing the questions and mentioning ways we aim to improve the paper.
>
> ### Questions
> *1. Besides specifying the scaling exponents which seem to be somewhat intuitive from the central limit theorem and law of large number type limits, can the explicit limiting kernel derivations (equations 5-10) be used to predict more detailed aspects of the training process?*
>
> This is a good question! The central limit theorem and law of large numbers do give you pretty good intuition in general and motivate the choice of scaling exponents. However, to make fine grained conclusions about the exact limiting dynamics one often needs to be careful with extra terms that appear in the dynamics known as response functions which are not apriori obvious (see comment below on non-trivial DMFT examples).
>
> One precise insight from our analysis is the symmetry across heads of the resulting limiting equations as key query dimension diverges $N \to \infty$. This led to our conclusion that heads degenerate into the same dynamics and motivated us to characterize the infinite head $\mathcal H \to \infty$ limit. We can also give some similar qualitative insights about the types of dynamics one obtains as $L \to \infty$ from our equations (for instance response functions are negligible for $\alpha_L = 1$ but survive for $\alpha_L = \frac{1}{2}$). However, the exact equations in the general case are complicated non-Gaussian stochastic processes. In **linear** transformers, we suspect these tools can give much more interpretable insights, but this case is not as realistic.
>
> *2. There should be discussion on the effectiveness of each scaling with respect to compute. For example in the experiments, which configuration was optimal in terms of flops and how well does the theory corroborate this?*
>
> We have added some preliminary comparisions of this kind. It does seem that in terms of width scaling, increasing $\mathcal H$ is preferable to increasing $N$ with $\alpha_{\mathcal A} = 1$. In addtion, when scaling depth $L$ it appears that $\alpha_L = 1$ is preferable to $\alpha_L = \frac{1}{2}$, consistent with our theory.
>
> *3. The distinction between $\alpha$ and is explained in Appendix C.4, what happens if interpolates between the two extremes?*
>
> For $\alpha_{\mathcal A}$ between $\frac{1}{2}$ and $1$, the the variables $\mathcal A = k \cdot q / N^{\alpha}$ will concentrate to zero at initialization for any $\alpha > \frac{1}{2}$ leading to the head-collapse issue as $N \to \infty$. If one tunes learning rates to make $\mathcal A$ change substantially with SGD, it will lead to a divergence in $N$ unless $\alpha < 1$. Thus $\alpha = 1$ is special as it enables stability as $N \to \infty$ and $\alpha = \frac{1}{2}$ is special as it preserves $\Theta(1)$ diversity of heads at initialization.
>
> *4. What are some previous results or potential approaches towards analyzing limiting dynamics of neural networks under Adam?*
>
> Some recent works have attempted to investigate the limiting behavior of Adam using Tensor programs https://arxiv.org/abs/2308.01814 or from an empirical scaling perspective https://openreview.net/pdf/579c102a8c067102c85e27612c36d7a356ea9b0b.pdf. One insight into these analyses is that the $\epsilon$ parameter in Adam may have to be scaled with width to obtain a reasonable limit.

---

> ### Author Response · Authors · 2024-08-06
> **Some non-trivial DMFT conclusions**
>
> ### A very simple example: GOE/Wigner Linear Dynamics
>
> In this example we show that the DMFT path integral is computing something non-trivial about the kinds of dynamics induced by a linear dynamical system with a random matrix. In this linear example, the DMFT path integral encodes spectral properties of the random matrix.
>
> Let's consider the simplest possible example: $\frac{d}{dt} h_i(t) = \frac{1}{\sqrt N} \sum_{j=1}^N W_{ij} h_j(t)$ where $W_{ij} = W_{ji}$ is a Gaussian symmetric matrix (GOE). This matrix is **fixed** while the state $h(t) \in \mathbb{R}^N$ evolves. The path integral approch would tell you that in the $N \to \infty$ limit, every neuron $i$ has identical statistics given by the stochastic integro-differential equation
> \begin{align}
>     &\partial_t h(t) = u(t) + \int_0^t ds R(t,s) h(s)  \ , \ u(t) \sim \mathcal{GP}(0, C(t,s))
>     \\
>     &C(t,s) = \left< h(t) h(s) \right> \ , \ R(t,s) = \left< \frac{\delta h(t)}{\delta u(s)} \right>
> \end{align}
> where $\left< \cdot \right>$ denotes an average over the random variables $u(t)$. This stochastic equation can be used to close the evolution equations for the correlation $C(t,s)$ and linear response function $R(t,s)$.
>
> A generic result of this path integral DMFT picture is
> 1. All neurons decouple statistically. The presence of all other neurons only enters through "macroscopic" quantities $C(t,s)$ and $R(t,s)$ known as the correlation and response functions. The distribution of these functions over random realizations satisfies a large deviations principle $p(C,R) \sim e^{- N S(C,R)}$ where $S$ is the DMFT action.
> 2. Extra *memory terms* like $\int_0^t R(t,s) h(s)$ appear which depend on the state at earlier times $s < t$. The Markovian (deterministic) system for $p(h|W)$ becomes stochastic and non-markovian after marginalizing $p(h) = \int dW p(h|W) p(W)$. I would argue these memory terms are not obvious apriori but are systematic to compute in this framework.
>
> Since this toy example is a **linear dynamical system**, one could also obtain the correlation $C(t,s)$ and response $R(t,s)= \frac{1}{N} \text{Tr} \exp\left( W (t-s) \right) = \int d\lambda \rho(\lambda) e^{ \lambda (t-s)}$ where $\rho(\lambda)$ is the eigenvalue density of $W$. In fact a Fourier transform of our DMFT equation recovers the semicircle law $\rho(\lambda) = \frac{1}{\pi} \text{Im} R(i \lambda) = \frac{1}{2\pi} \sqrt{[4-\lambda^2]_+}$ for the eigenvalues.
>
> In general, one can think of DMFT as a more powerful version of this method that can also handle nonlinearities.
>
> ### Do Memory / Response Terms Matter in Deep Networks?
>
> In this section I will try showing how this DMFT approach can give useful insights into reasoning about learning updates which are not obvious apriori (in our opinion). While our paper advocates for taking depth $L \to \infty$ in a residual network, we first thought about simply scaling depth in a standard MLP. Below we show how the proliferation of response terms gives a different predicted scaling with $L$ than if we naively disregarded response terms.
>
> Consider a non-residual linear MLP network with $\mu$P/mean-field scaling with $L$ hidden layers with $N \to \infty$. Train the model for a single step of gradient descent with learning rate $\eta$ on a data point $(x,y)$ with $|x|^2 = 1$ and $y=1$. The feature variance after $t=1$ step of gradient descent $H^{\ell} = \left< h^\ell(1)^2 \right>$ after $t=1$ step, the final layer
> \begin{align}
>     H^{L} \sim \begin{cases} 1 +  \frac{1}{3} \eta^2 \gamma_0^2 \ L^3 & \text{DMFT Response Included (Full DMFT)}
>     \\
>     1 + \eta^2 \gamma_0^2 \ L & \text{DMFT Response Neglected }
>     \end{cases}
> \end{align}
>
> We see that without the response terms we get a totally different scaling prediction with $L$!
>
>
> For $\frac{1}{\sqrt L}$ residual block scaling, the response functions are still very important and contribute $\Theta_L(1)$ corrections to the feature learning dynamics as $L \to \infty$.  However, for the $1/L$ block multiplier scaling, the response functions do not contribute in the limit. These facts are **not apriori obvious** (to us at least) but follow from the DMFT analysis (either path integral or cavity approach).

---

> > ### Comment · Reviewer_UGyb · 2024-08-09
> >
> > Thank you for the detailed explanations including interesting examples and new (to me) ideas. I will maintain my positive review of the paper.

---

### Official Review · Reviewer_CMzU · 2024-07-09

**Soundness:** 3
**Presentation:** 2
**Contribution:** 3
**Rating:** 7
**Confidence:** 2

**Summary:**

The authors investigate multi-head transformer dynamics by scaling to infinite limits in key/query dimension, heads, and depth respectively using dynamical mean field theory and discover different statistical behaviors.

**Strengths:**

- Give detailed analysis (and closed form) on dynamics of the updates
- Conduct experiments in realistic settings

**Weaknesses:**

- The paper is hard to understand, the notations are convoluted and most of the community might not be familiar with dynamical mean field theory , the authors might want to offer more background in the main text

**Questions:**

1. In line 78, it is mentioned that gamma_0 controls the rate of feature learning, but I couldn't find it in equation (1)
2. Why according to Table 1, SGD's learning rate scales with N and H, but Adam's learning rate scales down when N and H decreases, any intuition to understand this phenomenon?
3. Why 3/2 appears in eq3
4. Why the limit can be straightforwardly computed from the saddle point of DMFT action?

**Limitations:**

Limitations are addressed in the last session of the paper

---

> ### Author Rebuttal · Authors · 2024-08-06
>
> ### Strengths
>
> We thank the reviewer for appreciating these aspects of our paper and for their support.
>
> ### Weaknesses
> *The paper is hard to understand, the notations are convoluted and most of the community might not be familiar with dynamical mean field theory , the authors might want to offer more background in the main text.*
>
> Thank you for this advice. We will now provide more detail in the main text about how the DMFT works and also add a new Appendix section which gives a primer on the DMFT methods used in this work.
>
> ### Questions:
>
> *In line 78, it is mentioned that gamma_0 controls the rate of feature learning, but I couldn't find it in equation (1)*
>
> The parameter $\gamma_0$ is a constant $\Theta_{N,H,L}(1)$ hyperparameter that controls the scale of hidden feature updates in the same way described by Chizat and Bach. The $\gamma_0 \to 0$ limit gives a lazy learning limit. We borrow this notation from previous papers on the DMFT limits.
>
> To make this more clear we give the equation that defines $f$ its own line
>
> \begin{align}
>     f = \frac{1}{ \gamma_0 N \mathcal H} w^L  \cdot \left( \frac{1}{\mathcal S} \sum_{\mathfrak s}  h^L_{\mathfrak s} \right)
> \end{align}
>
>
> *Why according to Table 1, SGD's learning rate scales with N and H, but Adam's learning rate scales down when N and H decreases, any intuition to understand this phenomenon?*
>
> Yes, this phenomenon is due to the fact that Adam updates are approximately normalized. Consider a simple MLP layer. We would have a change to the weight of the size $\delta W_{ij} \sim \eta \frac{g_{i} \phi_j}{\sqrt{g_i^2 \phi_j^2 + \epsilon}}$. We need to control the size of the forward pass
> \begin{align}
>     \frac{1}{\sqrt N } \delta W \phi(h) \sim \Theta( \eta N^{1/2} )
> \end{align}
> We want this to be $\Theta(1)$ so we need $\eta \sim N^{-1/2}$. If you desire more details, check out Appendix C.3.
>
> *Why 3/2 appears in eq3?*
>
> The $3/2$ is due to the fact that feature updates have to be controlled to prevent $k \cdot q / N^{\alpha}$ from diverging. We work this out in Appendix C.2.
>
> *Why the limit can be straightforwardly computed from the saddle point of DMFT action?*
>
> We have added a more detailed Appendix section on DMFT which shows how the path integral method works in more detail on a few simple problems. In a nutshell though, the DMFT approach shows that the distribution of the feature kernels and response functions at finite $N$ induced by randomly sampling initial weights looks like $p(Q) \sim \exp\left( - N S[Q]  \right)$ where $S$ is the DMFT action and $Q = \{f , \Phi, G, R, ... \}$ are the network outputs $f$, feature kernels $\Phi$, gradient kernels $G$ and response functions $R$ (everything that should concentrate as $N \to \infty$). By a steepest descent argument, the probability density $p(Q)$ will be dominated at large $N$ by the saddle point. This is a useful idea in mean field theory and high dimensional statistics.

---

> > ### Comment · Reviewer_CMzU · 2024-08-09
> >
> > Thank you for your response! I will keep my already positive score

---

### Official Review · Reviewer_rXAV · 2024-07-13

**Soundness:** 2
**Presentation:** 3
**Contribution:** 2
**Rating:** 6
**Confidence:** 3

**Summary:**

The authors identify parameterizations that lead to nontrivial feature learning as the limits of $N, H, L \to \infty$. Specifically, the study demonstrates the following:

- Under the limit $N\to\infty$ the $\mu P$ rule is required, causing all heads to collapse into the same dynamics.
- Under the limit $H\to\infty$, each head becomes statistically independent.
- The scaling under the depth limit $L\to\infty$ is also analyzed.

**Strengths:**

- To approximately preserve the magnitude of the parameter updates, the hyperparameters (e.g., step size) must be carefully chosen based on $N,H$, and $L$. The authors determine appropriate step size scaling for SGD and Adam to ensure feature learning during training.
- The observed degeneration of multiple attention heads under the limit $N\to\infty$ is consistent with empirical observations for $\mu P$.

**Weaknesses:**

- It is unclear whether considering the limit of the number of heads $H$ is practically relevant, as $H$ is typically set to around or fewer than 100 in real-world applications. This raises questions about whether the theory accurately explains the behavior of large language models (LLMs).
- The following paper is also relevant:

  - Lénaïc Chizat, Praneeth Netrapalli. The Feature Speed Formula: A flexible approach to scale hyper-parameters of deep neural networks. 2024.

**Questions:**

- Line 78: What is $\gamma_0$? Could you elaborate on this further?
- What is the maximum model size (i.e., number of parameter) used in the experiments?
- Do the results hold for any time t? If so, this feature is worth emphasizing, as most papers, such as [L. Chizat and P. Netrapalli (2024)] and [G. Yang and E.J. Hu (2021)], focus on the initialization phase.

**Limitations:**

Limitations are addressed in the paper.

---

> ### Author Rebuttal · Authors · 2024-08-06
>
> ### Strengths
>
> We thank the reviewer for appreciating these aspects of our contributions.
>
> ### Weaknesses
>
> Below we try to clarify our limits and also add a citation to the "Feature-Speed Formula" paper and discuss the similarities and differences of the conclusions we make in this paper. We hope that in response the reviewer would be willing to increase their score.
>
> ### Practical Utility of Studying Infinite Head Transformers
>
> This is a good question. To motivate the infinite head limits we point out the following few facts.
>
> 1. We first investigated the large key-query dimension $N \to \infty$ limit of transformers with $\mathcal H$ fixed. We showed that this limit was degenerate in the sense that all attention heads collapsed to the same dynamics, causing redundant computation in a multi-head model. We thus sought another way to take "width" to infinity that would allow variability across attention heads.
> 2. In the scaling law era, model sizes keep increasing and often heads $\mathcal H$ are increased as well as depth $L$ and key/query dimension $N$ as the models are scaled up. In the [GPT-3 technical report](https://arxiv.org/abs/2005.14165) Table 2.1, we can see that the last few models are increased by scaling up heads and layer count with $N$ (they call $d_{head}$) fixed. Understanding if this scaling behavior leads to a well defined limit is therefore an interesting and potentially practical question to guarantee stable convergence.
> 3. Even if models are finite, they often share important similarities with their larger (or infinite) width counterparts (for mean field/$\mu$P scalings) and thinking about limiting behavior can be useful when designing parameterizations for width/depth. For example, optimal learning rates often transfer across models of different [widths](https://arxiv.org/abs/2203.03466) and [depths](https://arxiv.org/abs/2309.16620) when a scaling limit exists. Models [learn similar representations](https://arxiv.org/abs/2305.18411) across different model sizes. Depending on the setting or number of training steps, the infinite limit can be very descriptive of models with widths as modest as a few hundred. For Figure 4(a) we show that $\mathcal H = 16$ and $\mathcal H = 128$ models can have very similar training dynamics on CIFAR-5M.
> 4. Developing mean field theory for infinite limits (specifically the DMFT action) can enable one to obtain the [dynamics of finite size corrections](https://arxiv.org/abs/2304.03408) to the theory. These would capture an approximate evolution of the $\frac{1}{\sqrt{\mathcal H}}$ deviation from the infinite head limit if one finds the infinite head limit unrealistic.
>
> ### Feature-Speed Formula
>
> Thank you for pointing out this interesting paper.  The authors of this work develop theory for deep network Jacobians at initialization and early feature learning updates. We have added a citation to this work in the related works section.
>
> #### Desiderata of Chizat and Netrapalli
> They point out four desiderata of a scaling limit to be the following four criteria at initialization
> 1. Signal propagation
> 2. Feature learning
> 3. Loss Decay
> 4. Balanced Contributions across layers
>
> The authors analyze conditions under which these four conditions hold in large depth networks at initialization. They conclude that residual networks $(h^\ell+1 = h^\ell + \beta W^\ell \phi(h^\ell)$) with branch scale $\beta = 1/\sqrt L$ are necessary.
>
> #### Why are Our Conclusions about Depth Different?
> Some differences between our work and this work's conclusions
> 1. We consider transformer models where there are multiple layers per residual block. In these models, if the residual scale factor is $\beta = 1/\sqrt{L}$ then the hidden key/query weights $W_K^\ell, W_Q^\ell$ in the attention layer can be treated as frozen in the $L \to \infty$ limit. However, under the $\beta = 1/L$ branch scaling, these matrices update non-negligbly, leading to additional feature learning (see Figure 1(c) in Rebuttal pdf).
> 2. While each layer's initial contribution to the initial neural tangent kernel is not balanced if $\beta = 1/L$, over the course of training they will become balanced.
>
> In the related works section, we add a citation to Chizat and Netripalli and also include the sentence
>
> "In this work, we pursue large depth limits of transformers by scaling the residual branch as $L^{-\alpha_{L}}$ with $\alpha_{L} \in [\frac{1}{2},1]$ ... However, we argue that in transformers that $\alpha_L = 1$ is preferable as it enables the attention layers to update non-negligibly as $L \to \infty$."
>
>
> ### Response to Questions
>
> *Line 78: What is $\gamma_0$? Could you elaborate on this further?*
>
> The parameter $\gamma_0$ is a constant $\Theta_{N,H,L}(1)$ hyperparameter that controls the scale of hidden feature updates ([laziness/richness](https://papers.nips.cc/paper_files/paper/2019/hash/ae614c557843b1df326cb29c57225459-Abstract.html)). The $\gamma_0 \to 0$ limit gives a lazy learning limit. We borrow this notation from previous papers on the DMFT limits.
>
> To make this more clear we give the equation that defines $f$ its own line
>
> \begin{align}
>     f = \frac{1}{ \gamma_0 N \mathcal H} w^L  \cdot \left( \frac{1}{\mathcal S} \sum_{\mathfrak s}  h^L_{\mathfrak s} \right)
> \end{align}
>
> *What is the maximum model size (i.e., number of parameter) used in the experiments?*
>
> The maximum number of parameters in our CIFAR-5M plots are around $100$M parameters (which was the $\mathcal H = 2048$ model in Figure 3) while for the C4 language experiments the maximum model size was $150$M parameters.
>
> *Do the results hold for any time t? Prior works focus on the initialization phase.*
>
> We are also explicitly keeping timesteps fixed as we scale the model size (not scaling these jointly). This was mentioned in a footnote and in the limitations section, but we added an extra sentence in the main text to emphasize this.

---

> > ### Author Response · Authors · 2024-08-11
> >
> > As the discussion period is ending soon, we were hoping to follow up to see if our response answered the reviewer's questions. If so, we would hope that they would be willing to increase their score. If not, we would be happy to address any additional questions or concerns. We thank you again for your time and reviews.

---

### Official Review · Reviewer_FJYA · 2024-07-13

**Soundness:** 3
**Presentation:** 2
**Contribution:** 3
**Rating:** 7
**Confidence:** 3

**Summary:**

The authors study transformer training dynamics under various limits (infinite embedding dimension, infinite number of heads, infinite depth). They point out interesting and subtle behaviours that can happen in these limits. For example:
- taking the embedding dimension to infinity can make heads redundant with each other, while fixing embedding dimension and scaling head count avoids that issue)
- 1/depth block multipliers have less interesting kernels at the start of training than 1/sqrt(depth) but this doesn't matter after sufficient training

The authors also present technical calculations of various kernels that emerge in these limits, derived using path integrals. The authors are honest that these calculations are only expected to be relevant when training time is small compared to the other dimensions in the problem.

I'm going to be honest that I have not attempted to parse the derivations, although I have experience in the area and the results and conclusions all seem plausible and very interesting to me. Even if some of the calculations turn out to be incorrect / make flawed assumptions, I think the paper is still interesting to anyone in this subfield.

**Strengths:**

- the analysis and subtle issues raised about different ways of taking the depth limit (what block multiplier) and different ways of taking the attention limit (heads versus embedding dim) are very interesting
- the plots in the paper are really interesting, and should be of interest to someone doing practical transformer training
- the theoretical analysis involving path integrals may be an important contribution, however I am not sure here

**Weaknesses:**

I am going to provide feedback here that is intended to be constructive. I think that sometimes I can have a blunt style so I just want to start by saying that I think this paper is really cool and it was great to read it. But here goes...

### **Technical tools**

The authors apply a path integral formalism to derive their results. A major question I have about the work is "are path integrals necessary here?"... Why do you use them? It would be worth adding a section to the paper, perhaps to the introduction, explaining the relative merits of the path integral approach to other approaches that people are doing. Ideally this section should be easy to understand by someone who doesn't already know about path integrals!

Now I have physics training myself and found the path integral approach to theoretical physics very beautiful. But there are many ways to skin a cat. For example, I recall that you can take the differential equation $\ddot{x} = 0$ which is trivial to solve and is the subject of high school calculus, and throw path integrals at it. Is that what you're doing here? Or is there actually a reason why path integrals are the right way to solve this problem?

### **Exposition and clarity**

I think there are a few ways you could improve the exposition. First of all, another round of proof reading. E.g. $\gamma_0$ at the bottom of page 2 is undefined. Second, I think that whenever you mention a variable, e.g. $\mathcal{H}$, you should always precede it by the simple English word, e.g. "the number of heads $\mathcal{H}$". This helps the reader parse the paper---remember that they don't have all the same mental variable bindings as you, and it's especially important in a paper like this where there are about 15 different variables floating around. Third, you should make sure that your figure captions can be read and understood by an informed reader in isolation from the rest of the paper---I think again the main problem here is that Greek letters are often used without English signposting. I also want to point out that your plots are missing basic labelling. E.g. on figure 4 b) there is no colour bar or scale.

### **Missing related work on non-asymptotic approaches to scaling**

The authors cite a lot of related works on asymptotic approaches to scaling and learning rate transfer, for example writing that "Further, theoretical results about their limits can often be obtained using Tensor Programs [14] or dynamical mean field theory (DMFT) techniques [15, 17]." However, they do not cite or mention a body of work that works out non-asymptotic analyses of feature learning and scaling. For example, this paper arxiv.org/abs/2002.03432 broaches the topic of learning rate transfer and its relation to architectural dimensions a year before the muP work came out (and at a time when most other theory researchers were working on NTK analyses). There are clear advantages to the non-asymptotic approach in that it is easy to apply to different initialisations and different base optimisers.

**Questions:**

Please see the Weaknesses section.

**Limitations:**

I think the authors do a good job talking about limitations at the end of the paper. I think it would be helpful to the reader to clarify the mechanism of the limitations. E.g. "If training time was much larger, then even for very large embedding dimensions, initially negligible stochastic fluctuations between heads could gradually amplify and lead to different large training time behaviour than what we describe here".

By the way, I am willing to upgrade my score. But will be keen to see and engage with the opinions of the other reviewers.

---

> ### Author Rebuttal · Authors · 2024-08-06
>
> We thank the reviewer for their detailed feedback and for allowing us the chance to clarify our theoretical methods. We hope that upon implementing these proposed changes the reviewer will consider improving their score.
>
> ### Strengths
> We appreciate these comments on these strengths of our contributions!
>
> ### Weaknesses
>
> ### Exposition and Clarity
>
> We thank the reviewer for this bit of feedback. We have updated the draft to include a larger amount of signposting and exposition (blue text is what has been added since initial submission).
>
>
> ### Missing Citations to Non-Asymptotic Approaches
>
> We thank the reviewer for bringing this body of work to our attention and pointing out its absence in our paper. We now add the following to the related works
>
> "In addition to work on infinite width and depth limits of deep networks, there is also a non-asymptotic approach to optimizer design and scaling based on controlling the norm of weight updates \cite{bernstein2020distance}. This approach coincides with $\mu$P width-scaling when the spectral norm of the weights is used as the measure of distance \cite{yang2023spectral}, and can achieve hyperparameter transfer for a wide array of optimizers and initialization schemes \cite{bernstein2023automatic, large2024scalable}."
>
> We also point out that Large et al argue for $1/L$ depth scaling, which is similar to our conclusion of the way to scale up depth for transformers.
>
> ### Technical Tools and Path Integral Approach
>
> *A major question I have about the work is "are path integrals necessary here?"... Why do you use them?*
>
> This is a great question!
>
> If one is mainly interested in arguing about whether feature and NN logit updates are $\Theta_{N,L,\mathcal H}(1)$ under a given parameterization, then the path integral approach is not necessary. However, if one wants to characterize the exact limits we consider when initializing randomly, then the DMFT method is useful. However, even to derive the limit the path integral method is not the only way to obtain the correct limits (the **dynamical cavity** method also works).
>
> In response, we have added a new Appendix section D where we explained more clearly what the path integral is computing by giving some simpler examples (see the response section "Simple Path Integral Examples"). We will also provide a companion derivation using the **dynamical cavity method** in a simpler setting (generic ResNet with no MHSA blocks) and show that it agrees with the path integral computation. This is very similar to the computation of [Bordelon et al '24](https://arxiv.org/abs/2309.16620) on infinite depth ResNets. Below, we will provide more details about how the path integral is non-vacuous and more systematic than the cavity approach.
>
> ### DMFT Path Integral is Non-Vacuous
> DMFT is useful primarily in settings where there is a source of randomness in a high-dimensional dynamical system that *correlates state vectors across time*. The path integral approach gives one the exact asympotic description of the limiting dynamics and can also give finite size corrections. In our problem of interest (training dynamics of transformers) the randomness comes from the random initial weights in each layer, while the states are the features computed on forward and backward passes. In this setting the use of path integrals is far from vacuous (like the $\ddot x = 0$ example).
>
> This formalism is very flexible and when the dynamics correspond to some kind of optimization procedure it can be viewed as an alternative to other disordered systems methods (replica method, etc) which *respects the choice of initialization and optimizer*. Some recent examples are studying SGD/momentum with [random data on general linear models](https://arxiv.org/abs/2006.06098) or the [training dynamics of random feature models](https://arxiv.org/abs/2402.01092). We will provide a couple very simple examples that gives intuition for what this approach is computing (see the comment below). It can also be used for problems where there is no equilibrium distribution (like [random RNNs](https://arxiv.org/abs/1809.06042)) or [non-gradient descent learning rules](https://arxiv.org/abs/2210.02157).
>
>
> #### Cavity Method Alternative
> While the path integral approach gets the correct answer, there is an alternative **cavity method** derivation that provides a different set of intuitions while recovering the same limiting dynamics. We will include a simple cavity derivation in Appendix D. The idea of the cavity method is to consider what happens when a single neuron is added to the residual stream or one of the hidden layers. This new neuron gives small corrections to all other neurons, but these add up to give the extra response terms.
>
>
> #### Merits of Path Integral Compared to Cavity Method
> 1. It starts by giving the non-asymptotic distribution of the feature kernels and response functions at finite $N$ induced by randomly sampling initial weights. This distribution looks like $p(Q) = \frac{1}{Z} \exp\left( - N S[Q]  \right)$ where $S$ is the DMFT action and $Q = \{f , \Phi, G, R, ... \}$ are the network outputs $f$, feature kernels $\Phi$, gradient kernels $G$ and response functions $R$ (everything that should concentrate as $N \to \infty$).
> 2. While the $N \to \infty$ limit is computed as a saddle point (first derivative) of the DMFT action $\frac{\partial S}{\partial Q} = 0$), finite width corrections can also be obtained from [higher order derivatives](). One can track the leading order $\mathcal{O}(N^{-1})$ corrections to the dynamics of $Q$ from the Hessian $\frac{\partial^2 S}{\partial Q \partial Q}$.
> 3. The cavity method often requires "having a sense of the final result" before doing the computation. The computation is easy if you already possess intuition for the kind of mean-field limit you expect to get and what response functions will appear. The path integral method is more systematic and requires less mean field theory "intuition".

---

> ### Author Response · Authors · 2024-08-06
> **Some Non-Vacuous Path Integral Examples**
>
> ### A very simple example: GOE/Wigner Linear Dynamics
>
> In this example we show that the DMFT path integral is computing something non-trivial about the kinds of dynamics induced by a linear dynamical system with a random matrix. In this linear example, the DMFT path integral encodes spectral properties of the random matrix.
>
> Let's consider the simplest possible example: $\frac{d}{dt} h_i(t) = \frac{1}{\sqrt N} \sum_{j=1}^N W_{ij} h_j(t)$ where $W_{ij} = W_{ji}$ is a Gaussian symmetric matrix (GOE). This matrix is **fixed** while the state $h(t) \in \mathbb{R}^N$ evolves. The path integral approch would tell you that in the $N \to \infty$ limit, every neuron $i$ has identical statistics given by the stochastic integro-differential equation
> \begin{align}
>     &\partial_t h(t) = u(t) + \int_0^t ds R(t,s) h(s)  \ , \ u(t) \sim \mathcal{GP}(0, C(t,s))
>     \\
>     &C(t,s) = \left< h(t) h(s) \right> \ , \ R(t,s) = \left< \frac{\delta h(t)}{\delta u(s)} \right>
> \end{align}
> where $\left< \cdot \right>$ denotes an average over the random variables $u(t)$. This stochastic equation can be used to close the evolution equations for the correlation $C(t,s)$ and linear response function $R(t,s)$.
>
> A generic result of this path integral DMFT picture is
> 1. All neurons decouple statistically. The presence of all other neurons only enters through "macroscopic" quantities $C(t,s)$ and $R(t,s)$ known as the correlation and response functions. The distribution of these functions over random realizations satisfies a large deviations principle $p(C,R) \sim e^{- N S(C,R)}$ where $S$ is the DMFT action.
> 2. Extra *memory terms* like $\int_0^t R(t,s) h(s)$ appear which depend on the state at earlier times $s < t$. The Markovian (deterministic) system for $p(h|W)$ becomes stochastic and non-markovian after marginalizing $p(h) = \int dW p(h|W) p(W)$. I would argue these memory terms are not obvious apriori but are systematic to compute in this framework.
>
> Since this toy example is a **linear dynamical system**, one could also obtain the correlation $C(t,s)$ and response $R(t,s)= \frac{1}{N} \text{Tr} \exp\left( W (t-s) \right) = \int d\lambda \rho(\lambda) e^{ \lambda (t-s)}$ where $\rho(\lambda)$ is the eigenvalue density of $W$. In fact a Fourier transform of our DMFT equation recovers the semicircle law $\rho(\lambda) = \frac{1}{\pi} \text{Im} R(i \lambda) = \frac{1}{2\pi} \sqrt{[4-\lambda^2]_+}$ for the eigenvalues.
>
> In general, one can think of DMFT as a more powerful version of this method that can also handle nonlinearities.
>
> ### Do Memory / Response Terms Matter in Deep Networks?
>
> In this section I will try showing how this DMFT approach can give useful insights into reasoning about learning updates which are not obvious apriori (in our opinion). While our paper advocates for taking depth $L \to \infty$ in a residual network, we first thought about simply scaling depth in a standard MLP. Below we show how the proliferation of response terms gives a different predicted scaling with $L$ than if we naively disregarded response terms.
>
> Consider a non-residual linear MLP network with $\mu$P/mean-field scaling with $L$ hidden layers with $N \to \infty$. Train the model for a single step of gradient descent with learning rate $\eta$ on a data point $(x,y)$ with $|x|^2 = 1$ and $y=1$. The feature variance after $t=1$ step of gradient descent $H^{\ell} = \left< h^\ell(1)^2 \right>$ after $t=1$ step, the final layer
> \begin{align}
>     H^{L} \sim \begin{cases} 1 +  \frac{1}{3} \eta^2 \gamma_0^2 \ L^3 & \text{DMFT Response Included (Full DMFT)}
>     \\
>     1 + \eta^2 \gamma_0^2 \ L & \text{DMFT Response Neglected }
>     \end{cases}
> \end{align}
>
> We see that without the response terms we get a totally different scaling prediction with $L$!
>
>
> For $\frac{1}{\sqrt L}$ residual block scaling, the response functions are still very important and contribute $\Theta_L(1)$ corrections to the feature learning dynamics as $L \to \infty$.  However, for the $1/L$ block multiplier scaling, the response functions do not contribute in the limit. These facts are **not apriori obvious** (to us at least) but follow from the DMFT analysis (either path integral or cavity approach).

---

> > ### Author Response · Authors · 2024-08-11
> >
> > As the discussion period is ending soon, we were hoping to follow up to see if our response answered the reviewer's questions. If so, we would hope that they would be willing to increase their score. If not, we would be happy to address any additional questions or concerns. We thank you again for your time and reviews.

---

> > > ### Comment · Reviewer_FJYA · 2024-08-11
> > >
> > > Hi---sorry, I've been engaging a lot with 4 other papers in my batch, and it's prevented me from yet reaching yours. I plan to respond thoroughly, so stay tuned, but for now consider this acknowledgement of your rebuttal.

---

> > ### Comment · Reviewer_FJYA · 2024-08-13
> >
> > Hi---I'm sorry for the delay.
> >
> > I think this is a good piece of science, and the principles you raise are very very interesting. I'm not sure how rigorous your calculations are (again, I haven't checked them) but I certainly think this paper is worth presenting on a scientific level. I want to share some thoughts on your work that are intended to be well-meaning although perhaps provocative:
> >
> > - I talked to a mathematician friend about path integrals, and he told me mathematicians view them as non-rigorous and don't do them. He told they worked out alternate ways to do the same calculations using Morse theory or Floer homology. I don't know to what extent this is academic tribalism or whether there's something to it, but I thought I'd share the perspective.
> >
> > - I think that if you want the wider community to seriously engage with your methods, you might need to try to find a killer app still and also put a lot of effort into honing the presentation. I say this since I do have the sense that there are other ways to do the scaling calculations that you're doing, and perhaps simpler ones. Of course I may be wrong about this, and again I do like the science that you're doing.
> >
> > Regarding your rebuttal, I think the changes that you've committed to will strengthen the paper, and I'm comfortable increasing my score.

---

> > > ### Author Response · Authors · 2024-08-13
> > >
> > > Thank you for your comments! We think including the alternative cavity derivation of our DMFT will be worth including as this approach (in other settings at least) can be made rigorous, such as in the work https://arxiv.org/abs/2210.06591.
> > >
> > > We will also aim to improve the presentation where possible and motivate our calculations more clearly.

---

### Author Rebuttal · Authors · 2024-08-06

## Global Response

We thank the reviewers for all of their detailed comments and advice on ways to improve the paper. Below we go through some of the concerns which arose in comments from many reviewers and outline how we plan to address them in the newer version of the draft.

### Repeated Concerns

1. Path integrals / DMFT / Saddle points may not be familiar to the ML audience so some additional exposition about these would be useful. We also point out that deriving the DMFT action (at least in principle) gives a procedure to also extract finite size effects.
2. What is the motivation for taking infinite limits?
3. Exposition and labeling: we have tried making the paper more readable with additional signposting and better labeling on Figures and Figure captions. Some variables (such as $\gamma_0$) are not well motivated or defined.
4. Some missing citations to relevant prior works.
5. Request for additional clarification of the limitations of our theory.
6. Compute optimal comparisions: how do these parameterizations or scaling limits compare as compute is varied?

### Updates to the Paper in Response

In response to these issues, we will make the following updates which will appear in any future version of the paper.

1. We have added a short expository section in the main text and a new Appendix section which gives a primer on DMFT methods and what the path integral approach is computing. In addition, we provide simple but qualitatively similar examples to deep network dynamics where the path integral gives the correct limiting stochastic dynamics of hidden neurons. We also now provide a companion derivation of the limit using the **dynamical cavity method** showing a physical interpretation of the response functions.
2. We try motivating infinite limits since (1) models improve in performance as parameter count increases (2) infinite limits can be descriptive of finite models (3) theory for infinite models can often be extended to approximate finite models.
3. We aim to clean up the exposition. Before using a letter $N,\mathcal{H},L$ we preempt with "number of attention heads $\mathcal{H}$". We also will fix some legibility and colorbar issues with our plots.  We now explicitly define the feature learning scale $\gamma_0$ which is controls the [laziness/richness](https://papers.nips.cc/paper_files/paper/2019/hash/ae614c557843b1df326cb29c57225459-Abstract.html) of the training. This notation is adopted following [prior works on mean field limits](https://iopscience.iop.org/article/10.1088/1742-5468/ad01b0/meta).
4. We have added citations to works on non-asymptotic approaches to stable training across widths and depths (including Bernstein et al 2021 and the Modula paper from Large et al 2024) which have an interesting alternative non-asymptotic perspective on hyperparameter transfer and width/depth scaling. We also now cite and discuss the "Feature-speed formula" paper from Chizat and Netrapalli which discusses the stable $\frac{1}{\sqrt L}$ branch scaling in residual networks in the feature learning regime. See the Attached Figure 1c in rebuttal for information about why $\alpha_{L}=1$ uniquely allows attention layers to update.
5. We now emphasize that the derived theory holds for fixed training horizons (training time is treated as a constant that is not scaling jointly with width/heads/depth). As reviewer FJYA points out, finite size effects from the stochastic initialization can accumulate over time.
6. We provide some plots of performance as a function of compute in our experiments in the attached rebuttal document. We find that scaling $\mathcal H$ is preferable to scaling $N$ in the parameterizations that admit suitable limits. In addition, we also find that $\alpha_L = 1$ is preferable to $\alpha_L = \frac{1}{2}$, consistent with our theory. We plan to add these and more experiments like these to the paper.

### Added Expository paragraph on DMFT in Main Text

"To obtain the exact infinite limits of interest when scaling dimension-per-head $N$, the number of heads $\mathcal H$, or the depth $L$ to infinity, we work with a tool from physics known as dynamical mean field theory (DMFT). Classically, this method has been used to analyze high dimensional disordered systems such as spin glasses, random recurrent neural networks, or learning algorithms with high dimensional data. We use this method to reason about the dynamics of randomly initialized neural networks by tracking a set of deterministic correlation functions (feature and gradient kernels) as well as response functions (see Appendix D). The core conceptual idea of this method is that in the infinite limit, all neurons remain statistically independent throughout training and only interact through collective variables (feature kernels, neural network outputs, etc). Collective variables are *averages* over the distribution of neurons in each hidden layer or along the residual stream."

### Added Sentence about Fixed Training Time

At the bottom of page 2 we added the sentence
"The analysis of these limits is performed with batch size and number of training steps $t$ fixed while the other architectural parameters are taken to infinity."

### New Appendix Section

Our new Appendix D "Primer on DMFT" will contain more detailed information about DMFT and the path integral method. We motivate DMFT as a general technique for dealing with dynamical systems that depend on **fixed sources of randomness**. We provide a few simple examples where one can see that the resulting stochastic process is non-trivial including

1. A linear dynamical system driven by a random matrix
2. Feature updates in deep linear neural networks and residual networks.

In both of these settings, the DMFT linear-response functions give non-trivial corrections to the limiting dynamics.

We are also adding information about the alternative **dynamical cavity** method to derive the DMFT equations which does not require the use of path integrals.

---

### Decision · Program_Chairs · 2024-09-25

**Decision:**

Accept (poster)

**Comment:**

This paper provides various scaling limit of multi-head attention in the feature learning regime as $N,H,L \to \infty$. In the limit of $N \to \infty$, the $\mu P$ parameterization is required, and the DMFT is applied to derive the training dynamics in the limit. Some numerical experiments are provided to show their convergence.

The analysis provides a precise description of the infinite limit of $N,H,L$. Although we need a careful setting of parameter scaling to achieve feature learning, this paper provides a comprehensive theoretical investigation to this issue.
Overall, the paper's technique is novel. Its contribution would be appreciated by the community.

As a summary, I think this paper is worth publication in NeurIPS. On the other hand, I recommend the authors to proofread the paper again.